# CLASSIFICATION VS. DEEP FEATURE LEARNING IN NORMALIZED SPACES WITH DIFFERENT SCALING

## ABSTRACT

In supervised scenarios, deep feature learning is typically implemented through the training of classification models. However, it should be noted that classification reflects the sample-wise local properties of models on a dataset, while deep feature learning aims to learn features with good sample-independent global properties such as intra-class compactness and inter-class separability on the dataset. This paper conducts an in-depth comparison of classification and deep feature learning in normalized spaces. We first reformulate the binary cross-entropy (BCE) loss aligning with the fundamental requirements of feature learning; then, we theoretically analyze and compare its minima with that of the cross-entropy (CE) loss used for classification tasks. Informed by the above analysis, we explore the convergence behavior of the two losses when the scale factor $\gamma$ changes, revealing the differences between classification and deep feature learning. Specifically, when $\gamma$ increases linearly, the convergence rate of the unbiased decision scores decay exponentially, resulting in poor feature properties for the trained models, although it does not affect their classification. As $\gamma$ decreases, the scores more readily reaches their global optimal value, which helps to improve the feature properties. However, if $\gamma > 0$ decreases linearly and approaches zero, the convergence rate of the unbiased decision scores decay linearly, leading to unsatisfactory feature properties and making the models' classification performance highly sensitive to minor disturbances. Our experiments fully validate these conclusions. The experimental results also demonstrate the advantages of using BCE over CE in more challenging scenarios such as long-tailed recognition and open-set recognition.

## 1 INTRODUCTION

Classification and deep feature learning are fundamental tasks in machine learning. In deep learning, classification tasks commonly use cross-entropy (CE) loss to train deep feature extraction models and linear classifiers. The classification methods effectively learn desirable features, making the trained feature extraction models widely applicable in downstream fields that require features with well properties such as face recognition (Liu et al., 2023a), object detection and tracking (Jain et al., 2024), image segmentation (Azad et al., 2024), and image-text retrieval (Vasu et al., 2024), etc.

For deep feature learning, while feature properties required in different scenarios are various, intra-class compactness and inter-class separability are most important in multi-class tasks. Intuitively, higher classification accuracy usually implies better feature properties, and conversely, better feature properties suggest higher classification accuracy. Despite that, it should be noted that classification accuracy of a model is calculated by checking whether each sample is correctly classified, reflecting the local properties of the model on a dataset. In contrast, intra-class compactness and inter-class separability are the global properties of the model on the whole dataset. In other words, there is an essential difference between classification and deep feature learning. However, there has not been a thorough analysis of this difference, and in supervised scenarios, deep feature learning is typically implemented through classification without considering global constraints on the sample features.

Normalizing features when training models can improve the training stability, accelerate the convergence, and enhance the models' generalization, which is a commonly used regularization strategy in various fields such as face recognition (Wang et al., 2017), anomaly detection (Reiss & Hoshen, 2023), and out-of-distribution detection (Regmi et al., 2024), etc. In the normalized space, as the

most commonly used loss function in classification, the normalized CE loss has been shown to lead to **neural collapse (NC)** (Papyan et al., 2020; Yaras et al., 2022), which means that when it reaches its minimum, it maximizes the intra-class compactness and inter-class separability of the sample features. This conclusion allows for the confident use of the CE loss to train feature extraction models through training classification models. However, as a classification loss, the CE loss is not fully effective in more challenging scenarios, such as class imbalance and open-set problems.

In the normalized space, the normalized features are typically multiplied by a scale factor $\gamma$ before being input into the loss, which helps to improve the model's performance. It is generally believed that a large $\gamma$ sharpens the probability distribution of features belonging to the true class (Zheng & Yang, 2024), enhancing the model's confidence in its classification decisions, while a small one smooths the probability distribution, suppressing noise effects. In contrastive learning (Hinton et al., 2015; He et al., 2020), the temperature coefficient $\tau$, which is applied to the denominator of the feature cosine similarities, serves a similar purpose to the scale factor $\gamma$. While the above empirical observations exist, there is currently a lack of in-depth theoretical understanding of the scale factor or a convincing explanation of how it affects the deep feature learning.

In this paper, for the first time, we derive a loss function from the basic requirements of deep feature learning, which coincidentally is the binary cross-entropy (BCE) loss used for multi-class tasks. We then conduct an in-depth comparison between the CE and BCE losses in the normalized space, demonstrating that the BCE can also lead to NC. Furthermore, in the supervised, normalized feature spaces with closed-set scenarios, we reveal the differences between the tasks of classification and deep feature learning by comparing the convergence rates of the CE and BCE when the scale factor varies. In future, we will analyze the deep feature learning in other scenarios. In summary:

- We reformulate the BCE loss with the fundamental requirements of deep feature learning, and then theoretically compare the CE and BCE losses in the normalized space, demonstrating that the BCE loss can also lead to NC, i.e., when it reaches a minimum, it maximizes the intra-class compactness and inter-class separability of the features. Furthermore, we point out that, regarding the classifier biases, the CE loss has infinitely many minimum points, while the BCE has only one; therefore, during the model training, the classifier biases of the BCE play a substantially role in the feature learning, whereas that of the CE do not.

- We make an in-depth analysis of the convergence behavior of the CE and BCE as the scale factor $\gamma$ changes, to reveal the differences between classification and deep feature learning. As $\gamma$ increases linearly, the convergence rates of the two losses may exponentially decay. Therefore, when $\gamma$ is very large within a large normalized space, classification performs well while feature learning performs poorly. Conversely, as $\gamma$ linearly decreases towards zero, the convergence rates decrease linearly. Then, when $\gamma > 0$ is very small, it is unsuitable for deep feature learning and even less so for classification in a very small space.

- We conduct extensive experiments using CNNs and Transformer. The experimental results indicate that, when the scale factor $\gamma = 64$ is very large, both ResNet and ViT achieve 100% classification accuracy on the training set, but the intra-class compactness and inter-class separability of features they extract are comparatively poor. When $\gamma = 0.1$ is very small, the models' feature properties are unsatisfactory but superior to those at $\gamma = 64$, while the classification performs very poor. When $\gamma$ is at a moderate level, the models' classification and feature properties can both reach their optimal points. Furthermore, compared to the CE loss, the BCE loss achieves better results on long-tailed recognition and open-set recognition.

## 2 RELATED WORKS

**Classification and deep feature learning.** The success of deep models such as ResNet (He et al., 2016) and ViT (Dosovitskiy et al., 2021) in classification has continually driven the development of deep learning. The deep classification models have been consistently developed, including DenseNet (Huang et al., 2017), MobileNet (Howard et al., 2017), Swin Transformer (Liu et al., 2021), and ConvNeXt (Liu et al., 2022), etc., all of them consist of deep feature extraction models with CNN or Transformer architectures along with linear classifiers. In supervised scenarios, employing the CE loss and training these models on large amounts of labeled data can yield good classification alongside robust feature extraction capabilities, making them widely applicable in fields such as computer vision, natural language processing, and multi-modal large models.

For the tasks that require better feature properties, such as face recognition and long-tailed recognition, normalizing sample features and the classifier vectors can effectively enhance the performance of deep models. NormFace (Wang et al., 2017) and SphereFace (Liu et al., 2017) are among the earliest face recognition works, and advanced methods such as CosFace (Wang et al., 2018), ArcFace (Deng et al., 2019), TopoFR (Dan et al., 2024), and GFace (Zhao et al., 2025) continue to be developed in the normalized space. On imbalanced datasets, normalization can suppress the scale differences between the classifier vectors, thereby enhancing classification accuracy (Liu et al., 2020). In current, various re-balancing strategies have been designed based on the normalized CE and are widely applied in long-tailed recognition (LTR) tasks (Li et al., 2023; Han, 2023; Liang et al., 2024; Chen et al., 2025).

**Scale factor and temperature.**   In the normalized space, a scale factor $\gamma$ (Wang et al., 2017) is typically used to adjust the size of feature space, where it is applied to the denominator in contrastive learning (He et al., 2020), referred to as temperature coefficient $\tau$. An inappropriate scale factor can significantly degrade the model performance, yet there is no clear and reasonable explanation for its effects. A commonly circulated notion is that large scale factor sharpens the probability distribution of the models' outputs, increasing the confidence in the classification decisions, while the small one smooths the probability distribution, helping to mitigate the effects of noise and other influences. Then, in the classification losses, the scale factor is usually greater than 1, with a typical value of $\gamma = 64$ in face recognition (Deng et al., 2019; Zhou et al., 2023). The temperature used in the contrastive learning can also scale the models' outputs, leading to a similar role as that of the scale factor, and in the contrastive learning, the temperature in the denominator is usually set to a value less than 1. In this paper, for the first time, we reveal the role of the scale factor in the deep feature learning, which also helps us understand the temperature coefficient.

**Neural collapse (NC).**   Papyan et al. (2020) first discovered that at the terminal phase of deep model training, the sample features and classifier vectors form a simple geometric structure, which includes (1) NC1, the features of each class converge to their class center; (2) NC2, the class centers form an simplex equiangular tight frame (ETF) with equal and maximized cosine distance between every pair of them; (3) NC3, the class center is ideally aligned with the classifier vector.

The current theoretical works on NC primarily revolves around the mean squared error loss (Han et al., 2022) and CE loss (Zhu et al., 2021; Lu & Steinerberger, 2022b; Yaras et al., 2022). The NC studies on CE loss has been extended into the scenarios such as class imbalance (Fang et al., 2021; Dang et al., 2024; Gao et al., 2024), out-of-distribution data (Chen et al., 2024), and fixed classifiers (Kim & Kim, 2024), as well as the focal loss, label smoothing loss. Li et al. (2025) compare BCE and CE from perspective of NC in Euclidean space. In the normalized space, Lu & Steinerberger (2022a) and Yaras et al. (2022) theoretically analyzed the CE loss and proved that its global minima are all NC solutions. In this paper, we will reformulate the BCE in normalized space and compare its minima with that of the CE to reveal the difference between classification and deep feature learning.

## 3 PRELIMINARIES

Let $\mathcal{D} = \bigcup_{k=1}^{K} \left\{ \boldsymbol{X}_i^{(k)} \right\}_{i=1}^{n_k}$ be a sample set captured from $K$ classes, where $\boldsymbol{X}_i^{(k)}$ is the $i$-th sample of the $k$-th class and $n_k$ is number of samples in the class. In deep learning, for $\forall \boldsymbol{X} \in \mathcal{D}$, a deep feature extractor $\mathcal{M}$ maps the sample into its feature $\boldsymbol{h} = \mathcal{M}(\boldsymbol{X}) \in \mathbb{R}^d$, where $d$ is feature dimension.

For classification, a classifier $\mathcal{C}$ converts the feature into a **decision vector** $\boldsymbol{z} \in \mathbb{R}^K$, i.e., $\boldsymbol{z} = \mathcal{C}\big(\mathcal{M}(\boldsymbol{X})\big) = \mathcal{C}(\boldsymbol{h})$. In deep networks, the classifier $\mathcal{C}$ is commonly represented by a fully connection layer, which contains a weight matrix $\boldsymbol{W} = [\boldsymbol{w}_1, \boldsymbol{w}_2, \cdots, \boldsymbol{w}_K]^\top \in \mathbb{R}^{K \times d}$ and a bias vector $\boldsymbol{b} = [b_1, b_2, \cdots, b_K]^\top \in \mathbb{R}^K$, then $\boldsymbol{z} = \mathcal{C}(\boldsymbol{h}) = \boldsymbol{W}\boldsymbol{h} - \boldsymbol{b}$. In normalized feature space, both sample features and classifier vectors are normalized to $\|\boldsymbol{h}\|_2 = 1$ and $\|\boldsymbol{w}_j\|_2 = 1$, for $\forall j$. In practice, the normalization is usually implemented by dividing the vectors using their Euclidean norms, and a scale factor $\gamma > 0$ is multiplied on the decision vector to adjust the feature space, i.e., $\boldsymbol{z} = \gamma\boldsymbol{W}\boldsymbol{h} - \boldsymbol{b}$ with $\|\boldsymbol{w}_j\|_2 = \|\boldsymbol{h}\|_2 = 1$.

### 3.1 CLASSIFICATION

In a classification task, for $\forall \boldsymbol{X} \in \mathcal{D}$ with feature $\boldsymbol{h} = \mathcal{M}(\boldsymbol{X})$, its predicted class label $\hat{k}$ is decided by the vector $\boldsymbol{z} = [\gamma\boldsymbol{w}_j^\top \boldsymbol{h} - b_j]_{j=1}^K$ and $\hat{k} = \arg\max_j \{\gamma\boldsymbol{w}_j^\top \boldsymbol{h} - b_j\}_{j=1}^K$. For any sample $\boldsymbol{X}^{(k)}$ from

class $k$, in its decision vector $\boldsymbol{z}^{(k)} = \gamma \boldsymbol{W} \boldsymbol{h}^{(k)} - \boldsymbol{b}$, we refer to the $k$-th component $\gamma \boldsymbol{w}_k^\top \boldsymbol{h}^{(k)} - b_k$ as its **positive decision score** and the others $\{\gamma \boldsymbol{w}_j^\top \boldsymbol{h}^{(k)} - b_j\}_{j \neq k}$ as **negative decision scores**. Then, for correct classification, the positive decision score should be larger than all the negative ones,

$$\gamma \boldsymbol{w}_k^\top \boldsymbol{h}^{(k)} - b_k > \max\{\gamma \boldsymbol{w}_j^\top \boldsymbol{h}^{(k)} - b_j\}_{j \neq k}. \tag{1}$$

During the model training, the decision vector $\boldsymbol{z}^{(k)}$ is transformed using Softmax into the predicted probabilities $\left\{\hat{p}_j = \frac{\exp(\gamma \boldsymbol{w}_j \boldsymbol{h}^{(k)} - b_j)}{\sum_{\ell=1}^K \exp(\gamma \boldsymbol{w}_\ell \boldsymbol{h}^{(k)} - b_\ell)}\right\}_{j=1}^K$ of the sample belonging to each class. For $\boldsymbol{X}^{(k)}$, its true probabilities belonging to each class are $p_k = 1$ and $p_j = 0$ for $j \neq k$. Then, cross-entropy (CE) loss is calculated and minimized to drive the model training,

$$\mathcal{L}_{\text{ce}}(\boldsymbol{z}^{(k)}) = -\sum_{j=1}^K p_j \log(\hat{p}_j) = -\log\left(\frac{\mathrm{e}^{\gamma \boldsymbol{w}_k^\top \boldsymbol{h}^{(k)} - b_k}}{\sum_{\ell=1}^K \mathrm{e}^{\gamma \boldsymbol{w}_\ell^\top \boldsymbol{h}^{(k)} - b_\ell}}\right). \tag{2}$$

Classification accuracy $\mathcal{A}$ is calculated by checking whether each sample is correctly classified via Eq. (1), equal to ratio of number of correctly classified samples to sample number of dataset $\mathcal{D}$. Then, the classification performance of $\mathcal{M}$ and $\mathcal{C}$ reflect their *sample-wise* local property on the dataset.

## 3.2 Deep Feature Learning

In supervised settings, deep features can be learned during the training of classification models using the CE loss, while this loss does not take into account the sample-independent global constrains required for inter-class compactness and inter-class separability of all sample features on dataset.

For the sample features $\{\boldsymbol{h}_i^{(k)} = \mathcal{M}(\boldsymbol{X}_i^{(k)})\}_{i=1}^{n_k}$ of class $k$, taking the $k$-th classifier vector $\boldsymbol{w}_k$ as an anchor, then the high intra-class compactness requires that all the features should lie close to $\boldsymbol{w}_k$. To explicitly measure this closeness, we take a unified threshold $t_k'$ as small as possible satisfying $\max\{\|\boldsymbol{w}_k - \boldsymbol{h}_i^{(k)}\|_2^2\}_{i=1}^{n_k} < t_k'$, which as $\|\boldsymbol{w} - \boldsymbol{h}\|_2^2 = 2 - 2\boldsymbol{w}^\top \boldsymbol{h}$ in the normalized feature space, is equivalent to a unified threshold $t_k = 1 - t_k'/2$ as large as possible satisfying

$$\min\{\boldsymbol{w}_k^\top \boldsymbol{h}_i^{(k)}\}_{i=1}^{n_k} > t_k. \tag{3}$$

Similarly, to measure the inter-class separability between the class $k$ and class $j$, we take a unified threshold $t_j$ to separate the features of class $k$ and the $j$-th anchor/classifier vector $\boldsymbol{w}_j$,

$$\max\{\boldsymbol{w}_j^\top \boldsymbol{h}_i^{(k)}\}_{i=1}^{n_k} < t_j, \quad \forall j \neq k. \tag{4}$$

Clearly, according to Eqs. (3) and (4), desirable feature properties require that all positive unbiased decision scores uniformly exceed as large a threshold as possible, while all negative ones uniformly remain below as small a threshold as possible, which are *sample-independent* on the whole dataset.

Set $b_k = \gamma t_k$. In the model training, for $\forall \boldsymbol{X}^{(k)}$ from class $k$, applying Sigmoid on its positive decision score, one can compute the predicted probability that it satisfies Eq. (3), $\hat{p}_{kk} = \sigma(\gamma \boldsymbol{w}_k^\top \boldsymbol{h}^{(k)} - b_k)$; similarly, applying Sigmoid on its negative decision scores, one can compute the predicted probabilities that it satisfies Eq. (4), $\hat{p}_{kj} = 1 - \sigma(\gamma \boldsymbol{w}_j^\top \boldsymbol{h}^{(k)} - b_j)$. Then, for well feature properties, we calculate $K$ binary cross-entropy (BCE) losses for the sample and minimizes them together,

$$\mathcal{L}_{\text{bce}}(\boldsymbol{z}^{(k)}) = -\sum_{j=1}^K \log \hat{p}_{kj} = \log\left(1 + \mathrm{e}^{-\gamma \boldsymbol{w}_k^\top \boldsymbol{h}^{(k)} + b_k}\right) + \sum_{\substack{j=1 \\ j \neq k}}^K \log\left(1 + \mathrm{e}^{\gamma \boldsymbol{w}_j^\top \boldsymbol{h}^{(k)} - b_j}\right). \tag{5}$$

This BCE loss can also be deduced by decomposing a multi-class classification into multiple binary classification tasks, which has been widely used in multi-label classification (Kobayashi, 2023). We here reformulate it based on the global constraints of deep feature learning, directly revealing for the first time its connection to the global properties of sample features on the whole dataset.

## 4 Main Theoretical Results

In this section, we first theoretically analyze the minima of the CE and BCE in normalized space, and then compare classification and feature learning tasks by varying the scale factor $\gamma$.

### 4.1 Minima of Normalized BCE and CE

Following Zhu et al. (2021) and Yaras et al. (2022), we simplify the analysis by using unconstrained feature model (UFM) on balanced dataset. Specifically, we take the sample features $\bigcup_{k=1}^{K} \{h_i^{(k)}\}_{i=1}^{n_k}$, classifier vectors $\{w_j\}_{j=1}^{K}$, and classifier biases $\{b_j\}_{j=1}^{K}$ as free variables, without considering the parameters within the feature extractor $\mathcal{M}$, and assume that $n = n_k$, for $\forall k \in [K]$. Let

$$f_\mu(\boldsymbol{W}, \boldsymbol{H}, \boldsymbol{b}) = \frac{1}{nK} \sum_{k=1}^{K} \sum_{i=1}^{n} \mathcal{L}_\mu(\boldsymbol{z}_i^{(k)}), \tag{6}$$

where $\mu \in \{\mathrm{ce}, \mathrm{bce}\}$ indicating the normalized CE and BCE losses in Eqs. (2) and (5), and

$$\boldsymbol{H} = \begin{bmatrix} \boldsymbol{H}_1, \boldsymbol{H}_2, \cdots, \boldsymbol{H}_K \end{bmatrix} \in \mathbb{R}^{d \times (nK)} \quad \text{with} \quad \boldsymbol{H}_k = \begin{bmatrix} \boldsymbol{h}_1^{(k)}, \boldsymbol{h}_2^{(k)}, \cdots, \boldsymbol{h}_n^{(k)} \end{bmatrix} \in \mathbb{R}^{d \times n}. \tag{7}$$

For the minima of normalized CE and BCE, we achieve the following theorems.

**Theorem 1.** *Suppose $d \geq K - 1$, i.e., the feature dimension is greater than the number of classes. The loss function $f_{\mathrm{ce}}(\boldsymbol{W}, \boldsymbol{H}, \boldsymbol{b})$ defined using $\mathcal{L}_{\mathrm{ce}}$ in Eq. (2) satisfies*

$$f_{\mathrm{ce}}(\boldsymbol{W}, \boldsymbol{H}, \boldsymbol{b}) \geq \log\left(1 + (K-1)e^{-\frac{\gamma K}{K-1}}\right), \tag{8}$$

*and the equality is attained if and only if the minimizers $(\boldsymbol{W}^\star, \boldsymbol{H}^\star, \boldsymbol{b}^\star)$ satisfy*

$$\boldsymbol{w}_k^\star = \boldsymbol{h}_i^{(k)\star}, \forall k \in [K], i \in [n], \quad \boldsymbol{b}^\star = b^\star \mathbf{1}_K, \quad \text{and} \quad \boldsymbol{W}^\star \boldsymbol{W}^{\star T} = \frac{K}{K-1}\left(\boldsymbol{I}_K - \frac{1}{K}\mathbf{1}_K \mathbf{1}_K^\top\right) \tag{9}$$

*where $\boldsymbol{I}_K \in \mathbb{R}^{K \times K}$ is the identity matrix, $\mathbf{1}_K \in \mathbb{R}^K$ contains only 1, and $b^\star \in \mathbb{R}$ is any constant.*

***Proof:*** *See Theorem 17 in supplementary.*

**Theorem 2.** *Suppose $d \geq K - 1$. The loss $f_{\mathrm{bce}}(\boldsymbol{W}, \boldsymbol{H}, \boldsymbol{b})$ defined using $\mathcal{L}_{\mathrm{bce}}$ in Eq. (5) satisfies*

$$f_{\mathrm{bce}}(\boldsymbol{W}, \boldsymbol{H}, \boldsymbol{b}) \geq -2\gamma - (K-2)b^\star + \log\left(1 + e^{\gamma - b^\star}\right) + (K-1)\log\left(1 + e^{b^\star + \frac{\gamma}{K-1}}\right), \tag{10}$$

*and the equality holds if and only if the minimizer $(\boldsymbol{W}^\star, \boldsymbol{H}^\star, \boldsymbol{b}^\star)$ satisfies Eq. (9) with*

$$b^\star = \log\left((K-2)e^{-\frac{\gamma}{K-1}} + \sqrt{(K-2)^2 e^{-\frac{2\gamma}{K-1}} + 4(K-1)e^{\gamma - \frac{\gamma}{K-1}}}\right) - \log 2. \tag{11}$$

***Proof:*** *See Theorem 10 in supplementary.*

**Unbiased decision scores.** According to Theorems 1 and 2, at the minimum points of normalized CE or BCE, the features $\{h_i^{(k)}\}_{i=1}^n$ of samples from the same class $k$ are equal to each other and coincide with their classifier vector $w_k$, which indicates the maximization of the intra-class compactness (NC1) and the positive unbiased decision scores of all samples converge to $\gamma$, i.e.,

$$s_i^{(kk)} = \gamma \boldsymbol{w}_k^\top \boldsymbol{h}_i^{(k)} \to \gamma, \quad \forall k \in [K], \forall i \in [n]. \tag{12}$$

With Eq. (9), at the minima, the classifier vectors $\{w_k\}_{k=1}^K$ form an equiangular tight frame (ETF), i.e., for $\forall j \neq k$, $\boldsymbol{w}_j^\top \boldsymbol{w}_k = -\frac{1}{K-1}$; meanwhile, the $K$ class centers $\{\bar{\boldsymbol{h}}_k = \frac{1}{n}\sum_{i=1}^n \boldsymbol{h}_i^{(k)}\}_{k=1}^K$ also form an ETF, indicating maximization of inter-class separability (NC2), and all the negative unbiased decision scores converge, i.e.,

$$s_i^{(jk)} = \gamma \boldsymbol{w}_j^\top \boldsymbol{h}_i^{(k)} \to -\frac{\gamma}{K-1}, \quad \forall k \neq j, \forall i. \tag{13}$$

Clearly, as $\gamma$ increases, both the positive and negative decision scores at the minima increase linearly. According to Theorems 1 and 2 and combining NC1 and NC2, one can find that the normalized CE and BCE lead to neural collapse (NC) (Papyan et al., 2020) when they achieve their minima.

**The classifier biases.** When the normalized CE and BCE losses reach their minima, their classifier bias vectors become multiples of the all-one vector $\mathbf{1}_K$. For the normalized CE loss, the multiple factor $b^\star \in \mathbb{R}$ can be any number. As long as $b_k = b_j, \forall k \neq j$, the terms related to the classifier

biases in the Softmax can be canceled out, which indicates that now they are useless in enhancing the compactness and separability of features as they cannot present any substantial constrain to the decision scores. Actually, Yaras et al. (2022) and Lu & Steinerberger (2022a) have demonstrated conclusions similar to Theorem 1, which directly ignore the classifier biases in the CE.

In contrast, for the normalized BCE loss, the multiple factor $b^\star$ can be figured out via Eq. (11), indicating the BCE loss has only one minimum point in terms of the classifier biases. On the contrary, these classifier biases play a substantial role in training models using the BCE loss, necessitating careful consideration of their initialization and other related settings.

## 4.2 CLASSIFICATION AND FEATURE LEARNING WITH DIFFERENT SCALING

As the unbiased decision scores converge to fixed values related to scale factor $\gamma$ when the CE and BCE reach their minima, we can analyze the convergence of the losses by analyzing the convergence behavior of the unbiased decision scores. Meanwhile, when the classifier biases are equal, i.e., $b_k = b_j, \forall k, j$, the unbiased decision scores $\{\gamma \boldsymbol{w}_j^\top \boldsymbol{h}_i^{(k)}\}_{j,k,i}$ not only reflects the feature properties but also determines each sample's classification. Therefore, the convergence behavior of the unbiased decision scores will also help to compare classification and feature learning in the normalized spaces with different scaling. Before these analysis, we first define two critical conditions.

**Definition 3.** *Critical condition I states that for every sample in dataset $\mathcal{D}$, its positive unbiased decision score exceeds the negative ones, i.e.,*

$$\gamma \boldsymbol{w}_k^\top \boldsymbol{h}_i^{(k)} > \max\{\gamma \boldsymbol{w}_j^\top \boldsymbol{h}_i^{(k)} : j \in [K], j \neq k\}, \quad \forall k \in [K], \ i \in [n]. \tag{14}$$

**Definition 4.** *Critical condition II states that for all the samples of dataset $\mathcal{D}$, the positive unbiased decision scores are greater than zero and the negative ones are less than zero, i.e.,*

$$\min \bigcup_{k=1}^{K} \{\gamma \boldsymbol{w}_k^\top \boldsymbol{h}_i^{(k)} : \forall i \in [n]\} > 0 > \max \bigcup_{k=1}^{K} \bigcup_{\substack{j=1 \\ j \neq k}}^{K} \{\gamma \boldsymbol{w}_j^\top \boldsymbol{h}_i^{(k)} : \forall i \in [n]\}. \tag{15}$$

When the classifier biases are equal, critical condition I is equivalent to all the samples being correctly classified, while critical condition II not only implies that the classification accuracy is 100%, but also that their positive and negative unbiased decision scores are uniformly bounded by the threshold $t = 0$, indicating better intra-compactness and inter-class separability of sample features.

**Theorem 5.** *(1) When training the model using the CE loss $f_{ce}$, as $\gamma$ approaches zero, the linear decrease in $\gamma$ leads to a linear decay in the convergence rate of unbiased decision scores.*

*(2) Once the critical condition I is satisfied, as $\gamma$ linearly approaches positive infinity, the convergence rate of unbiased decision scores decay exponentially.*

***Proof:*** *See Theorem 24 in supplementary.*

**Theorem 6.** *(1) When training the model using the BCE loss $f_{bce}$, as $\gamma$ approaches zero, the linear decrease in $\gamma$ leads to a linear decay in the convergence rate of unbiased decision scores.*

*(2) In contrary, once the critical condition II is satisfied, as $\gamma$ linearly approaches positive infinity, the convergence rate of unbiased decision scores decay exponentially.*

***Proof:*** *See Theorem 23 in supplementary.*

Although the scale factor $\gamma$ is typically fixed before the model training, Theorems 5 and 6 reveal the differences between classification and deep feature learning within normalized spaces with varying $\gamma$. As the theorems state, when critical conditions I and II respectively hold during the model training, increasing the scale factor $\gamma$ toward positive infinity causes an exponential slowdown in the convergence of the unbiased decision scores, which will makes it progressively harder for those scores to reach their theoretical extrema and for the model to achieve the optimal feature properties. Despite that, since that critical condition II inherently implies better feature properties than critical condition I, for very large $\gamma$, using the BCE rather than the CE is more likely to yield better intra-class compactness or inter-class separability. In contrast, according to Eqs. (12) and (13), a larger $\gamma$ corresponds a wider gap between the theoretical extrema of positive and negative unbiased decision scores, increasing the likelihood of correctly classifying each sample and achieving

favorable classification for the model. In total, **in a normalized space with a large scale factor $\gamma$, classification would perform well, but deep feature learning performs poorly.**

As the scale factor $\gamma$ decreases and approaches zero, the smaller $\gamma$ leads to a linear decay of convergence for the unbiased decision scores in the training with both CE and BCE, which also makes it not easy for the unbiased decision scores of different samples to reach their theoretical extremes and results in unsatisfactory feature properties. As $\gamma$ approaches zero, the linear decay of convergence for the unbiased decision scores is less than the exponential decay induced when $\gamma$ approaches positive infinity. Therefore, the properties of features learned when $\gamma$ is very small would be superior to that of features learned when $\gamma$ is very large. However, when $\gamma > 0$ is very small, the theoretical gap between the positive and negative unbiased decision scores is also very small, then even slight variance in the decision scores and biases $\{b_j\}_{j=1}^K$ or minor disturbances in the training can significantly reduce the final classification results. In short, **when $\gamma > 0$ is too small, it is not suited to deep feature learning and even less suited to classification**.

Theorems 5 and 6 also imply that when $\gamma$ takes on an appropriate intermediate value on interval of $(0, +\infty)$, the convergence rate of the decision score peaks, at which point the losses converges most rapidly. Our experimental results in Fig. 1 and Table 1 illustrate that with a fixed training strategy, a moderate $\gamma$ can simultaneously optimize classification and feature properties.

## 5 EXPERIMENTS

To validate the conclusions about classification and deep feature learning as well as the differences between the CE and BCE, we trained CNN (He et al., 2016) and Transformer (Dosovitskiy et al., 2020) on MNIST (Lecun et al., 1998), CIFAR10, and CIFAR100 (Krizhevsky et al., 2009).

**Neural collapse (NC) at the minima of losses across models, datasets, and optimizers.** When $\gamma = 8$, with the CE and BCE, we trained ResNet18, ResNet50, DenseNet121, and ViT on the three datasets using SGD and AdamW, respectively, and we employed the metric of $\mathcal{NC}_1, \mathcal{NC}_2$, and $\mathcal{NC}_3$ presented by Zhu et al. (2021) and Liu et al. (2023b) to measure the evolution of NC during the model training. See supplementary C.2 for detail results, which align with our analysis, namely that both the normalized BCE and CE can lead to NC with different models, datasets, and optimizers, when they reach their minima. Furthermore, the BCE converges faster than the CE.

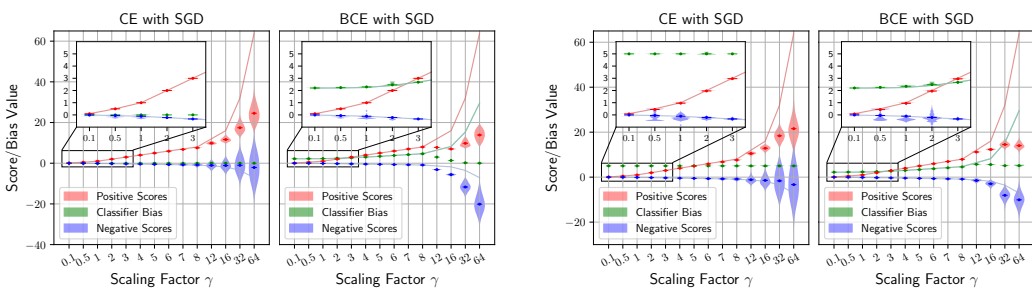

Figure 1: The distribution of the final classifier biases and the positive/negative unbiased decision scores for ResNet18 (left, without the final ReLU activation) and ViT (right) trained on CIFAR10, with various scale factor $\gamma$. The initial mean ($\bar{b} = \frac{1}{K} \sum_{j=1}^K b_j$) of the biases is 0 for ResNet18 and 5 for ViT, and the initial variance is both 0. The solid line represents the theoretical extremum when achieving NC.

**Classification accuracy vs. feature properties with various $\gamma$.** As our analysis in Sec. 4.2, classification and feature learning perform different in the normalized space with different scale factor $\gamma$. To verify their difference, using the CE and BCE with SGD and AdamW, we trained 56 ResNet18s and 56 ViTs on CIFAR10 by setting $\gamma$ to 0.1, 0.5, 1, 2, 3, 4, 5, 6, 7, 8, 12, 16, 32, and 64, respectively. Fig. 1 illustrates that the distribution of their final classifier biases and unbiased decision scores on the training data for SGD. In the figure, the red, blue, and green solid lines represent the theoretical values of the final positive/negative unbiased decision scores and classifier biases for various $\gamma$.

As Fig. 1 shows, when $\gamma \leq 2$ is very small, although the mean of all positive/negative unbiased decision scores eventually aligns with their theoretical extremum values, their variance is quite large,

Table 1: The classification accuracy (%) and feature properties of ResNet18 (without the final ReLU activation) with various $\gamma$ on training data of CIFAR10. The accuracies of $\mathcal{A}$ and $\mathcal{A}^*$ are calculated using the decision scores $\{\gamma \boldsymbol{w}_j^\top \boldsymbol{h}^{(k)} - b_j\}_{j=1}^K$ and the unbiased ones $\{\gamma \boldsymbol{w}_j^\top \boldsymbol{h}^{(k)}\}_{j=1}^K$, respectively. The higher $\mathcal{E}_{\text{com}}$ and $\mathcal{E}_{\text{sep}}$ respectively indicate the better intra-class compactness and inter-class separability.

| | CE with SGD | | | | | | | BCE with SGD | | | | | | |
|---|---|---|---|---|---|---|---|---|---|---|---|---|---|---|
| $\gamma$ | 0.1 | 0.5 | 1 | 12 | 16 | 32 | 64 | 0.1 | 0.5 | 1 | 12 | 16 | 32 | 64 |
| $\mathcal{A}$ | 26.24 | 54.28 | 79.50 | 100.0 | 100.0 | 100.0 | 100.0 | 25.47 | 52.07 | 76.73 | 100.0 | 100.0 | 100.0 | 100.0 |
| $\mathcal{A}^*$ | 27.27 | 57.00 | 80.01 | 100.0 | 100.0 | 100.0 | 100.0 | 27.77 | 54.75 | 77.76 | 100.0 | 100.0 | 100.0 | 100.0 |
| $\mathcal{E}_{\text{com}}$ | 0.965 | 0.995 | 0.998 | 0.905 | 0.862 | 0.789 | 0.766 | 0.965 | 0.995 | 0.997 | 0.988 | 0.973 | 0.899 | 0.842 |
| $\mathcal{E}_{\text{sep}}$ | 0.526 | 0.554 | 0.555 | 0.454 | 0.394 | 0.288 | 0.221 | 0.526 | 0.554 | 0.555 | 0.459 | 0.410 | 0.354 | 0.313 |

particularly for the negative unbiased scores of ViT in the right side, which indicates that at this point, the losses has not fully converged to their minima. When $3 \leq \gamma \leq 8$, all the final unbiased decision scores fall along their theoretical extreme values on the corresponding real lines, as well as the final classifier biases for the BCE, which indicates that the losses has fully converged at this stage. As $\gamma$ continues to increase to 12 or more, the positive/negative unbiased decision scores of the trained model gradually deviate from their theoretical extremum, and they are distributed over an increasingly larger areas, particularly for models trained with the CE, which shows an even broader distribution, which indicates that the losses have totally not converged.

Since the substantial role of classifier biases in the BCE during the model training, when $\gamma \leq 8$, they eventually converges to their theoretical extremum. In contrast, the final biases of the CE are almost entirely dependent on their initial value, i.e. 0 for ResNet18 and 5 for ViT in the figure.

Table 1 presents the classification accuracy and feature properties of ResNet18 trained with small and large $\gamma$, and the results are calculated on the training data of CIFAR10. The expressions of $\mathcal{A}$, $\mathcal{A}^*$, $\mathcal{E}_{\text{com}}$, and $\mathcal{E}_{\text{sep}}$ are presented in supplementary (Sec. B.2). When $\gamma = 0.1$, the accuracies $\mathcal{A}$ of the models are very low, while the unbiased accuracies $\mathcal{A}^*$ are relatively high, indicating the slight variance in the classifier biases significantly affect the classification. However, with $\gamma = 0.1$ or 0.5, the final feature properties are not so bad, as the intra-class compactness $\mathcal{E}_{\text{com}}$ and inter-class separability $\mathcal{E}_{\text{sep}}$ are high, although they have not yet reached their maximum. In contrast, when $\gamma = 32$ or 64, the models' accuracies reach 100%, while the intra-class compactness and inter-class separability of their features are comparatively poor and worse than that learned with small $\gamma$.

Combining Fig. 1 and Table 1, one can find that when $\gamma = 32$ or 64, although the CE and BCE after training do not converge, the models trained by them satisfy critical condition I and II, respectively.

The applications of the BCE in the face recognition have been explored by Wen et al. (2022) and Zhou et al. (2023), and we here explore its advantages over the CE on other more challenging tasks such as **long-tailed recognition (LTR)** and **open-set recognition (OSR)**. Both set $\gamma = 32$.

**LTR.** On the imbalanced datasets, CIFAT10-LT and CIFAR100-LT, when the imbalance factors of the training sets are 10, 50, and 100, we trained ResNet32 using the normalized CE and BCE with SGD, respectively. Table 2 shows the classification results on the balanced test set of CIFAT10 and CIFAR100. The BCE consistently achieves better LTR results on the six pairs of models. We believe that, compared to the CE, which couples the $K$ decision scores of each sample into one Softmax, the BCE decouples them using $K$ Sigmoids, mitigating the imbalance effects caused by the imbalanced datasets and improving the LTR performance.

Table 2: The classification (%) on the test sets of CIFAR10-LT and CIFAR100-LT.

| $\mathcal{D}$ | Loss | 10 | 50 | 100 |
|---|---|---|---|---|
| CIFAR10 | CE | 93.68 | 87.80 | 83.37 |
| | BCE | **93.96** | **88.75** | **84.47** |
| CIFAR100 | CE | 69.30 | 55.15 | 49.47 |
| | BCE | **69.49** | **58.53** | **52.15** |

**OSR.** In the open-set experiments, we evaluate the performances of the CE and BCE on the MNIST, SVHN, CIFAR10 and CIFAR+50 dataset configurations, and the model and training details follow APRL (Chen et al., 2021). For the CE and BCE, Table 3 presents the OSR results in terms of the area under the receiver operating characteristic

Table 3: The OSR results under various setups.

| | MNIST | | SVHN | | CIFAR10 | | CIFAR+50 | |
|---|---|---|---|---|---|---|---|---|
| | CE | BCE | CE | BCE | CE | BCE | CE | BCE |
| AUROC | 98.6 | **99.2** | 94.3 | **95.1** | 84.2 | **85.8** | 87.8 | **90.2** |
| OSCR | 98.5 | **99.0** | 92.4 | **93.4** | 81.9 | **83.7** | 85.7 | **88.5** |

(AUROC) and the open set classification rate (OSCR). On the various OSR experiments, the BCE consistently achieves superior performance across the two metrics. We believe that in the BCE, the uniform and explicit constraints of the classifier biases on the positive/negative decision scores are beneficial for learning a clear decision boundary for sample features, which in turn aids the model in discriminating unknown classes and enhances the performance in the OSR.

## 6 DISCUSSION, LIMITATION, AND FUTURE WORK

**Weight decay factor**.   In Euclidean space without normalization on sample features or classifier vectors, $L_2$ regularization with weight decay factor $\lambda$ are typically added in CE and BCE, which constrain the features within a bounded space. Intuitively, a small $\lambda$ results in a large feature space, while a large $\lambda$ leads to a small one. Li et al. (2025) have compared the NC of CE and BCE in Euclidean space but do not reveal the difference of classification and feature learning. We conjecture that, similar to the scale factor $\gamma$ in normalized space, when $\lambda \geq 0$ is very small, it is difficult for the losses to reach their minima, resulting in poor feature properties; when $\lambda$ is very large, the small theoretical gap between the positive and negative decision scores will harm the classification. However, in Euclidean space, it is not easy to rigorously analyze convergence behavior of the losses.

**Temperature coefficient**.   In contrastive learning (CL), a temperature $\tau$ is applied on the denominator of the feature cosine similarity in normalized space. In CL, the Softmax-based losses are typically applied, similar to the CE loss in classification. When the CL losses reach their minima, the scaled feature similarity $\frac{1}{\tau}\langle \boldsymbol{h}, \boldsymbol{h}_* \rangle$ of any positive or negative sample pair converge to fixed values, according to NC studies by Graf et al. (2021) and Koromilas et al. (2024). We here conjecture that the impact of the temperature $\tau$ on the convergence rate of Softmax-based losses parallels that of the scale factor $\gamma$ on the convergence rate of CE loss in classification, i.e., too large or too small $\tau$ would result in slow convergence of the CL losses and thereby poor feature properties.

**BCE for deep feature learning**.   In Sec. 3.2, with the global constrains of deep feature learning, we reformulated the BCE loss in the multi-class setting. However, according to Eq. (3), when the threshold $t_j$ measures the intra-class compactness of the $j$-th class, a larger value is preferable; whereas according to Eq. (4), when $t_j$ measures the inter-class separability between the class $k$ and class $j$, a smaller value is preferable. This tension implies that, during the BCE-based training, although the classifier biases $\{b_j\}_{j=1}^K$ impose substantial constraints on the learning of decision scores that reflect the feature properties, these constraints do not always favor strengthening the feature properties. It deserves to further explore a loss that totally matches the deep feature learning.

**Minima of losses in more challenging scenarios**.   In more challenging scenarios, such as low-dimensional spaces where the feature dimension is smaller than the number of classes (i.e., $d < K$) and in class-imbalanced settings, theoretically analyzing the minima and convergence of the losses becomes more difficult. Currently, we are investigating the minima of the BCE loss under class imbalance, and we believe that the scale factor $\gamma$ has similar effects on its convergence.

**Deep feature learning in other scenarios.**   In this paper, within the normalized, supervised, closed-set, and low-dimensional feature spaces, we analyze the convergence behavior of CE and BCE losses, to compare the tasks of classification and deep feature learning. When this setting changes, the convergence behavior of the losses may also change, thereby altering the conclusions regarding the two tasks. However, we conjecture that in both supervised and self-supervised contrastive learning, as scale factor $\gamma$ linearly increases, the convergence rate of the existing Softmax-based losses decays exponentially, which implies that larger feature spaces are less favorable for the contrastive learning.

## 7 CONCLUSIONS

We conduct an in-depth comparison for classification and deep feature learning in normalized space, by theoretically analyzing the minima and convergence of CE and BCE losses. We point out that classification accuracy reflects the models' sample-wise local properties on a dataset, while the intra-class compactness and inter-class separability of features represent the sample-independent global properties on the dataset. As the scale factor $\gamma$ changes, classification and feature learning perform differently in normalized spaces with varying sizes. As the BCE could obtain better features, it outperforms the CE in more challenging tasks such as long-tailed and open-set recognitions.

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

# Supplementary

## Contents

# A  THE USAGE OF LLMs

In this work, we only utilized LLMs to assist us in polishing the English texts. The LLMs primarily helped us check English grammar and perform complex bilingual translations.

The original conception of this work, experimental design, and drafting of the paper's text were all completed by the authors, with no involvement from the LLMs in these steps.

# B  NEURAL COLLAPSE AND FEATURE PROPERTIES

## B.1  NEURAL COLLAPSE

Neural collapse (NC) was first found by Papyan et al. (2020) that occurs in the terminal training phase of the classification model, manifesting as elegant geometric structures in the sample features and classifier vectors. With neural collapse, the geometric structure of sample features and classifier vectors manifests as follows:

- For any class $k$, all the sample features $\left\{\boldsymbol{h}_i^{(k)}\right\}_{i=1}^{n_k}$ of this class converge to the feature center $\bar{\boldsymbol{h}}_k = \frac{1}{n_k} \sum_{i=1}^{n_k} \boldsymbol{h}_i^{(k)}$, i.e., $\boldsymbol{h}_i^{(k)} = \boldsymbol{h}_j^{(k)}$, for $\forall i,j \in [n_k] = \{1, 2, \cdots, n_k\}$;

- The feature center vectors of any two classes have the same angular distance and equal magnitudes, i.e., $\|\bar{\boldsymbol{h}}_k\| = \|\bar{\boldsymbol{h}}_\ell\|$ and $\frac{\langle \bar{\boldsymbol{h}}_k, \bar{\boldsymbol{h}}_\ell \rangle}{\|\bar{\boldsymbol{h}}_k\| \|\bar{\boldsymbol{h}}_\ell\|} = \frac{\langle \bar{\boldsymbol{h}}_{k'}, \bar{\boldsymbol{h}}_{\ell'} \rangle}{\|\bar{\boldsymbol{h}}_{k'}\| \|\bar{\boldsymbol{h}}_{\ell'}\|}$, for $\forall k \neq \ell, k' \neq \ell' \in [K]$, forming an equiangular tight frame (ETF);

- The feature center vector of any class is parallel to its corresponding classifier vector with a fixed constant between them, i.e., $\exists \alpha > 0$, such that $\bar{\boldsymbol{h}}_k = \alpha \boldsymbol{w}_k$, for $\forall k \in [K]$.

In Zhu et al. (2021); Zhou et al. (2022), the authors defined metrics of $\mathcal{NC}_1, \mathcal{NC}_2, \mathcal{NC}_3$ to evaluate the above properties. In this paper, we take them to compare the evolution of NC of the CE and BCE in the normalized feature space.

## B.2  CLASSIFICATION

In our experiments, we apply four metrics to comprehensively compare the performance of the BCE and CE losses in the normalized feature space, i.e., classification accuracy $\mathcal{A}$, unbiased classification accuracy $\mathcal{A}^*$, features compactness $\mathcal{E}_{\text{com}}$, and features separability $\mathcal{E}_{\text{sep}}$. These metrics will be maximized when reaching the neural collapse.

For a classification task, suppose a dataset

$$\mathcal{D} = \bigcup_{k=1}^{K} \mathcal{D}_k = \bigcup_{k=1}^{K} \left\{ \boldsymbol{X}_i^{(k)} \right\}_{i=1}^{n_k} \tag{16}$$

from $K$ categories, where $\boldsymbol{X}_i^{(k)}$ denotes the $i$th sample from the class $k$. For any sample $\boldsymbol{X}_i^{(k)}$, a model $\mathcal{M}$ converts it into its feature $\boldsymbol{h}_i^{(k)} = \mathcal{M}(\boldsymbol{X}_i^{(k)}) \in \mathbb{R}^d$, where $d$ is dimension of the feature space. To classify the sample, a linear, full connection classifier $\mathcal{C} = \left\{ (\boldsymbol{w}_k, b_k) \right\}_{k=1}^{K}$ transform the feature into $K$ decision scores $\left\{ \gamma \boldsymbol{w}_j^\top \boldsymbol{h}_i^{(k)} - b_j \right\}_{j=1}^{K}$, where $\{\boldsymbol{w}_k\}_{k=1}^{K}$ are classifier vectors and $\{b_k\}_{k=1}^{K}$ are classifier biases. Then, the sample is classified into class $\hat{k}$,

$$\hat{k} = \arg \max_j \left\{ \gamma \boldsymbol{w}_j^\top \boldsymbol{h}_i^{(k)} - b_j \right\}_{j=1}^{K}, \tag{17}$$

and correct classification for the sample is achieved when $k = \hat{k}$, which is equivalent to

$$\gamma \boldsymbol{w}_k^\top \boldsymbol{h}_i^{(k)} - b_k = \max \left\{ \gamma \boldsymbol{w}_j^\top \boldsymbol{h}_i^{(k)} - b_j \right\}_{j=1}^{K}. \tag{18}$$

**Classification accuracy.**  The commonly used classification accuracy $\mathcal{A}$ is defined as

$$\mathcal{A}(\mathcal{M}, \mathcal{C}) = \frac{|\mathcal{D}(\mathcal{M}, \mathcal{C})|}{|\mathcal{D}|} \times 100\%, \tag{19}$$

where

$$\mathcal{D}(\mathcal{M},\mathcal{C}) = \bigcup_{k=1}^{K} \left\{ \boldsymbol{X}^{(k)} : k = \arg\max_{\ell}\{\gamma\boldsymbol{w}_{\ell}^{\top}\boldsymbol{h}^{(k)} - b_{\ell}\}, \boldsymbol{X}^{(k)} \in \mathcal{D}_k, \boldsymbol{h}^{(k)} = \mathcal{M}(\boldsymbol{X}^{(k)}) \right\}, \quad (20)$$

consisting of all the samples correctly classified by $\mathcal{M}$ and $\mathcal{C}$ in $\mathcal{D}$.

**Unbiased classification accuracy.** Similarly, the unbiased classification accuracy $\mathcal{A}^*$ is computed without using the classifier biases $\{b_k\}_{k=1}^{K}$, i.e.,

$$\mathcal{A}^*(\mathcal{M},\mathcal{C}) = \frac{|\mathcal{D}^*(\mathcal{M},\mathcal{C})|}{|\mathcal{D}|} \times 100\%, \quad (21)$$

where

$$\mathcal{D}^*(\mathcal{M},\mathcal{C}) = \bigcup_{k=1}^{K} \left\{ \boldsymbol{X}^{(k)} : k = \arg\max_{\ell}\{\gamma\boldsymbol{w}_{\ell}^{\top}\boldsymbol{h}^{(k)}\}, \boldsymbol{X}^{(k)} \in \mathcal{D}_k, \boldsymbol{h}^{(k)} = \mathcal{M}(\boldsymbol{X}^{(k)}) \right\}. \quad (22)$$

The difference between the classification accuracy $\mathcal{A}$ and unbiased one $\mathcal{A}^*$ reflects the affect of the variance of the biases $\{b_j\}_{j=1}^{K}$ to the classification.

### B.3 FEATURE PROPERTIES

There is a close and intricate relationship between the sample classification and their feature properties. To precisely measure the properties of sample features, we define their intra-class compactness $\mathcal{E}_{\text{com}}$ and inter-class separability $\mathcal{E}_{\text{sep}}$,

$$\mathcal{E}_{\text{com}} = \frac{1}{2}\left[\frac{1}{K}\sum_{k=1}^{K}\left(\frac{1}{n_k^2}\sum_{i=1}^{n_k}\sum_{i'=1}^{n_k}\frac{\langle\boldsymbol{h}_i^{(k)} - \bar{\boldsymbol{h}}, \boldsymbol{h}_{i'}^{(k)} - \bar{\boldsymbol{h}}\rangle}{\|\boldsymbol{h}_i^{(k)} - \bar{\boldsymbol{h}}\|\|\boldsymbol{h}_{i'}^{(k)} - \bar{\boldsymbol{h}}\|}\right) + 1\right], \quad (23)$$

$$\mathcal{E}_{\text{sep}} = \frac{1}{2}\left[1 - \frac{1}{K(K-1)}\sum_{k=1}^{K}\sum_{\substack{k'=1\\k'\neq k}}^{K}\left(\frac{1}{n_k}\frac{1}{n_{k'}}\sum_{i=1}^{n_k}\sum_{i'=1}^{n_{k'}}\frac{\langle\boldsymbol{h}_i^{(k)}, \boldsymbol{h}_{i'}^{(k')}\rangle}{\|\boldsymbol{h}_i^{(k)}\|\|\boldsymbol{h}_{i'}^{(k')}\|}\right)\right], \quad (24)$$

where $\bar{\boldsymbol{h}} = \frac{1}{|\mathcal{D}|}\sum_{k=1}^{K}\sum_{i=1}^{n_k}\boldsymbol{h}_i^{(k)}$ is the global feature center.

Due to the properties of neural collapse, when approaching the minima of the CE or BCE, the intra-class compactness $\mathcal{E}_{\text{com}}$ might be higher than $\frac{1}{2} - \frac{1}{2(K-1)}$, and the inter-class separability $\mathcal{E}_{\text{sep}}$ might be lower than $\frac{1}{2} + \frac{1}{2(K-1)}$, for the model $\mathcal{M}$ and classifier $\mathcal{C}$ trained on the dataset $\mathcal{D}$.

Table 4: Detailed experimental settings.

| | | Neural Collapse | | | | Classification | |
|---|---|---|---|---|---|---|---|
| | | setting-1 | setting-2 | setting-3 | setting-4 | setting-5 | setting-6 |
| Hyper-parameter | epochs | 200 | 200 | 200 | 200 | 200 | 200 |
| | optimizer | SGD | AdamW | SGD | AdamW | SGD | AdamW |
| | batch size | 128 | 128 | 128 | 128 | 128 | 128 |
| | learning rate | 0.01 | 0.001 | 0.03 | 0.0003 | 0.1 | 0.005 |
| | learning rate decay | step | cosine | step | cosine | step | cosine |
| | weight decay $\lambda$ | ✗ | ✗ | ✗ | ✗ | 0.0001 | 0.05 |
| | warmup epochs | 0 | 0 | 0 | 0 | 0 | 0 |
| Data Aug. | random cropping | ✗ | ✗ | ✗ | ✗ | ✓ | ✓ |
| | horizontal flipping | ✗ | ✗ | ✗ | ✗ | 0.5 | 0.5 |
| | random rotation | ✗ | ✗ | ✗ | ✗ | 15 | 15 |
| | label smoothing | ✗ | ✗ | ✗ | ✗ | 0.1 | 0.1 |
| | mixup alpha | ✗ | ✗ | ✗ | ✗ | 0.8 | 0.8 |
| | cutmix alpha | ✗ | ✗ | ✗ | ✗ | 1.0 | 1.0 |
| | mixup prob. | ✗ | ✗ | ✗ | ✗ | 0.8 | 0.8 |
| | normalization | mean = $[0.4914, 0.4822, 0.4465]$, std = $[0.2023, 0.1994, 0.2010]$ | | | | | |

# C EXPERIMENTS

## C.1 EXPERIMENTAL SETTING DETAILS

In Sec. 5, we conducted extensive experiments with multiple models across various datasets to validate our theoretical findings and analyses. We present additional experimental details in Table 4.

**Neural collapse on balanced classification**. By default, to validate the phenomenon of neural collapse and the convergence of models, we train ResNet18, ResNet50 (He et al., 2016), and DenseNet121 on the MNIST (Lecun et al., 1998), CIFAR10, and CIFAR100 (Krizhevsky et al., 2009) by using settings-1 and 2, and we train ViT (Dosovitskiy et al., 2020) from scratch on the CIFAR10 dataset using settings-3 and 4, without applying additional regularization or data augmentation techniques during the training.

To assess classification performance on balanced datasets, we train ResNet18 and ResNet50 on the CIFAR10, CIFAR100, and Tiny-ImageNet (Wu et al., 2017) datasets using settings-5 and 6, employing the commonly used data augmentation techniques for a fair comparison. When training ViT models on the CIFAR10 and CIFAR100 datasets with the AdamW optimizer, we follow the weight decay setting of Chhabra (2024), and conduct a learning rate search based on setting-5 and 6. For training ViT on the Tiny-ImageNet dataset, we follow the network architecture and hyperparameter settings of Lee et al. (2021). The classification performance of the models trained using CE and BCE losses are shown in Table 9.

**Long-tailed recognition**. The long-tailed datasets, CIFAR10-LT and CIFAR100-LT , are produced by sampling the training samples of the original datasets, using an exponential decay imbalance mode across classes, following the work of Cao et al. (2019). For each of them, we produced three variants of long-tailed datasets by using three different imbalance factors (IF), $10, 50,$ and $100$. The IF is defined as the number of training samples in the largest class divided by the smallest. The experimental setting details for long-tailed recognition can be found in the work of Alshammari et al. (2022).

**Open-set recognition**. We sample from original datasets to generate the corresponding known and unknown classes, a simple summary is provided: (1) For MNIST, SVHN (Netzer et al., 2011) and CIFAR10, six known classes and four unknown classes are randomly sampled. (2) For the CIFAR+50 experiments, four classes are sampled from CIFAR10 for training and 50 nonoverlapping classes are used as unknown classes, which are sampled from the CIFAR100 dataset. We use the area under the receiver operating characteristic (AUROC) curve (Neal et al., 2018) and open set classification Rate (OSCR) (Chen et al., 2021) as evaluation metrics in the experiments. The experimental setting details for open-set recognition can be found in the work of Chen et al. (2021).

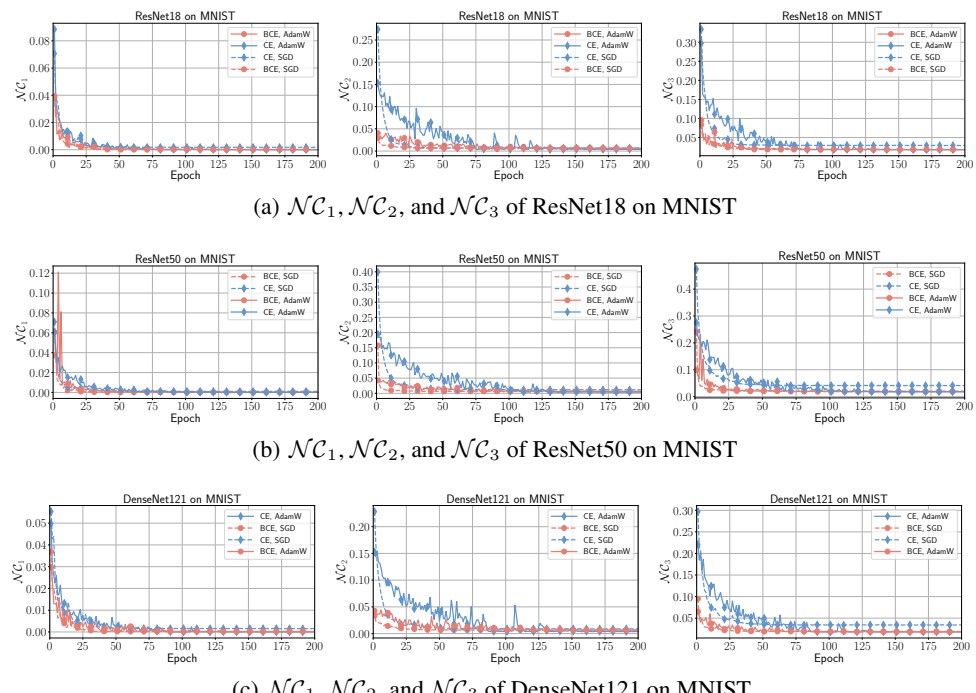

(a) $\mathcal{NC}_1, \mathcal{NC}_2$, and $\mathcal{NC}_3$ of ResNet18 on MNIST

(b) $\mathcal{NC}_1, \mathcal{NC}_2$, and $\mathcal{NC}_3$ of ResNet50 on MNIST

(c) $\mathcal{NC}_1, \mathcal{NC}_2$, and $\mathcal{NC}_3$ of DenseNet121 on MNIST

Figure 2: The evolution of the three NC metrics in the training of ResNet18 (top), ResNet50 (middle) and DenseNet121 (bottom) on MNIST with CE and BCE losses using SGD and AdamW, respectively.

## C.2 NEURAL COLLAPSE OF BCE AND CE LOSSES

This section presents more results for the neural collapse (NC) that support the conclusions drawn in the paper. These results are computed on the training data from the the datasets. As we have mentioned before, we take $\mathcal{NC}_1, \mathcal{NC}_2$, and $\mathcal{NC}_3$ presented in Zhu et al. (2021) and Zhou et al. (2022) to evaluate the evolution of NC for the CE and BCE.

When $\gamma = 8$, Figs. 2 - 5 shows the evolution in the training of ResNet18, ResNet50, DenseNet121, and ViT on the training datasets of MNIST, CIFAR10, and CIFAR100 with CE and BCE losses. In the training on MNIST and CIFAR10 dataset, the three NC metrics of both CE and BCE losses approach zero at the terminal phase of training, indicating that both have approached the neural collapse. Additionally, it is obvious that BCE decreases faster and converges to a higher degree than CE in the first 50 epochs. However, during the training on CIFAR100, which is a more challenging dataset than MNIST and CIFAR10, the NC metrics of models trained with the SGD optimizer, especially $\mathcal{NC}_2$ and $\mathcal{NC}_3$, do not approach zero, whereas those of models trained with AdamW approach zero. We can still observe that in most cases, the NC metrics of the BCE loss decreases faster and converges better than the CE loss.

## C.3 IMPACT OF CLASSIFIER BIASES AND SCALE FACTOR

**The mean of initial classifier biases $b$.** In Sec. 4.1, we point out that classifier biases differs substantially between the BCE and CE losses; specifically, at the global minimizer, the bias under the BCE loss is uniquely determined, whereas under the CE loss there are infinitely many solutions. To validate this finding, we present the distribution of the final classifier biases and positive/negative unbiased decision scores for ResNet18 trained on MNIST, and ViT trained from scratch on CIFAR10 using different optimizer, with varying mean of initial classifier biases.

Figs. 6 and 7 show that the final classifier biases of CE-trained models are determined by their initial values, while the biases of BCE-trained models always converge to the same value, consistent with our theoretical analysis in the paper.

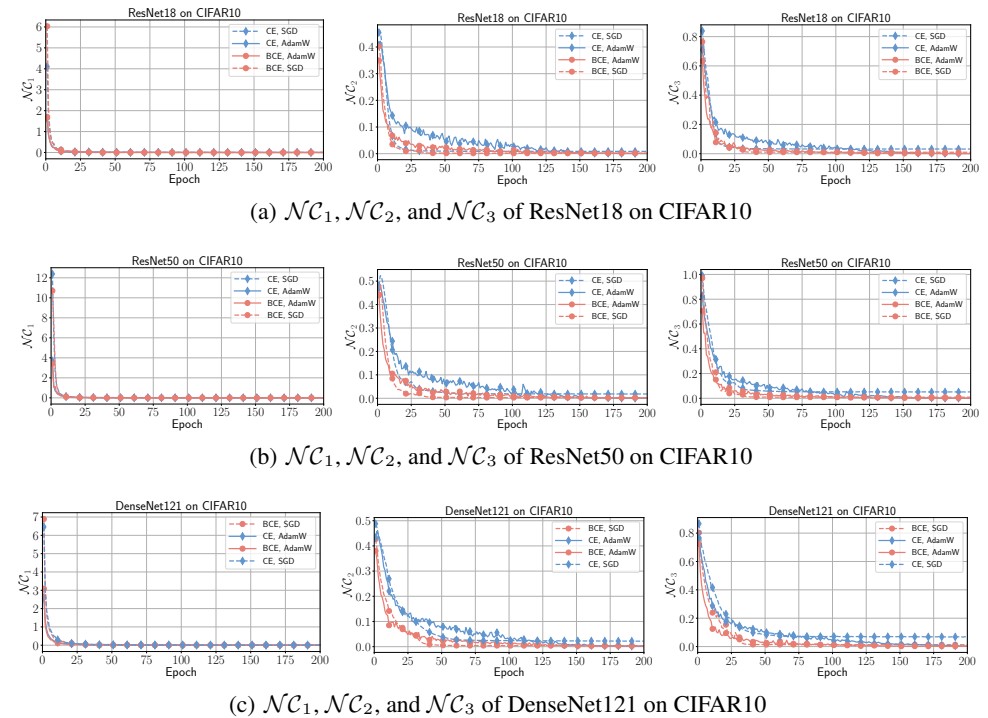

(a) $\mathcal{NC}_1, \mathcal{NC}_2$, and $\mathcal{NC}_3$ of ResNet18 on CIFAR10

(b) $\mathcal{NC}_1, \mathcal{NC}_2$, and $\mathcal{NC}_3$ of ResNet50 on CIFAR10

(c) $\mathcal{NC}_1, \mathcal{NC}_2$, and $\mathcal{NC}_3$ of DenseNet121 on CIFAR10

Figure 3: The evolution of the three NC metrics in the training of ResNet18 (top), ResNet50 (middle) and DenseNet121 (bottom) on CIFAR10 with CE and BCE losses using SGD and AdamW, respectively.

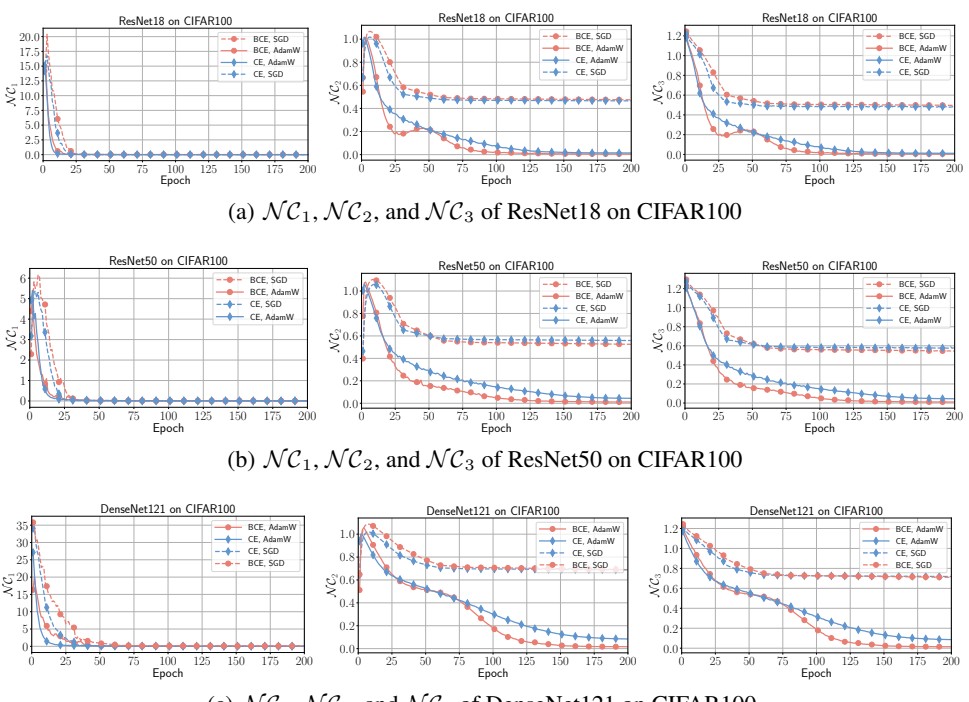

(a) $\mathcal{NC}_1, \mathcal{NC}_2$, and $\mathcal{NC}_3$ of ResNet18 on CIFAR100

(b) $\mathcal{NC}_1, \mathcal{NC}_2$, and $\mathcal{NC}_3$ of ResNet50 on CIFAR100

(c) $\mathcal{NC}_1, \mathcal{NC}_2$, and $\mathcal{NC}_3$ of DenseNet121 on CIFAR100

Figure 4: The evolution of the three NC metrics in the training of ResNet18 (top), ResNet50 (middle) and DenseNet121 (bottom) on CIFAR100 with CE and BCE losses using SGD and AdamW, respectively.

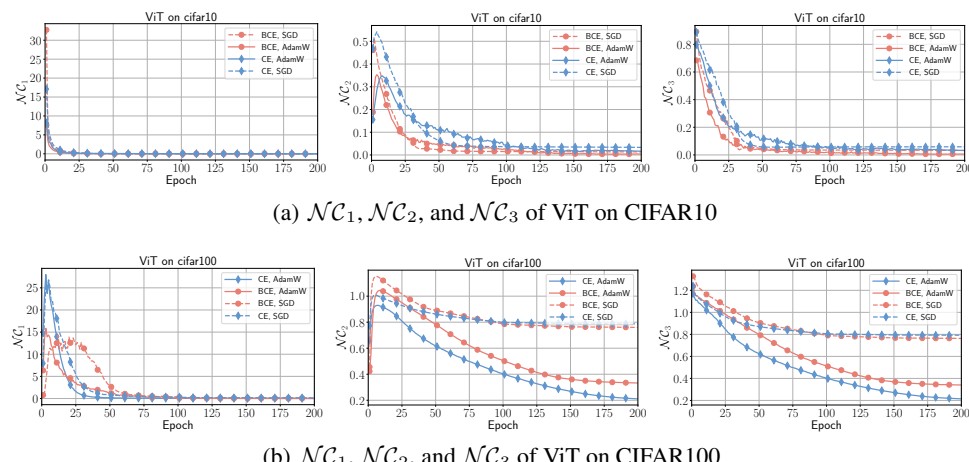

(a) $\mathcal{NC}_1$, $\mathcal{NC}_2$, and $\mathcal{NC}_3$ of ViT on CIFAR10

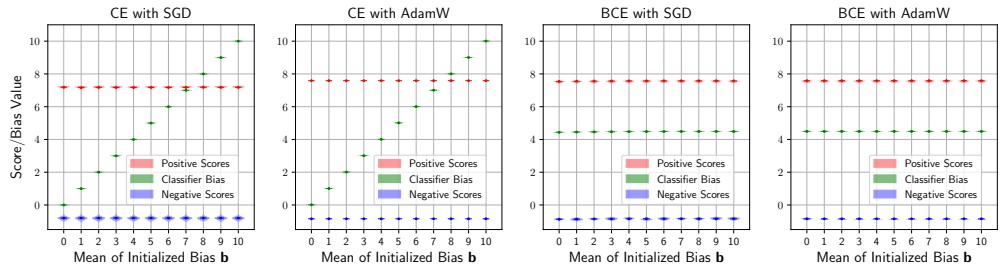

(b) $\mathcal{NC}_1$, $\mathcal{NC}_2$, and $\mathcal{NC}_3$ of ViT on CIFAR100

Figure 5: The evolution of the three NC metrics in the training of ViT on CIFAR10 (top) and CIFAR100 (bottom) with CE and BCE losses using SGD and AdamW, respectively.

Figure 6: The distribution of the final classifier biases and positive/negative unbiased decision scores for ResNet18 trained on MNIST with varying mean of initial classifier biases, while the variance of initial classifier biases is $0$ and $\gamma = 8$. The final classifier biases of CE-trained models are determined by their initial values, while that of BCE-trained models always converge to the same value.

It is notable to point out a difference between CNNs and Transformers. For the CNN models such as ResNet and DenseNet, the output sample features are processed by the ReLU activation function, which makes that each component of the generated features is a nonnegative number and results in the sum of all sample features not being equal to zero (unless all the features are zero vectors, which is meaningless), conflicting with Eqs. (85) and (175) required for proving Theorems. Therefore, the perfect results for NC cannot be achieved based on the ResNet and DenseNet, as demonstrated by the obvious difference between the converged positive unbiased decision scores and the theoretical value $\gamma = 8$, regardless of whether the model is trained using CE or BCE. This phenomenon has been found and discussed by Zhu et al. (2021). In contrast, the features output by ViT are not processed by ReLU or any other activation functions, which aligns more closely with the unconstrained feature model (Zhu et al., 2021; Yaras et al., 2022). When training ViT with the CE or BCE, the converged positive and negative unbiased decision scores can converge to $\gamma = 8$ and $-\frac{\gamma}{K-1} = -\frac{8}{9}$, respectively.

**The variance of initial classifier biases $b$.** It is well known that the variance of the parameters at model initialization determines whether gradient signals propagate stably in deep neural networks, thereby affecting convergence speed, training stability, and final generalization performance. Based on this consensus, to more comprehensively validate our theoretical findings, we conduct experiments with a fixed mean of initial classifier biases of $0$, when the variances of initial classifier biases are set to $0, 0.1, 0.5, 1, 2, 3, 4, 5, 6$, respectively, to further compare the BCE and CE losses.

Figs. 8 and 9 visually show the distributions of the final unbiased decision scores and classifier biases for ResNet18 trained on MNIST and that for ViT trained from scratch on CIFAR10. For the different network architectures, optimizers, and scale factors, the BCE generally achieves better convergence

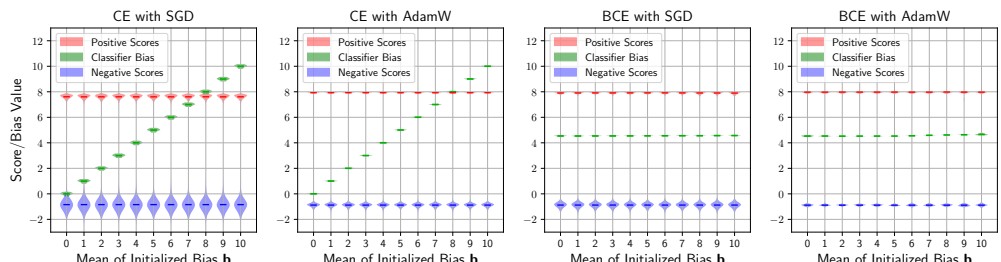

Figure 7: The distribution of the final classifier biases and positive/negative unbiased decision scores for ViT trained on CIFAR10 with varying mean of initial classifier biases, while the variance of initial classifier biases is $0$ and $\gamma = 8$. The final classifier biases of CE-trained models are determined by their initial values, while that of BCE-trained models always converge to the same value.

compared to the CE, especially when $\gamma = 8$. When $\gamma = 16, 32$ or $64$, the large normalized feature space makes it difficult for the models trained with either CE or BCE to converge, but the BCE-trained models usually yield better convergence.

### C.4  IMPACT OF SCALE FACTOR TO MODEL TRAINING

**The scale factor $\gamma$.**   To illustrate the geometric effect of the scale factor for model training in the normalized feature space, we trained ResNet18 and ViT from scratch on the MNIST and CIFAR10 datasets when $\gamma$ varies from $0.1$ to $64$.

Figs. 10 - 13 visually show the distributions of the final decision scores and classifier biases for ResNet18 and ViT trained from scratch on the MNIST and CIFAR10 datasets. One can observe that, when $\gamma \leq 8$ is small, the positive and negative unbiased decision scores, as well as the final classifier biases, of the models trained with BCE and CE losses closely align with the theoretical curves. However, as $\gamma$ increases, they increasingly deviate from their theoretical values. Specifically, when $\gamma$ is reaching $32$ or even $64$, the distributions of the model's positive/negative unbiased decision scores span large ranges and do not converge.

Furthermore, one can find that, in the CE-trained models, the converged classifier biases largely depend on their initial values, providing no correlation with the positive/negative unbiased decision scores. In contrast, with appropriately small $\gamma$, regardless of the initial value of the biases, the final classifier biases of the BCE-trained models converge to the theoretical values.

To further demonstrate the impact of different scale factor $\gamma$ on the tasks of classification and deep feature learning, we recorded the training epochs at which the models' classification accuracy $\mathcal{A}$, inter-class compactness $\mathcal{E}_{\text{com}}$, and inter-class separability $\mathcal{E}_{\text{sep}}$ reach their respective extrema for ResNet18 and ViT trained on CIFAR10 in Figs. 14 and 15. According to the analysis in our paper, a larger scale factor $\gamma$ is beneficial for classification task but detrimental to feature learning; thus, when $\gamma$ is too large, the classification accuracy quickly reaches $100\%$, while the number of training epochs required to achieve the extrema of feature properties increases with $\gamma$ increasing, until it cannot reach their extrema within 200 epochs. Conversely, the losses with a very small $\gamma$ perform poorly in both classification and feature learning; however, the feature properties at this time are often superior to those when $\gamma$ is very large. When the scale factor $\gamma$ is moderate, both the model's classification performance and feature properties can reach their extrema within 200 epochs.

### C.5  NUMERICAL RESULTS OF RESNET18 AND VIT TRAINED ON CIFAR10

In this section, Tables 5 and 6 provide the classification results and feature properties of ResNet18 and ViT trained from scratch on the MNIST and CIFAR10 datasets, when the scale factor $\gamma$ varies from $0.1$ to $64$.

It is evident that when $\gamma$ takes a very small value, it can significantly affect the model's classification performance, as the distance between the positive and negative decision scores becomes small. In this situation, even slight disturbance in the model training or small variance in the classifier biases can

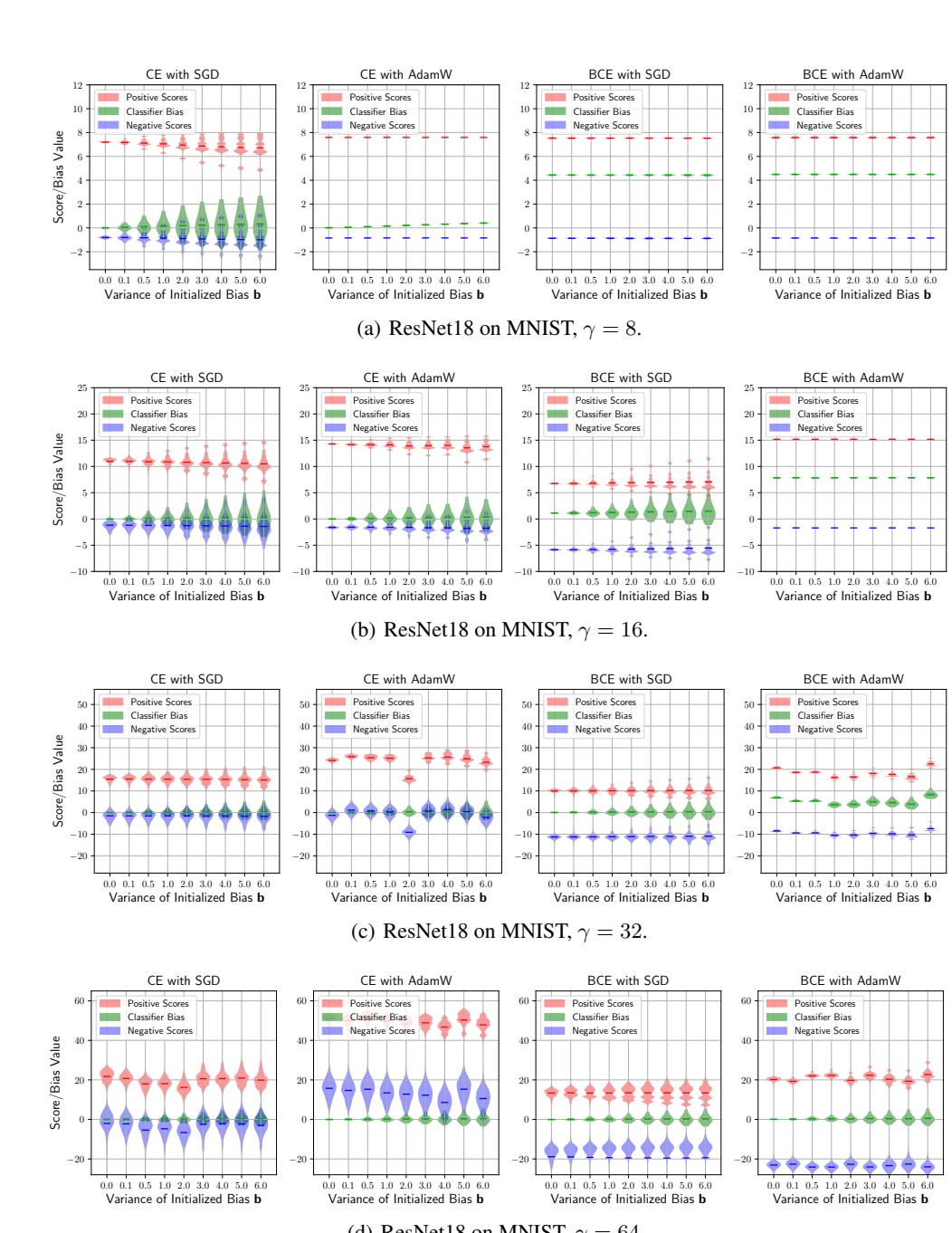

(a) ResNet18 on MNIST, $\gamma = 8$.

(b) ResNet18 on MNIST, $\gamma = 16$.

(c) ResNet18 on MNIST, $\gamma = 32$.

(d) ResNet18 on MNIST, $\gamma = 64$.

Figure 8: The distribution of the final classifier biases and positive/negative unbiased decision scores of ResNet18 trained on MNIST with varying initial variance of the classifier biases. The different scale factors, $\gamma = 8$ (1-st row), $\gamma = 16$ (2-nd row), $\gamma = 32$ (3-rd row), and $\gamma = 64$ (4-th row), also significantly impact the convergence of decision scores and the final classifier biases.

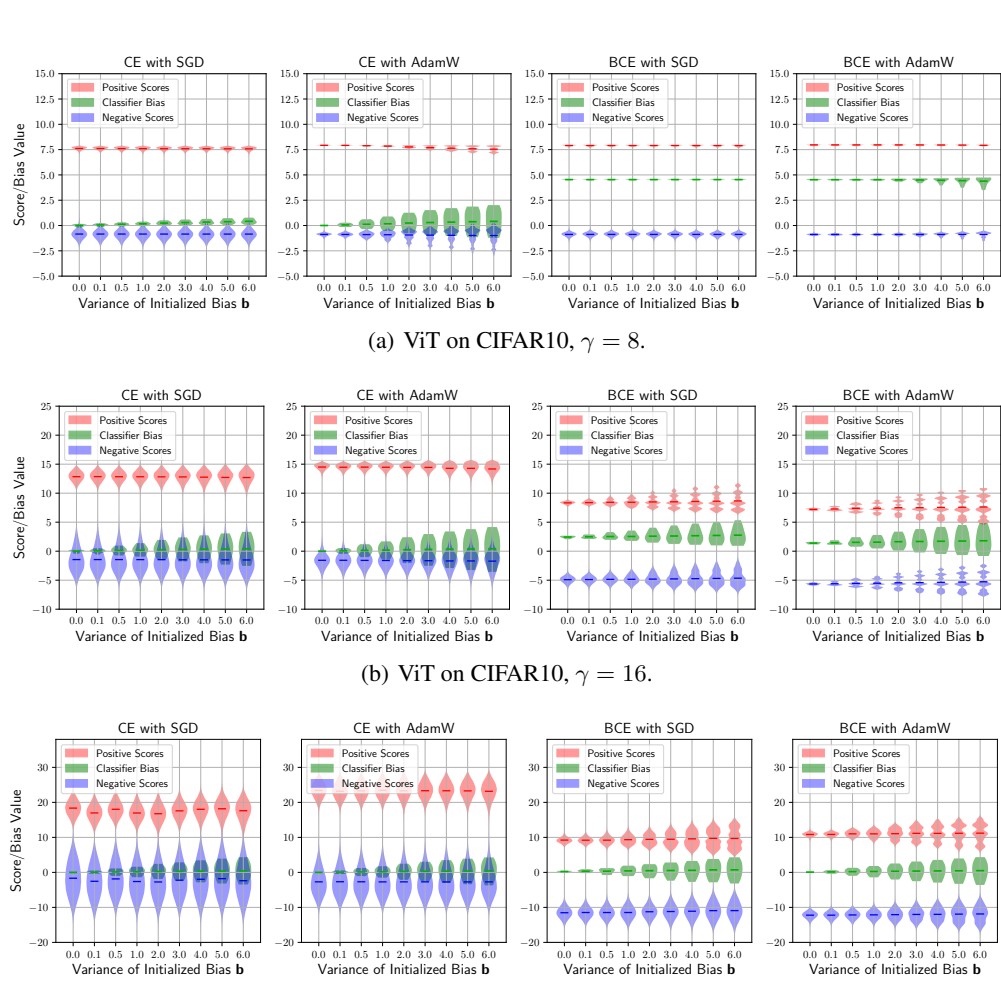

(a) ViT on CIFAR10, $\gamma = 8$.

(b) ViT on CIFAR10, $\gamma = 16$.

(c) ViT on CIFAR10, $\gamma = 32$.

Figure 9: The distribution of the final classifier bias and unbiased positive/negative decision scores of ViT trained on CIFAR10 with varying initial variance of the classifier bias. The different scale factors, $\gamma = 8$ (1-st row), $\gamma = 16$ (2-nd row), and $\gamma = 32$ (3-rd row), also significantly impact the convergence of decision scores and the final classifier bias.

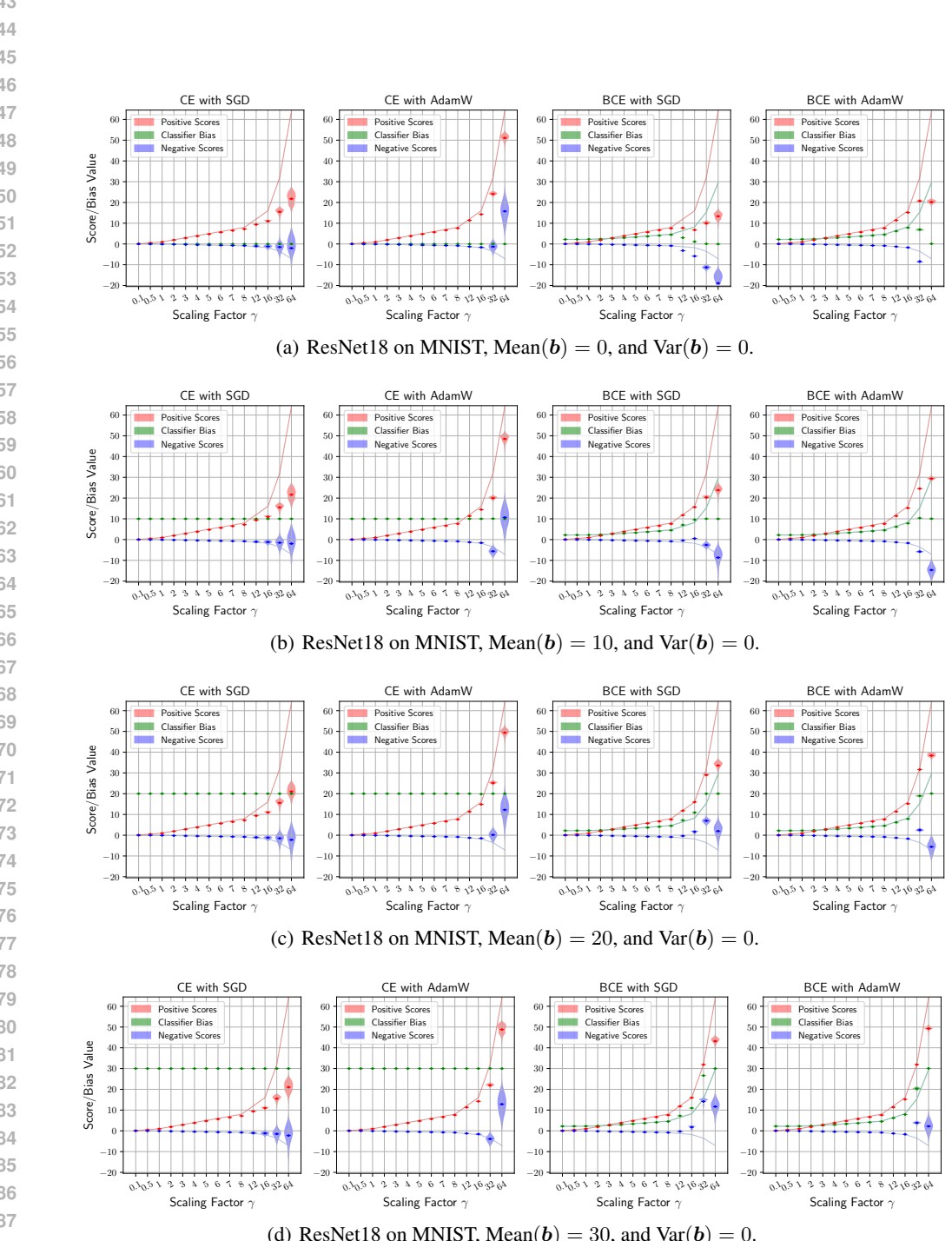

(a) ResNet18 on MNIST, Mean($\boldsymbol{b}$) = 0, and Var($\boldsymbol{b}$) = 0.

(b) ResNet18 on MNIST, Mean($\boldsymbol{b}$) = 10, and Var($\boldsymbol{b}$) = 0.

(c) ResNet18 on MNIST, Mean($\boldsymbol{b}$) = 20, and Var($\boldsymbol{b}$) = 0.

(d) ResNet18 on MNIST, Mean($\boldsymbol{b}$) = 30, and Var($\boldsymbol{b}$) = 0.

Figure 10: The distribution of the final classifier biases and positive/negative unbiased decision scores of ResNet18 trained on MNIST by using various scale factors $\gamma$. The variance of initial classifier biases is set to 0, and their initial mean takes different values, 0, 10, 20, and 30, respectively.

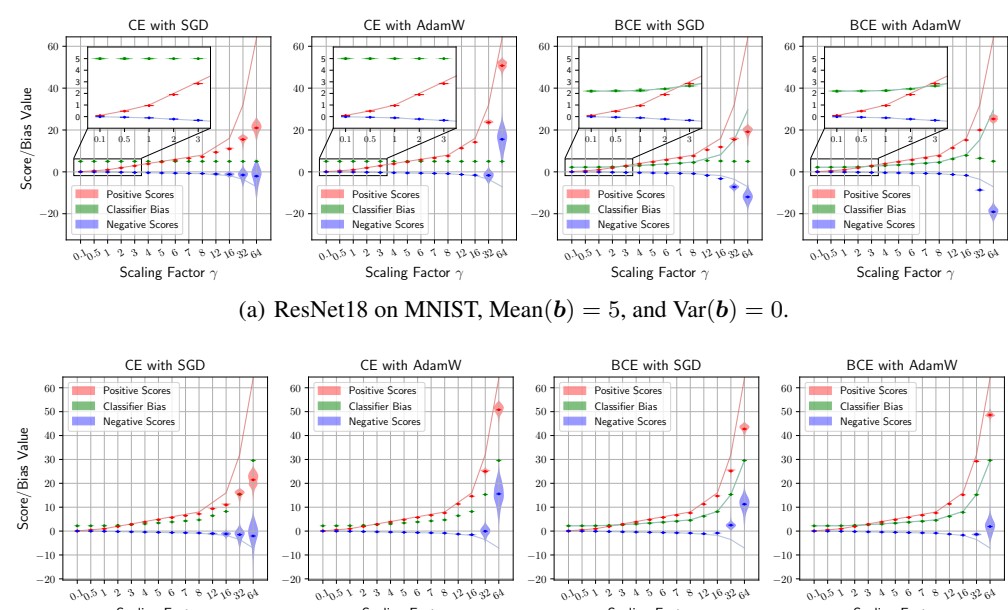

(a) ResNet18 on MNIST, Mean($\boldsymbol{b}$) = 5, and Var($\boldsymbol{b}$) = 0.

(b) ResNet18 on MNIST, Var($\boldsymbol{b}$) = 0, while the mean (Mean($\boldsymbol{b}$)) of initial classifier biases set to the theoretical value of the BCE loss (defined by *Eq.* (11)).

Figure 11: The distribution of the final classifier biases and positive/negative unbiased decision scores of ResNet18 trained on MNIST by using various scale factors $\gamma$. The variance of the initial classifier biases is set to 0.

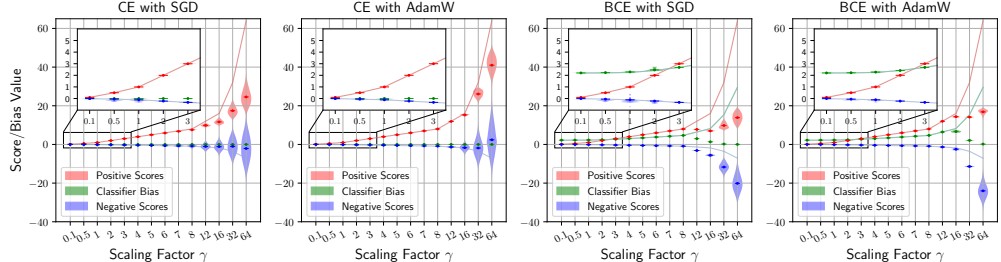

Figure 12: The distribution of the final classifier biases and positive/negative unbiased decision scores of ResNet18 (without the final ReLU activation) trained on CIFAR10 by using various scale factors $\gamma$. The mean and variance of the initial classifier biases is both set to 0.

substantially impact the classification accuracy. However, based on the results of $\mathcal{E}_{\text{com}}$ and $\mathcal{E}_{\text{sep}}$, we conclude that poor classification accuracy $\mathcal{A}$ does not imply that the model has learned unsatisfactory sample features.

Moreover, as analyzed in Sec. 4.1, the classifier biases of the BCE loss play a crucial role in the model training, explicitly constraining the model's positive and negative decision scores, which helps the model to learn features with better intra-class compactness and inter-class separability. One can find that, when $\gamma > 5$, the BCE-trained models usually exhibit higher $\mathcal{E}_{\text{com}}$ and $\mathcal{E}_{\text{sep}}$ than the CE-trained models, which provides strong support for our theoretical conclusions. However, when $\gamma \leq 1$, due to the small normalized feature space, the minor perturbations during the training could significantly affect the feature learning, overshadowing the role of classifier biases in the BCE-trained models, which prevents them from obtaining good feature properties.

Furthermore, Tables 7 and 8 present the numerical results of the final positive/negative unbiased decision scores and classifier biases, for the CE- and BCE-trained ResNet18 and ViT, with $\gamma$ varying from 0.1 to 64. When $\gamma$ is very small, the gap between the converged unbiased positive and negative decision scores is also very small. When $\gamma \leq 0.5$, despite the ResNet18 trained on CIFAR10 being

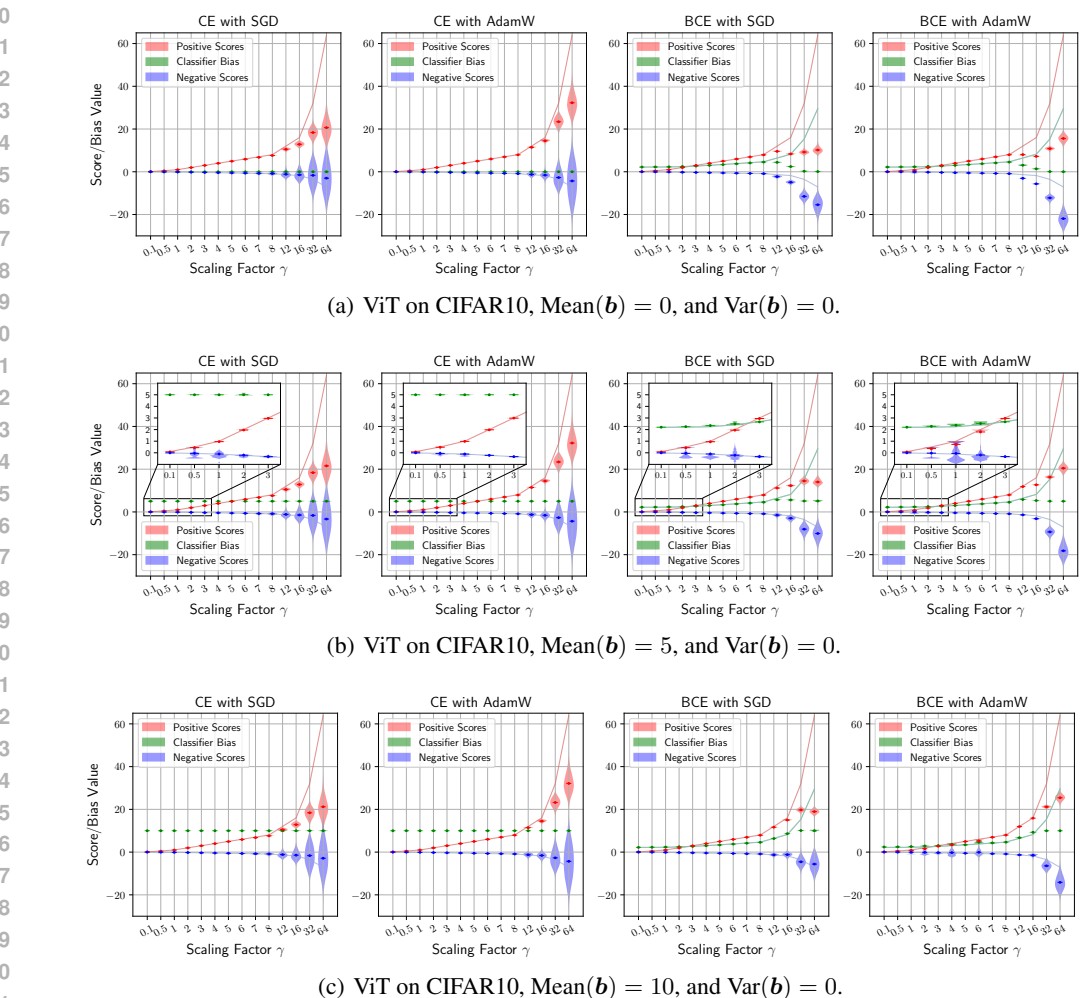

(a) ViT on CIFAR10, Mean($\boldsymbol{b}$) = 0, and Var($\boldsymbol{b}$) = 0.

(b) ViT on CIFAR10, Mean($\boldsymbol{b}$) = 5, and Var($\boldsymbol{b}$) = 0.

(c) ViT on CIFAR10, Mean($\boldsymbol{b}$) = 10, and Var($\boldsymbol{b}$) = 0.

Figure 13: The distribution of the final classifier biases and positive/negative unbiased decision scores of ViT trained on CIFAR10 by using various scale factors $\gamma$. The variance of the initial classifier biases is set to 0.

very close to neural collapse, as indicated by the relatively small variance of its positive and negative unbiased decision scores in Table 7, which are essentially converged, even a slight variance in the classifier bias can still significantly affect its classification accuracy, as indicated by the significantly gap between the classification accuracy $\mathcal{A}$ and the unbiased one $\mathcal{A}^*$ in the Table 5. When $\gamma \leq 2$, due to the disturbances during the model training, the ViT trained on CIFAR10 has not fully converged, which is reflected in the relatively large variance of its positive and negative unbiased decision scores. In fact, as shown in Fig. 13(b), when $\gamma \leq 2$, the distribution areas of the positive and negative unbiased decision scores of the ViT show significant overlapping, resulting in both low classification accuracy $\mathcal{A}$ and the unbiased one $\mathcal{A}^*$ for the ViT in Table 6.

Conversely, a larger $\gamma$ expands the normalized feature space, leading to a large gap between the theoretical values of the converged positive and negative unbiased decision scores, which ensures robust classification; thus, the small perturbations such as the variance of the classifier biases cannot significantly affect the classification results. However, when $\gamma$ is too large, the positive and negative decision scores struggle to converge, as evidenced by the large variance of the final unbiased decision scores, indicating that the sample features have poor intra-class compactness and inter-class separability at this time.

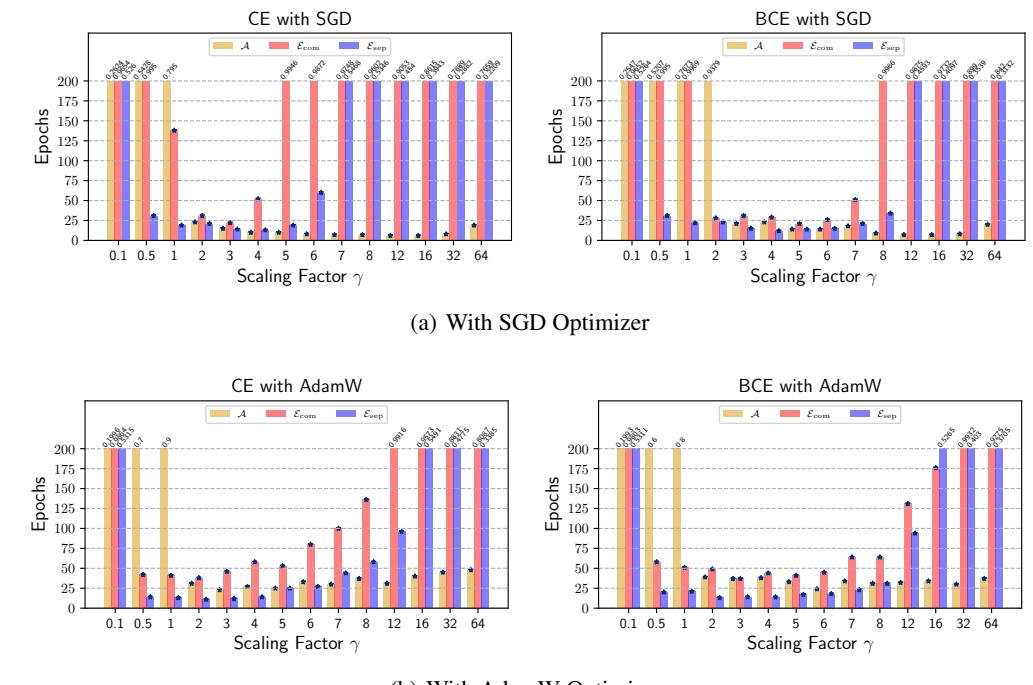

(a) With SGD Optimizer

(b) With AdamW Optimizer

Figure 14: The training epochs at which metrics of classification accuracy $\mathcal{A}$, feature inter-class compactness $\mathcal{E}_{\mathrm{com}}$, and feature inter-class separability $\mathcal{E}_{\mathrm{sep}}$ reach thresholds 0.9975, 0.9975, and 0.5525, respectively, when training ResNet18 (without the final ReLU activation) on CIFAR10 with different scale factors $\gamma$. If a threshold is not reached, show the evaluation at the final epoch (i.e., Epoch=200) in the corresponding bar chart.

Table 5: The classification accuracy (%) and feature properties of ResNet18 (without the final ReLU activation) with various $\gamma$ on training data of CIFAR10. The accuracies of $\mathcal{A}$ and $\mathcal{A}^*$ are calculated using the decision score $\{\gamma \boldsymbol{w}_j^\top \boldsymbol{h}^{(k)} - b_j\}$ and $\{\gamma \boldsymbol{w}_j^\top \boldsymbol{h}^{(k)}\}$, respectively. The higher $\mathcal{E}_{\mathrm{com}}$ and $\mathcal{E}_{\mathrm{sep}}$ respectively indicate the smaller intra-class variability and the bigger inter-class separability. The $\mathrm{Mean}(\boldsymbol{b}) = 0$, and $\mathrm{Var}(\boldsymbol{b}) = 0$.

| $\gamma$ | CE with SGD | | | | BCE with SGD | | | | CE with AdamW | | | | BCE with AdamW | | | |
|---|---|---|---|---|---|---|---|---|---|---|---|---|---|---|---|---|
| | $\mathcal{A}$ | $\mathcal{A}^*$ | $\mathcal{E}_{\mathrm{com}}$ | $\mathcal{E}_{\mathrm{sep}}$ | $\mathcal{A}$ | $\mathcal{A}^*$ | $\mathcal{E}_{\mathrm{com}}$ | $\mathcal{E}_{\mathrm{sep}}$ | $\mathcal{A}$ | $\mathcal{A}^*$ | $\mathcal{E}_{\mathrm{com}}$ | $\mathcal{E}_{\mathrm{sep}}$ | $\mathcal{A}$ | $\mathcal{A}^*$ | $\mathcal{E}_{\mathrm{com}}$ | $\mathcal{E}_{\mathrm{sep}}$ |
| 0.1 | 26.24 | 27.27 | 0.965 | 0.526 | 25.47 | 27.77 | 0.965 | 0.526 | 19.96 | 43.64 | 0.986 | 0.532 | 19.93 | 42.66 | 0.980 | 0.531 |
| 0.5 | 54.28 | 57.00 | 0.995 | 0.554 | 52.07 | 54.75 | 0.995 | 0.554 | 70.00 | 84.59 | 1.0 | 0.555 | 60.00 | 74.56 | 1.0 | 0.555 |
| 1 | 79.50 | 80.01 | 0.998 | 0.555 | 76.73 | 77.76 | 0.997 | 0.555 | 90.00 | 94.17 | 1.0 | 0.555 | 80.00 | 82.80 | 1.0 | 0.555 |
| 2 | 99.97 | 99.97 | 0.999 | 0.555 | 93.79 | 93.78 | 0.999 | 0.553 | 100.0 | 100.0 | 1.0 | 0.555 | 100.0 | 100.0 | 1.0 | 0.555 |
| 3 | 100.0 | 100.0 | 0.999 | 0.555 | 99.97 | 99.97 | 0.999 | 0.555 | 100.0 | 100.0 | 1.0 | 0.555 | 100.0 | 100.0 | 1.0 | 0.555 |
| 4 | 100.0 | 100.0 | 0.998 | 0.555 | 99.99 | 99.99 | 0.999 | 0.555 | 100.0 | 100.0 | 1.0 | 0.555 | 100.0 | 100.0 | 1.0 | 0.555 |
| 5 | 100.0 | 100.0 | 0.995 | 0.555 | 100.0 | 100.0 | 0.999 | 0.555 | 100.0 | 100.0 | 1.0 | 0.555 | 100.0 | 100.0 | 1.0 | 0.555 |
| 6 | 100.0 | 100.0 | 0.987 | 0.553 | 100.0 | 100.0 | 0.999 | 0.555 | 100.0 | 100.0 | 1.0 | 0.555 | 100.0 | 100.0 | 1.0 | 0.555 |
| 7 | 100.0 | 100.0 | 0.975 | 0.547 | 100.0 | 100.0 | 0.998 | 0.555 | 100.0 | 100.0 | 1.0 | 0.555 | 100.0 | 100.0 | 1.0 | 0.555 |
| 8 | 100.0 | 100.0 | 0.960 | 0.535 | 100.0 | 100.0 | 0.997 | 0.554 | 100.0 | 100.0 | 0.999 | 0.555 | 100.0 | 100.0 | 1.0 | 0.555 |
| 12 | 100.0 | 100.0 | 0.905 | 0.454 | 100.0 | 100.0 | 0.988 | 0.459 | 100.0 | 100.0 | 0.992 | 0.555 | 100.0 | 100.0 | 0.999 | 0.555 |
| 16 | 100.0 | 100.0 | 0.862 | 0.394 | 100.0 | 100.0 | 0.973 | 0.410 | 100.0 | 100.0 | 0.957 | 0.549 | 100.0 | 100.0 | 0.998 | 0.527 |
| 32 | 100.0 | 100.0 | 0.789 | 0.288 | 100.0 | 100.0 | 0.899 | 0.354 | 100.0 | 100.0 | 0.883 | 0.478 | 100.0 | 100.0 | 0.993 | 0.403 |
| 64 | 100.0 | 100.0 | 0.766 | 0.221 | 100.0 | 100.0 | 0.842 | 0.313 | 100.0 | 100.0 | 0.809 | 0.339 | 100.0 | 100.0 | 0.928 | 0.371 |

## C.6 BALANCED CLASSIFICATION

To further show the potential of BCE loss, we provide the classification performance on balanced dataset here.

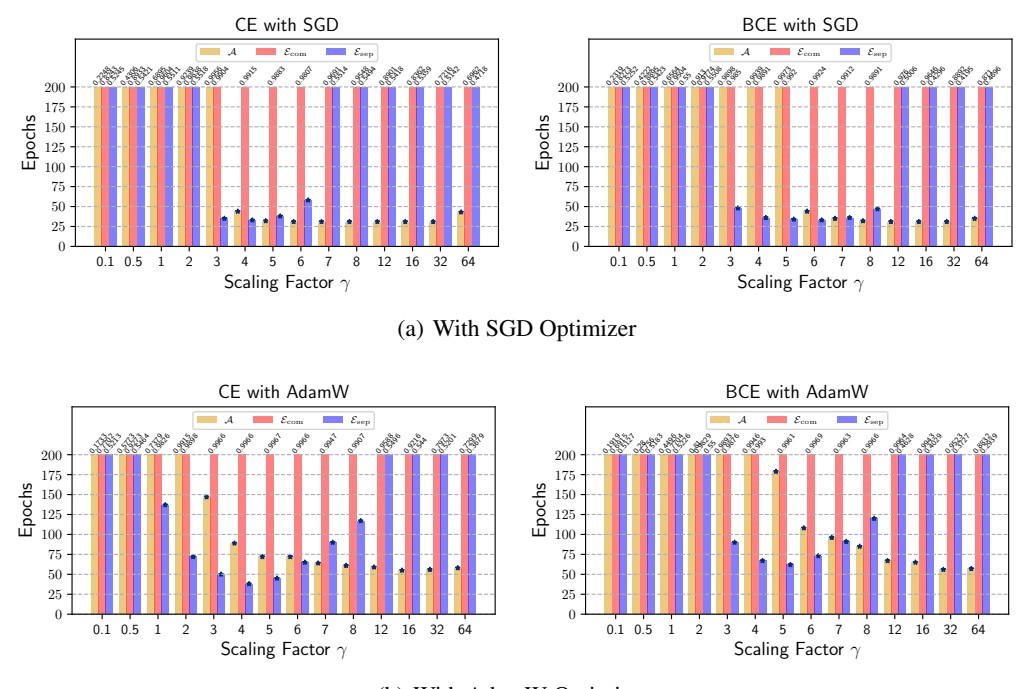

(a) With SGD Optimizer

(b) With AdamW Optimizer

Figure 15: The training epochs at which metrics of classification accuracy $\mathcal{A}$, feature inter-class compactness $\mathcal{E}_{\mathrm{com}}$, and feature inter-class separability $\mathcal{E}_{\mathrm{sep}}$ reach thresholds 0.9975, 0.9975, and 0.5525, respectively, when training ViT on CIFAR10 with different scale factors $\gamma$. If a threshold is not reached, show the evaluation at the final epoch (i.e., Epoch=200) in the corresponding bar chart.

Table 6: The classification accuracy (%) and feature properties of ViT with various $\gamma$ on training data of CIFAR10. The accuracies of $\mathcal{A}$ and $\mathcal{A}^*$ are calculated using the decision score $\{\gamma \boldsymbol{w}_j^\top \boldsymbol{h}^{(k)} - b_j\}$ and $\{\gamma \boldsymbol{w}_j^\top \boldsymbol{h}^{(k)}\}$, respectively. The higher $\mathcal{E}_{\mathrm{com}}$ and $\mathcal{E}_{\mathrm{sep}}$ respectively indicate the smaller intra-class variability and the bigger inter-class separability. The Mean($\boldsymbol{b}$) = 5, and Var($\boldsymbol{b}$) = 0.

| $\gamma$ | CE with SGD | | | | BCE with SGD | | | | CE with AdamW | | | | BCE with AdamW | | | |
|---|---|---|---|---|---|---|---|---|---|---|---|---|---|---|---|---|
| | $\mathcal{A}$ | $\mathcal{A}^*$ | $\mathcal{E}_{\mathrm{com}}$ | $\mathcal{E}_{\mathrm{sep}}$ | $\mathcal{A}$ | $\mathcal{A}^*$ | $\mathcal{E}_{\mathrm{com}}$ | $\mathcal{E}_{\mathrm{sep}}$ | $\mathcal{A}$ | $\mathcal{A}^*$ | $\mathcal{E}_{\mathrm{com}}$ | $\mathcal{E}_{\mathrm{sep}}$ | $\mathcal{A}$ | $\mathcal{A}^*$ | $\mathcal{E}_{\mathrm{com}}$ | $\mathcal{E}_{\mathrm{sep}}$ |
| 0.1 | 27.19 | 27.20 | 0.826 | 0.523 | 26.42 | 26.45 | 0.823 | 0.524 | 23.45 | 23.47 | 0.791 | 0.519 | 20.57 | 20.58 | 0.760 | 0.513 |
| 0.5 | 45.63 | 45.65 | 0.892 | 0.542 | 41.51 | 41.54 | 0.886 | 0.542 | 65.21 | 65.24 | 0.961 | 0.551 | 24.47 | 24.35 | 0.769 | 0.519 |
| 1 | 69.31 | 69.32 | 0.962 | 0.551 | 57.02 | 57.00 | 0.954 | 0.549 | 79.29 | 79.28 | 0.983 | 0.553 | 35.28 | 35.25 | 0.762 | 0.513 |
| 2 | 92.14 | 92.14 | 0.982 | 0.552 | 90.57 | 90.57 | 0.977 | 0.551 | 99.23 | 99.23 | 0.991 | 0.555 | 76.80 | 76.82 | 0.905 | 0.542 |
| 3 | 99.52 | 99.52 | 0.990 | 0.554 | 98.94 | 98.94 | 0.985 | 0.554 | 99.79 | 99.79 | 0.997 | 0.555 | 98.93 | 98.93 | 0.988 | 0.554 |
| 4 | 99.87 | 99.87 | 0.992 | 0.555 | 99.43 | 99.43 | 0.990 | 0.554 | 99.98 | 99.98 | 0.997 | 0.555 | 99.48 | 99.48 | 0.993 | 0.555 |
| 5 | 99.98 | 99.98 | 0.988 | 0.554 | 99.75 | 99.75 | 0.992 | 0.555 | 100.0 | 100.0 | 0.997 | 0.555 | 99.80 | 99.80 | 0.997 | 0.555 |
| 6 | 100.0 | 100.0 | 0.981 | 0.553 | 99.90 | 99.90 | 0.993 | 0.555 | 100.0 | 100.0 | 0.997 | 0.555 | 99.92 | 99.92 | 0.997 | 0.555 |
| 7 | 100.0 | 100.0 | 0.969 | 0.551 | 99.96 | 99.96 | 0.992 | 0.555 | 100.0 | 100.0 | 0.995 | 0.555 | 99.98 | 99.98 | 0.997 | 0.555 |
| 8 | 100.0 | 100.0 | 0.954 | 0.549 | 99.98 | 99.98 | 0.990 | 0.554 | 100.0 | 100.0 | 0.991 | 0.555 | 100.0 | 100.0 | 0.997 | 0.555 |
| 12 | 100.0 | 100.0 | 0.890 | 0.542 | 100.0 | 100.0 | 0.972 | 0.541 | 100.0 | 100.0 | 0.959 | 0.550 | 100.0 | 100.0 | 0.995 | 0.553 |
| 16 | 100.0 | 100.0 | 0.837 | 0.536 | 100.0 | 100.0 | 0.950 | 0.501 | 100.0 | 100.0 | 0.920 | 0.544 | 100.0 | 100.0 | 0.989 | 0.500 |
| 32 | 100.0 | 100.0 | 0.722 | 0.513 | 100.0 | 100.0 | 0.882 | 0.470 | 100.0 | 100.0 | 0.795 | 0.520 | 100.0 | 100.0 | 0.940 | 0.422 |
| 64 | 100.0 | 100.0 | 0.716 | 0.474 | 100.0 | 100.0 | 0.896 | 0.447 | 100.0 | 100.0 | 0.728 | 0.369 | 100.0 | 100.0 | 0.876 | 0.311 |

Table 9 shows the classification results of the models trained on the balanced datasets, CIFAR10, CIFAR100, and Tiny-ImageNet dataset (Wu et al., 2017). It can be observed that, the classification performance of models trained with BCE loss is comparable to or even surpassing that of models trained with CE loss.

Table 7: The final positive/negative unbiased decision scores and classifier biases of ResNet18 (without the final ReLU activation) with various $\gamma$ on training data of CIFAR10 (mean $\pm$ standard deviation). The symbols $\boldsymbol{s}^{(kk)}$ and $\boldsymbol{s}^{(jk)}$ represent the positive and negative unbiased decision scores, respectively. The smaller standard deviation implies that the scores and biases tend to converge to similar values after training. The Mean($\boldsymbol{b}$) = 0, and Var($\boldsymbol{b}$) = 0.

(a) ResNet18 on CIFAR10 with SGD optimizer

| $\gamma$ | CE with SGD | | | BCE with SGD | | |
|---|---|---|---|---|---|---|
| | $\boldsymbol{s}^{(kk)}$ | $\boldsymbol{s}^{(jk)}$ | $\boldsymbol{b}$ | $\boldsymbol{s}^{(kk)}$ | $\boldsymbol{s}^{(jk)}$ | $\boldsymbol{b}$ |
| 0.1 | $0.10 \pm 0.02$ | $-0.01 \pm 0.10$ | $0.00 \pm 0.02$ | $0.10 \pm 0.02$ | $-0.01 \pm 0.10$ | $2.20 \pm 0.02$ |
| 0.5 | $0.50 \pm 0.03$ | $-0.05 \pm 0.26$ | $0.00 \pm 0.02$ | $0.50 \pm 0.03$ | $-0.05 \pm 0.26$ | $2.24 \pm 0.02$ |
| 1 | $1.00 \pm 0.04$ | $-0.11 \pm 0.31$ | $0.00 \pm 0.04$ | $1.00 \pm 0.05$ | $-0.11 \pm 0.34$ | $2.29 \pm 0.03$ |
| 2 | $2.00 \pm 0.04$ | $-0.22 \pm 0.02$ | $0.00 \pm 0.00$ | $2.00 \pm 0.05$ | $-0.22 \pm 0.34$ | $2.46 \pm 0.14$ |
| 3 | $3.00 \pm 0.02$ | $-0.33 \pm 0.02$ | $0.00 \pm 0.00$ | $3.00 \pm 0.06$ | $-0.33 \pm 0.03$ | $2.68 \pm 0.00$ |
| 4 | $3.99 \pm 0.01$ | $-0.44 \pm 0.05$ | $0.00 \pm 0.00$ | $4.00 \pm 0.05$ | $-0.44 \pm 0.03$ | $3.02 \pm 0.00$ |
| 5 | $4.97 \pm 0.01$ | $-0.55 \pm 0.11$ | $0.00 \pm 0.00$ | $4.99 \pm 0.00$ | $-0.55 \pm 0.03$ | $3.40 \pm 0.00$ |
| 6 | $5.91 \pm 0.03$ | $-0.66 \pm 0.23$ | $0.00 \pm 0.02$ | $5.99 \pm 0.00$ | $-0.67 \pm 0.05$ | $3.81 \pm 0.00$ |
| 7 | $6.78 \pm 0.07$ | $-0.75 \pm 0.41$ | $0.00 \pm 0.04$ | $6.98 \pm 0.01$ | $-0.78 \pm 0.07$ | $4.23 \pm 0.00$ |
| 8 | $7.55 \pm 0.13$ | $-0.84 \pm 0.60$ | $0.00 \pm 0.03$ | $7.96 \pm 0.01$ | $-0.89 \pm 0.10$ | $4.65 \pm 0.01$ |
| 12 | $9.86 \pm 0.53$ | $-1.05 \pm 1.30$ | $0.00 \pm 0.02$ | $7.76 \pm 0.10$ | $-3.15 \pm 0.23$ | $3.00 \pm 0.01$ |
| 16 | $11.65 \pm 0.98$ | $-1.14 \pm 1.97$ | $0.00 \pm 0.01$ | $7.01 \pm 0.22$ | $-5.59 \pm 0.43$ | $1.35 \pm 0.03$ |
| 32 | $17.44 \pm 2.32$ | $-1.02 \pm 4.08$ | $0.00 \pm 0.01$ | $9.79 \pm 1.39$ | $-11.70 \pm 2.09$ | $0.20 \pm 0.04$ |
| 64 | $24.56 \pm 4.37$ | $-2.10 \pm 6.86$ | $0.00 \pm 0.02$ | $13.87 \pm 2.55$ | $-20.13 \pm 4.55$ | $0.02 \pm 0.16$ |

(b) ResNet18 on CIFAR10 with AdamW optimizer

| $\gamma$ | CE with AdamW | | | BCE with AdamW | | |
|---|---|---|---|---|---|---|
| | $\boldsymbol{s}^{(kk)}$ | $\boldsymbol{s}^{(jk)}$ | $\boldsymbol{b}$ | $\boldsymbol{s}^{(kk)}$ | $\boldsymbol{s}^{(jk)}$ | $\boldsymbol{b}$ |
| 0.1 | $0.10 \pm 0.02$ | $-0.01 \pm 0.10$ | $0.00 \pm 0.02$ | $0.10 \pm 0.02$ | $-0.01 \pm 0.10$ | $2.21 \pm 0.02$ |
| 0.5 | $0.50 \pm 0.00$ | $-0.06 \pm 0.15$ | $0.00 \pm 0.03$ | $0.50 \pm 0.00$ | $-0.06 \pm 0.18$ | $2.22 \pm 0.01$ |
| 1 | $1.00 \pm 0.00$ | $-0.11 \pm 0.18$ | $0.00 \pm 0.03$ | $1.00 \pm 0.00$ | $-0.11 \pm 0.25$ | $2.27 \pm 0.04$ |
| 2 | $2.00 \pm 0.00$ | $-0.22 \pm 0.00$ | $0.00 \pm 0.00$ | $2.00 \pm 0.00$ | $-0.22 \pm 0.00$ | $2.41 \pm 0.00$ |
| 3 | $3.00 \pm 0.00$ | $-0.33 \pm 0.00$ | $0.00 \pm 0.00$ | $3.00 \pm 0.00$ | $-0.33 \pm 0.00$ | $2.68 \pm 0.00$ |
| 4 | $4.00 \pm 0.00$ | $-0.44 \pm 0.01$ | $0.00 \pm 0.00$ | $4.00 \pm 0.00$ | $-0.44 \pm 0.01$ | $3.02 \pm 0.00$ |
| 5 | $5.00 \pm 0.00$ | $-0.56 \pm 0.02$ | $0.00 \pm 0.00$ | $5.00 \pm 0.00$ | $-0.56 \pm 0.01$ | $3.40 \pm 0.00$ |
| 6 | $6.00 \pm 0.00$ | $-0.67 \pm 0.03$ | $0.00 \pm 0.00$ | $6.00 \pm 0.00$ | $-0.67 \pm 0.02$ | $3.81 \pm 0.00$ |
| 7 | $7.00 \pm 0.00$ | $-0.78 \pm 0.03$ | $0.00 \pm 0.00$ | $7.00 \pm 0.00$ | $-0.78 \pm 0.02$ | $4.24 \pm 0.00$ |
| 8 | $7.99 \pm 0.00$ | $-0.89 \pm 0.05$ | $0.00 \pm 0.00$ | $8.00 \pm 0.00$ | $-0.89 \pm 0.03$ | $4.67 \pm 0.00$ |
| 12 | $11.90 \pm 0.08$ | $-1.32 \pm 0.45$ | $0.01 \pm 0.01$ | $11.99 \pm 0.00$ | $-1.33 \pm 0.05$ | $6.44 \pm 0.00$ |
| 16 | $15.26 \pm 0.37$ | $-1.68 \pm 1.52$ | $0.01 \pm 0.02$ | $14.35 \pm 0.32$ | $-2.48 \pm 0.37$ | $6.66 \pm 0.40$ |
| 32 | $26.09 \pm 1.84$ | $-1.85 \pm 4.64$ | $0.00 \pm 0.02$ | $14.14 \pm 0.21$ | $-11.40 \pm 0.33$ | $2.09 \pm 0.14$ |
| 64 | $40.98 \pm 5.59$ | $2.40 \pm 7.98$ | $0.00 \pm 0.06$ | $16.85 \pm 1.24$ | $-24.01 \pm 3.88$ | $0.17 \pm 0.04$ |

It is worth noting that, when training ViT on the CIFAR10, CIFAR100, and Tiny-ImageNet datasets, its classification performance is significantly worse than that of the ResNet family, which is intuitive and reasonable. As noted by Zhu et al. (2023); Zhang et al. (2022), due to the influence of factors such as model architecture and inductive biases, the representation of ViT trained on small datasets is hugely different from ViT trained on large datasets, which may be the reason why the performance drops a lot on these small datasets. Setting aside the inherent differences between these models, we still find that training with the BCE loss can achieve performance on par with, or even surpassing, that of the CE loss. The BCE loss helps the model learn deep features with better intra-class compactness and inter-class separability, which is usually and intuitively beneficial for classification tasks.

## C.7 OPEN-SET RECOGNITION PERFORMANCE

In Sec. 5, we present the open-set recognition (OSR) performance of models trained by BCE and CE losses. To further analyze their differences and the impact of classifier biases in practice, we conducted additional OSR experiments with and without the classifier biases, respectively setting the

Table 8: The final positive/negative unbiased decision scores and classifier biases of ViT with varying $\gamma$ on training data of CIFAR10 (mean $\pm$ standard deviation). The symbols $\boldsymbol{s}^{(kk)}$ and $\boldsymbol{s}^{(jk)}$ represent the positive and negative unbiased decision scores, respectively. The smaller standard deviation implies that the scores and biases tend to converge to similar values after training. The $\text{Mean}(\boldsymbol{b}) = 5$, and $\text{Var}(\boldsymbol{b}) = 0$.

(a) ViT on CIFAR10 with SGD optimizer

| $\gamma$ | CE with SGD | | | BCE with SGD | | |
|---|---|---|---|---|---|---|
| | $\boldsymbol{s}^{(kk)}$ | $\boldsymbol{s}^{(jk)}$ | $\boldsymbol{b}$ | $\boldsymbol{s}^{(kk)}$ | $\boldsymbol{s}^{(jk)}$ | $\boldsymbol{b}$ |
| 0.1 | $0.08 \pm 0.06$ | $-0.01 \pm 0.10$ | $5.00 \pm 0.01$ | $0.08 \pm 0.06$ | $-0.01 \pm 0.10$ | $2.20 \pm 0.02$ |
| 0.5 | $0.44 \pm 0.15$ | $-0.05 \pm 0.34$ | $5.00 \pm 0.02$ | $0.44 \pm 0.16$ | $-0.05 \pm 0.36$ | $2.25 \pm 0.01$ |
| 1 | $0.96 \pm 0.19$ | $-0.11 \pm 0.40$ | $5.00 \pm 0.02$ | $0.95 \pm 0.22$ | $-0.10 \pm 0.52$ | $2.35 \pm 0.05$ |
| 2 | $1.96 \pm 0.24$ | $-0.21 \pm 0.35$ | $5.00 \pm 0.08$ | $1.95 \pm 0.26$ | $-0.21 \pm 0.48$ | $2.48 \pm 0.12$ |
| 3 | $2.97 \pm 0.22$ | $-0.33 \pm 0.10$ | $5.00 \pm 0.00$ | $2.95 \pm 0.33$ | $-0.33 \pm 0.13$ | $2.66 \pm 0.00$ |
| 4 | $3.97 \pm 0.15$ | $-0.44 \pm 0.11$ | $5.00 \pm 0.01$ | $3.96 \pm 0.32$ | $-0.44 \pm 0.13$ | $2.99 \pm 0.00$ |
| 5 | $4.94 \pm 0.08$ | $-0.55 \pm 0.16$ | $5.00 \pm 0.02$ | $4.96 \pm 0.27$ | $-0.55 \pm 0.13$ | $3.35 \pm 0.01$ |
| 6 | $5.88 \pm 0.07$ | $-0.65 \pm 0.26$ | $5.00 \pm 0.04$ | $5.96 \pm 0.20$ | $-0.66 \pm 0.15$ | $3.74 \pm 0.01$ |
| 7 | $6.77 \pm 0.11$ | $-0.75 \pm 0.39$ | $5.00 \pm 0.07$ | $6.94 \pm 0.15$ | $-0.77 \pm 0.17$ | $4.14 \pm 0.01$ |
| 8 | $7.60 \pm 0.17$ | $-0.84 \pm 0.54$ | $5.00 \pm 0.11$ | $7.92 \pm 0.12$ | $-0.88 \pm 0.23$ | $4.55 \pm 0.02$ |
| 12 | $10.51 \pm 0.53$ | $-1.17 \pm 1.29$ | $5.00 \pm 0.22$ | $11.18 \pm 0.17$ | $-1.50 \pm 0.63$ | $5.70 \pm 0.14$ |
| 16 | $12.85 \pm 0.91$ | $-1.45 \pm 2.18$ | $5.00 \pm 0.22$ | $12.31 \pm 0.32$ | $-2.96 \pm 0.97$ | $5.61 \pm 0.17$ |
| 32 | $18.38 \pm 2.38$ | $-1.64 \pm 5.03$ | $5.00 \pm 0.11$ | $14.51 \pm 1.03$ | $-8.08 \pm 2.18$ | $5.11 \pm 0.10$ |
| 64 | $21.56 \pm 4.99$ | $-3.33 \pm 7.85$ | $5.00 \pm 0.13$ | $14.01 \pm 1.32$ | $-10.09 \pm 3.29$ | $5.14 \pm 0.58$ |

(b) ViT on CIFAR10 with AdamW optimizer

| $\gamma$ | CE with AdamW | | | BCE with AdamW | | |
|---|---|---|---|---|---|---|
| | $\boldsymbol{s}^{(kk)}$ | $\boldsymbol{s}^{(jk)}$ | $\boldsymbol{b}$ | $\boldsymbol{s}^{(kk)}$ | $\boldsymbol{s}^{(jk)}$ | $\boldsymbol{b}$ |
| 0.1 | $0.08 \pm 0.06$ | $-0.01 \pm 0.10$ | $5.00 \pm 0.01$ | $0.07 \pm 0.07$ | $-0.00 \pm 0.10$ | $2.20 \pm 0.01$ |
| 0.5 | $0.48 \pm 0.10$ | $-0.05 \pm 0.22$ | $5.00 \pm 0.01$ | $0.36 \pm 0.27$ | $-0.03 \pm 0.43$ | $2.28 \pm 0.06$ |
| 1 | $0.98 \pm 0.14$ | $-0.11 \pm 0.31$ | $5.00 \pm 0.04$ | $0.72 \pm 0.41$ | $-0.05 \pm 0.62$ | $2.38 \pm 0.11$ |
| 2 | $1.98 \pm 0.19$ | $-0.22 \pm 0.07$ | $5.00 \pm 0.00$ | $1.80 \pm 0.55$ | $-0.19 \pm 0.59$ | $2.52 \pm 0.14$ |
| 3 | $2.99 \pm 0.15$ | $-0.33 \pm 0.06$ | $5.00 \pm 0.01$ | $2.96 \pm 0.34$ | $-0.33 \pm 0.12$ | $2.67 \pm 0.00$ |
| 4 | $3.99 \pm 0.07$ | $-0.44 \pm 0.06$ | $5.00 \pm 0.01$ | $3.97 \pm 0.31$ | $-0.44 \pm 0.11$ | $3.00 \pm 0.00$ |
| 5 | $4.98 \pm 0.04$ | $-0.55 \pm 0.05$ | $5.00 \pm 0.01$ | $4.98 \pm 0.24$ | $-0.55 \pm 0.09$ | $3.37 \pm 0.00$ |
| 6 | $5.98 \pm 0.04$ | $-0.66 \pm 0.06$ | $5.00 \pm 0.01$ | $5.98 \pm 0.18$ | $-0.66 \pm 0.08$ | $3.76 \pm 0.00$ |
| 7 | $6.96 \pm 0.02$ | $-0.77 \pm 0.10$ | $5.00 \pm 0.02$ | $6.98 \pm 0.11$ | $-0.78 \pm 0.08$ | $4.14 \pm 0.02$ |
| 8 | $7.92 \pm 0.04$ | $-0.88 \pm 0.20$ | $5.00 \pm 0.03$ | $7.98 \pm 0.05$ | $-0.89 \pm 0.06$ | $4.53 \pm 0.01$ |
| 12 | $11.44 \pm 0.26$ | $-1.26 \pm 0.84$ | $5.00 \pm 0.03$ | $11.85 \pm 0.04$ | $-1.36 \pm 0.14$ | $5.82 \pm 0.09$ |
| 16 | $14.47 \pm 0.58$ | $-1.57 \pm 1.37$ | $5.00 \pm 0.02$ | $12.68 \pm 0.13$ | $-3.16 \pm 0.29$ | $5.32 \pm 0.06$ |
| 32 | $23.39 \pm 2.27$ | $-2.67 \pm 3.99$ | $5.00 \pm 0.01$ | $16.29 \pm 0.60$ | $-9.34 \pm 1.51$ | $5.02 \pm 0.01$ |
| 64 | $32.25 \pm 4.95$ | $-4.43 \pm 8.43$ | $5.00 \pm 0.01$ | $20.45 \pm 1.60$ | $-18.18 \pm 3.38$ | $5.01 \pm 0.01$ |

mean of initial classifier biases as $0, 5$, and $10$ when the classifier biases were included in the losses. Table 10 presents the OSR results.

It can be observed that when the losses taking the classifier biases, regardless of their initial mean $(0, 5$, or $10$), the BCE-trained models consistently outperforms the CE-trained ones in the OSR experiments. In contrast, without the classifier biases, the OSR performance of BCE-trained models experiences a dramatic decline (over $8.92\%$ degradation), compared with the CE-trained models, especially on MNIST and SVHN. These results demonstrate again that the classifier biases play a crucial role in the training with BCE loss.

Table 9: The classification accuracy (%) on the test datasets of CIFAR10, CIFAR100 and Tiny-ImageNet. We conducted learning rate optimization for each experimental combination while keeping other hyperparameters fixed, and the $\gamma = 32$.

| $\mathcal{M}$ | Loss | CIFAR10 | | CIFAR100 | | Tiny $-$ ImageNet | |
|---|---|---|---|---|---|---|---|
| | | SGD | AdamW | SGD | AdamW | SGD | AdamW |
| ResNet18 | CE | 95.90 | **96.24** | 78.39 | 78.05 | 68.10 | 67.13 |
| | BCE | **95.97** | 95.96 | **78.46** | **78.38** | **69.10** | **69.12** |
| ResNet50 | CE | 96.30 | **96.51** | **80.89** | 78.06 | 71.65 | 69.49 |
| | BCE | **96.40** | 96.24 | 80.87 | **79.89** | **72.95** | **70.82** |
| ViT | CE | 79.51 | 87.00 | **64.82** | 63.19 | 50.42 | **54.83** |
| | BCE | **82.00** | **87.57** | 64.79 | **64.91** | **51.15** | 53.91 |

Table 10: The results of open-set recognition. Results are averaged among five randomized trials and the scale factor $\gamma = 32$.

| Bias Mean | Metrics | MNIST | | SVHN | | CIFAR10 | | CIFAR+50 | |
|---|---|---|---|---|---|---|---|---|---|
| | | CE | BCE | CE | BCE | CE | BCE | CE | BCE |
| – | AUROC | **96.74** | 90.22 | **94.44** | 66.60 | **74.34** | 68.14 | **78.34** | 62.17 |
| | OSCR | **96.59** | 90.04 | **92.56** | 71.00 | 60.10 | **67.00** | 61.22 | **63.87** |
| 0 | AUROC | 98.63 | **99.29** | 94.42 | **95.21** | 83.86 | **85.70** | 87.82 | **90.06** |
| | OSCR | 98.50 | **99.09** | 92.53 | **93.49** | 81.54 | **83.50** | 85.81 | **88.43** |
| 5 | AUROC | 98.58 | **99.15** | 94.30 | **95.07** | 84.17 | **85.78** | 87.76 | **90.15** |
| | OSCR | 98.46 | **98.96** | 92.42 | **93.37** | 81.87 | **83.66** | 85.71 | **88.51** |
| 10 | AUROC | 98.66 | **99.21** | 94.49 | **95.07** | 84.02 | **85.51** | 87.55 | **89.72** |
| | OSCR | 98.53 | **99.04** | 92.64 | **93.37** | 81.72 | **83.35** | 85.62 | **88.08** |

## D MORE DISCUSSION

**A related work.** A recent work Li et al. (2025) also analyzes the connections and differences between BCE and CE losses from the perspective of neural collapse, while it focus on the comparison of BCE and CE in the Euclidean feature space. In their comparison, the authors added regularization terms with a weight decay factor $\lambda$ in both the BCE and CE losses. Intuitively, the value of weight decay factor $\lambda$ can implicitly adjust the size of the feature space after the model training; a larger $\lambda$ corresponds to a smaller feature space, while a smaller $\lambda$ corresponds to a larger feature space. However, there is no strict correspondence between $\lambda$ and the size of the final feature space, as the final trained feature distribution in Euclidean space also depends on other factors such as the training strategy, dataset, and hyperparameter settings. In contrast, in the normalized feature space, the scale factor $\gamma$ explicitly determines the size of the feature space before the model training, allowing us to more rigorously analyze the geometric effects of the scale factor in the deep feature learning.

**Limitations and future works.** In the normalized feature space, this paper compares the differences between the BCE and CE in deep feature learning, from the perspective of neural collapse, revealing the geometric effects of the scale factor. Nevertheless, this work has some limitations and presents avenues for further research in the future.

- In the paper, we took the positive and negative unbiased decision scores to analyze the intra-class compactness and inter-class separability of sample features after the model training. However, the unbiased decision scores rely on classifier vectors as anchors and indirectly reflect the feature properties, without directly measuring the distances or similarities between the sample features. In the future, we will investigate the contrastive learning field by analyzing loss functions represented in forms of the BCE and CE, revealing their differences in feature learning.

- The theoretical results in this paper are based on the balanced datasets. Although we experimentally validated the advantages of the BCE on the imbalanced long-tail datasets, is there still a general advantage of BCE over CE on the imbalanced datasets? When BCE reaches a minimum on the imbalanced datasets, does it still lead to neural collapse? Alternatively, is BCE more likely to induce neural collapse than CE on the imbalanced datasets? These questions currently lack theoretical research.

- For the open-set recognition tasks, we conducted the simple experiments to illustrate the performance advantages of the BCE over CE, which lacks a systematic and thorough theoretical analysis and validation.

- In contrastive learning He et al. (2020); Chen et al. (2020), the temperature coefficient $\tau$ in the denominator plays a role similar to that of the scale factor $\gamma$, as it can adjust the size of the normalized feature space. However, whether it exhibits a geometric effect similar to that of the scale factor still requires further analysis. Additionally, the temperature coefficient $\tau$ is also commonly used in knowledge distillation (Zheng & Yang, 2024; Yang et al., 2024), and its magnitude profoundly affects the final performance of the model. Does it possess geometric effects similar to those of the scale factor? In the future, we will analyze the minimization problems of loss functions such as BCE and CE in the contrastive learning and distillation learning through theoretical analysis, revealing the role of the temperature coefficient in feature learning and knowledge distillation.

# E    PROOF OF THEOREMS

In the normalized feature space, Theorems 1 and 2 have concluded the neural collapse of the CE and BCE losses on balanced datasets. In this section, we present their detail proofs.

## E.1    BASICS

Before providing the proofs, we first formally present the definition of canonical $K$-simplex ETF in $d$-dimension space, and two lemmas required by the proofs.

**Definition 7.** *(K-simplex ETF) A K-simplex ETF is a collection of vectors in $\mathbb{R}^d$ specified by the columns of matrix*

$$\boldsymbol{M} = \sqrt{\frac{K}{K-1}} \boldsymbol{P} \left( \boldsymbol{I}_K - \frac{1}{K} \mathbf{1}_K \mathbf{1}_K^\top \right), \tag{25}$$

*where $\boldsymbol{I}_K \in \mathbb{R}^{K \times K}$ is the identity matrix and $\mathbf{1}_K \in \mathbb{R}^K$ is the ones vector, and $\boldsymbol{P} \in \mathbb{R}^{d \times K}(d \geq K)$ is a partial-orthogonal matrix such that $\boldsymbol{P}^\top \boldsymbol{P} = \boldsymbol{I}_K$.*

*Note that the matrix $\boldsymbol{M}$ satisfies:*

$$\boldsymbol{M}^\top \boldsymbol{M} = \frac{K}{K-1} \left( \boldsymbol{I}_K - \frac{1}{K} \mathbf{1}_K \mathbf{1}_K^\top \right). \tag{26}$$

**Lemma 8.** *(Young's Inequality) Let $p, q$ be positive numbers satisfying $\frac{1}{p} + \frac{1}{q} = 1$. Then for any $a, b \in \mathbb{R}$, we have*

$$|ab| \leq \frac{|a|^p}{p} + \frac{|b|^q}{q}, \tag{27}$$

*where the equality holds if and only if $|a|^p = |b|^q$. The case for $p = q = 2$ is just the AM-GM inequality which is $|ab| \leq \frac{1}{2}(a^2 + b^2)$, where the equality holds if and only if $|a| = |b|$.*

**Lemma 9.** *(Jensen's Inequality) Let $f$ be a convex function on an interval $I$. Then*

$$f(\lambda_1 x_1 + \cdots + \lambda_n x_n) \leq \lambda_1 f(x_1) + \cdots + \lambda_n f(x_n), \tag{28}$$

*where*

$$x_1, x_2, \cdots, x_n \in I, \quad \lambda_1, \lambda_2, \cdots, \lambda_n \in [0, 1]. \tag{29}$$

*When $f$ is strictly convex, let*

$$I_1 = \{k : \lambda_k \in (0, 1]\}, \text{ and } I_2 = \{k : \lambda_k = 0\}. \tag{30}$$

*The equality holds if and only if all $x_k, k \in I_1$ are equal. In practice, if the function $f$ is concave like $\exp(x), \log(x), \text{ or } \log(1 + e^x)$, Jensen's Inequality can be written as*

$$\exp\left( \frac{\sum_{i=1}^n x_i}{n} \right) \leq \frac{\sum_{i=1}^n \exp(x_i)}{n}, \tag{31}$$

$$\log\left( \frac{\sum_{i=1}^n x_i}{n} \right) \geq \frac{\sum_{i=1}^n \log(x_i)}{n}, \tag{32}$$

$$\log\left( 1 + \exp\left( \frac{\sum_{i=1}^n x_i}{n} \right) \right) \geq \frac{1}{n} \sum_{i=1}^n \log\left( 1 + \exp(x_i) \right). \tag{33}$$

## E.2    PROOF OF THEOREM 2

**Theorem 10.** *Assume that the feature dimension $d$ is larger than the number of classes $K$, i.e., $d \geq K - 1$, and $\gamma > 0$. In the normalized feature space, any global minimizer $(\boldsymbol{W}^\star, \boldsymbol{H}^\star, \boldsymbol{b}^\star)$ of*

$$f_{\text{bce}}(\boldsymbol{W}\boldsymbol{H}, \boldsymbol{b}) \triangleq \frac{1}{nK} \sum_{k=1}^K \sum_{i=1}^n \mathcal{L}_{\text{bce}}(\gamma \boldsymbol{W} \boldsymbol{h}_i^{(k)} - \boldsymbol{b}) \tag{34}$$

*with $\|\boldsymbol{w}_j\|_2 = \|\boldsymbol{h}_i^{(k)}\|_2 = 1$ for $\forall j, k \in [K], i \in [n]$, obeys the following*

$$\boldsymbol{w}_k^\star = \boldsymbol{h}_i^{(k)\star}, \quad \forall k \in [K], \ i \in [n], \tag{35}$$

$$\tilde{\boldsymbol{h}}_i^\star := \frac{1}{K}\sum_{j=1}^{K} \boldsymbol{h}_i^{(k)\star} = \boldsymbol{0}, \quad \forall i \in [n], \tag{36}$$

$$\boldsymbol{W}^\star \boldsymbol{W}^{\star\top} = \frac{K}{K-1}\left(\boldsymbol{I}_K - \frac{1}{K}\boldsymbol{1}_K\boldsymbol{1}_K^\top\right), \tag{37}$$

$$\boldsymbol{b}^\star = b^\star \boldsymbol{1}, \tag{38}$$

*where $b^\star$ satisfies*

$$b^\star = \log \frac{(K-2)\mathrm{e}^{-\frac{\gamma}{K-1}} + \sqrt{(K-2)^2\mathrm{e}^{-\frac{2\gamma}{K-1}} + 4(K-1)\mathrm{e}^{-\frac{\gamma}{K-1}+\gamma}}}{2}$$

$$= -\frac{\gamma}{K-1} + \log \frac{(K-2) + \sqrt{(K-2)^2 + 4(K-1)\mathrm{e}^{\gamma+\frac{\gamma}{K-1}}}}{2}. \tag{39}$$

*For any minimizer $(\boldsymbol{W}^\star, \boldsymbol{H}^\star, \boldsymbol{b}^\star)$,*

$$f_{\mathrm{bce}}(\boldsymbol{W}^\star, \boldsymbol{H}^\star, \boldsymbol{b}^\star) = -2\gamma - (K-2)b^\star + \log\left(1 + \mathrm{e}^{\gamma-b^\star}\right) + (K-1)\log\left(1 + \mathrm{e}^{b^\star+\frac{\gamma}{K-1}}\right). \tag{40}$$

*Proof.* According to Lemma 11, $f_{\mathrm{bce}}(\boldsymbol{W}, \boldsymbol{H}, \boldsymbol{b}) \geq f'_{\mathrm{bce}}(\boldsymbol{W}\boldsymbol{H}, \boldsymbol{b})$ and the equality holds if and only if $(\boldsymbol{W}, \boldsymbol{H}, \boldsymbol{b}) \in \mathbb{D}_1$. According to Lemma 14,

$$f'_{\mathrm{bce}}(\boldsymbol{W}, \boldsymbol{H}, \boldsymbol{b}) \geq -2\gamma - (K-2)b^\star + \log\left(1 + \mathrm{e}^{\gamma-b^\star}\right) + (K-1)\log\left(1 + \mathrm{e}^{b^\star+\frac{\gamma}{K-1}}\right), \tag{41}$$

and $f'_{\mathrm{bce}}(\boldsymbol{W}, \boldsymbol{H}, \boldsymbol{b})$ achieves the above lower boundary if and only if $(\boldsymbol{W}, \boldsymbol{H}, \boldsymbol{b}) \in \mathbb{D}_5$. Therefore, on $\mathbb{D}_1$, $f_{\mathrm{bce}}(\boldsymbol{W}, \boldsymbol{H}, \boldsymbol{b})$ achieves its minimum value if and only if $(\boldsymbol{W}, \boldsymbol{H}, \boldsymbol{b}) \in \mathbb{D}_5$; then, by the concavity of $f_{\mathrm{bce}}(\boldsymbol{W}, \boldsymbol{H}, \boldsymbol{b})$, it achieves its global minimum value on $\mathbb{D}_5$, which completes the proof associated with Lemma 15. $\qquad\square$

**Lemma 11.** *For $f_{\mathrm{bce}}(\boldsymbol{W}, \boldsymbol{H}, \boldsymbol{b})$ defined in Eq. (34),*

$$f_{\mathrm{bce}}(\boldsymbol{W}, \boldsymbol{H}, \boldsymbol{b}) \geq \log\left(1 + \exp\left(\sum_{i=1}^{n}\sum_{k=1}^{K}\frac{-\gamma\boldsymbol{w}_k^\top\boldsymbol{h}_i^{(k)} + b_k}{nK}\right)\right)$$

$$+ (K-1)\log\left(1 + \exp\left(\sum_{i=1}^{n}\sum_{k=1}^{K}\sum_{\substack{j=1\\j\neq k}}^{K}\frac{\gamma\boldsymbol{w}_j^\top\boldsymbol{h}_i^{(k)} - b_j}{nK(K-1)}\right)\right), \tag{42}$$

$$\triangleq f'_{\mathrm{bce}}(\boldsymbol{W}, \boldsymbol{H}, \boldsymbol{b}) \tag{43}$$

*and the inequality becomes an equality on the following point set*

$$\mathbb{D}_1 = \left\{(\boldsymbol{W}, \boldsymbol{H}, \boldsymbol{b}) : \gamma\boldsymbol{w}_k^\top\boldsymbol{h}_i^{(k)} - b_k = \gamma\boldsymbol{w}_{k'}^\top\boldsymbol{h}_{i'}^{(k')} - b_{k'}, \gamma\boldsymbol{w}_j^\top\boldsymbol{h}_i^{(k)} - b_j = \gamma\boldsymbol{w}_{j'}^\top\boldsymbol{h}_{i'}^{(k')} - b_{j'},\right.$$

$$\left.\forall k, k' \in [K], \ \forall j \neq k, j' \neq k' \in [K], \ \forall i, i' \in [n]\right\} \subset (\mathbb{S}^{d-1})^K \times (\mathbb{S}^{d-1})^{nK} \times \mathbb{R}^K. \tag{44}$$

*Proof.* By the concavities of $\log(1 + e^x)$,

$$f_{\mathrm{bce}}(\boldsymbol{W}, \boldsymbol{H}, \boldsymbol{b})$$

$$= \frac{1}{nK} \sum_{k=1}^{K} \sum_{i=1}^{n} \left[ \log\left(1 + \exp(-\gamma \boldsymbol{w}_k^\top \boldsymbol{h}_i^{(k)} + b_k)\right) + \sum_{\substack{j=1 \\ j \neq k}}^{K} \log\left(1 + \exp(\gamma \boldsymbol{w}_j^\top \boldsymbol{h}_i^{(k)} - b_j)\right) \right]$$

$$\geq \log\left(1 + \exp\left(\sum_{k=1}^{K} \sum_{i=1}^{n} \frac{-\gamma \boldsymbol{w}_k^\top \boldsymbol{h}_i^{(k)} + b_k}{nK}\right)\right) + \frac{1}{nK} \sum_{k=1}^{K} \sum_{i=1}^{n} \sum_{\substack{j=1 \\ j \neq k}}^{K} \log\left(1 + \exp(\gamma \boldsymbol{w}_j^\top \boldsymbol{h}_i^{(k)} - b_j)\right)$$

$$\tag{45}$$

$$\geq \log\left(1 + \exp\left(\sum_{i=1}^{n} \sum_{k=1}^{K} \frac{-\gamma \boldsymbol{w}_k^\top \boldsymbol{h}_i^{(k)} + b_k}{nK}\right)\right)$$

$$+ (K-1) \log\left(1 + \exp\left(\sum_{i=1}^{n} \sum_{k=1}^{K} \sum_{\substack{j=1 \\ j \neq k}}^{K} \frac{\gamma \boldsymbol{w}_j^\top \boldsymbol{h}_i^{(k)} - b_j}{nK(K-1)}\right)\right) = f_{\mathrm{bce}}'(\boldsymbol{W}, \boldsymbol{H}, \boldsymbol{b}), \tag{46}$$

while the first inequality becomes an equality if and only if

$$\gamma \boldsymbol{w}_k^\top \boldsymbol{h}_i^{(k)} - b_k = \gamma \boldsymbol{w}_{k'}^\top \boldsymbol{h}_{i'}^{(k')} - b_{k'}, \quad \forall k, k' \in [K], \forall i, i' \in [n], \tag{47}$$

and the second inequality becomes an equality if and only if

$$-\gamma \boldsymbol{w}_j^\top \boldsymbol{h}_i^{(k)} + b_k = -\gamma \boldsymbol{w}_\ell^\top \boldsymbol{h}_{i'}^{(k')} + b_\ell, \quad \forall j \neq k, j' \neq k' \in [K], \forall i, i' \in [n]. \tag{48}$$

When a point $(\boldsymbol{W}, \boldsymbol{H}, \boldsymbol{b})$ satisfies Eqs. (47) and (48), it must be contained in $\mathbb{D}_1$, which completes the proof. $\square$

**Lemma 12.** *On $\mathbb{D}_1$, the minimum points of $f_{\mathrm{bce}}'(\boldsymbol{W}, \boldsymbol{H}, \boldsymbol{b})$ are contained in*

$$\mathbb{D}_2 = \mathbb{D}_1 \bigcap \left\{ (\boldsymbol{W}, \boldsymbol{H}, \boldsymbol{b}) : b_k = \gamma \boldsymbol{w}_k^\top \boldsymbol{h}_i^{(j)} + \hat{b}_k, \forall k \in [K], j \neq k \right\}, \tag{49}$$

*where*

$$\hat{b}_k = \log \frac{(K-2) + \sqrt{(K-2)^2 + 4(K-1) \exp\left(\gamma \boldsymbol{w}_k^\top \boldsymbol{h}_i^{(k)} - \gamma \boldsymbol{w}_k^\top \boldsymbol{h}_i^{(j)}\right)}}{2}. \tag{50}$$

*Proof.* On $\mathbb{D}_1$,

$$f_{\mathrm{bce}}'(\boldsymbol{W}, \boldsymbol{H}, \boldsymbol{b}) = \log\left(1 + \exp\left(-\gamma \boldsymbol{w}_k^\top \boldsymbol{h}_i^{(k)} + b_k\right)\right)$$

$$+ (K-1) \log\left(1 + \exp\left(\gamma \boldsymbol{w}_k^\top \boldsymbol{h}_i^{(j)} - b_k\right)\right), \quad \forall j \neq k. \tag{51}$$

Then, at its minimum points,

$$0 = \frac{\partial f_{\mathrm{bce}}'}{\partial b_k} = 1 - \frac{1}{1 + \exp\left(-\gamma \boldsymbol{w}_k^\top \boldsymbol{h}_i^{(k)} + b_k\right)} - \frac{K-1}{1 + \exp\left(-\gamma \boldsymbol{w}_k^\top \boldsymbol{h}_i^{(j)} + b_k\right)}, \tag{52}$$

which leads to

$$b_k = \log \frac{(K-2) e^{\gamma \boldsymbol{w}_k^\top \boldsymbol{h}_i^{(j)}} \pm \sqrt{(K-2)^2 e^{2\gamma \boldsymbol{w}_k^\top \boldsymbol{h}_i^{(j)}} + 4(K-1) e^{\gamma \boldsymbol{w}_k^\top \boldsymbol{h}_i^{(k)} + \gamma \boldsymbol{w}_k^\top \boldsymbol{h}_i^{(j)}}}}{2} \tag{53}$$

$$= \gamma \boldsymbol{w}_k^\top \boldsymbol{h}_i^{(j)} + \log \frac{(K-2) + \sqrt{(K-2)^2 + 4(K-1) \exp\left(\gamma \boldsymbol{w}_k^\top \boldsymbol{h}_i^{(k)} - \gamma \boldsymbol{w}_k^\top \boldsymbol{h}_i^{(j)}\right)}}{2} \tag{54}$$

$$= \gamma \boldsymbol{w}_k^\top \boldsymbol{h}_i^{(j)} + \hat{b}_k. \tag{55}$$

Therefore, on $\mathbb{D}_1$, the minimum points of $f_{\mathrm{bce}}'(\boldsymbol{W}, \boldsymbol{H}, \boldsymbol{b})$ must be contained in $\mathbb{D}_2$. $\square$

**Lemma 13.** *On the point set of $\mathbb{D}_1$, for $\forall c_1, c_2 > 0$,*

$$f'_{\text{bce}}(\boldsymbol{W}, \boldsymbol{H}, \boldsymbol{b}) \geq f''_{\text{bce}}(\boldsymbol{W}, \boldsymbol{H}, \boldsymbol{b})$$

$$\triangleq -\frac{c_1}{1+c_1}\log(c_1) + \frac{1}{1+c_1}\sum_{i=1}^{n}\sum_{k=1}^{K}\frac{-\gamma\boldsymbol{w}_k^\top\boldsymbol{h}_i^{(k)} + b_k}{nK}$$

$$-(K-1)\frac{c_2}{1+c_2}\log(c_2) + \frac{1}{1+c_2}\sum_{i=1}^{n}\sum_{k=1}^{K}\sum_{\substack{j=1 \\ j\neq k}}^{K}\frac{\gamma\boldsymbol{w}_j^\top\boldsymbol{h}_i^{(k)} - b_j}{nK} + C, \quad (56)$$

*where*

$$C = \log(1+c_1) + (K-1)\log(1+c_2), \quad (57)$$

*and the inequality becomes an equality on point set of*

$$\mathbb{D}_3 = \Big\{ (\boldsymbol{W}, \boldsymbol{H}, \boldsymbol{b}; c_1, c_2) :$$

$$(\boldsymbol{W}, \boldsymbol{H}, \boldsymbol{b}) \in \mathbb{D}_1, \ c_1 = \boldsymbol{w}_k^\top\boldsymbol{h}_i^{(k)} - b_k, \ c_2 = -\gamma\boldsymbol{w}_j^\top\boldsymbol{h}_i^{(k)} + b_j \Big\}. \quad (58)$$

*Proof.* On $\mathbb{D}_1$, by the concavity of $\log(x)$,

$$f'_{\text{bce}}(\boldsymbol{W}, \boldsymbol{H}, \boldsymbol{b}) = \log\left(1 + \exp\left(\sum_{i=1}^{n}\sum_{k=1}^{K}\frac{-\gamma\boldsymbol{w}_k^\top\boldsymbol{h}_i^{(k)} + b_k}{nK}\right)\right)$$

$$+ (K-1)\log\left(1 + \exp\left(\sum_{i=1}^{n}\sum_{k=1}^{K}\sum_{\substack{j=1 \\ j\neq k}}^{K}\frac{\gamma\boldsymbol{w}_j^\top\boldsymbol{h}_i^{(k)} - b_j}{nK(K-1)}\right)\right)$$

$$= \log\left(\frac{c_1}{1+c_1}\frac{1+c_1}{c_1} + \frac{1+c_1}{1+c_1}\exp\left(\sum_{i=1}^{n}\sum_{k=1}^{K}\frac{-\gamma\boldsymbol{w}_k^\top\boldsymbol{h}_i^{(k)} + b_k}{nK}\right)\right)$$

$$+ (K-1)\log\left(\frac{c_2}{1+c_2}\frac{1+c_2}{c_2} + \frac{1+c_2}{1+c_2}\exp\left(\sum_{i=1}^{n}\sum_{k=1}^{K}\sum_{\substack{j=1 \\ j\neq k}}^{K}\frac{\gamma\boldsymbol{w}_j^\top\boldsymbol{h}_i^{(k)} - b_j}{nK(K-1)}\right)\right) \quad (59)$$

$$\geq \frac{c_1}{1+c_1}\log\left(\frac{1+c_1}{c_1}\right) + \frac{1}{1+c_1}\log\left((1+c_1)\exp\left(\sum_{i=1}^{n}\sum_{k=1}^{K}\frac{-\gamma\boldsymbol{w}_k^\top\boldsymbol{h}_i^{(k)} + b_k}{nK}\right)\right)$$

$$+ (K-1)\left[\frac{c_2}{1+c_2}\log\frac{1+c_2}{c_2} + \frac{1}{1+c_2}\log\left((1+c_2)\exp\left(\sum_{i=1}^{n}\sum_{k=1}^{K}\sum_{\substack{j=1 \\ j\neq k}}^{K}\frac{\gamma\boldsymbol{w}_j^\top\boldsymbol{h}_i^{(k)} - b_j}{nK(K-1)}\right)\right)\right]$$

$$(60)$$

$$= -\frac{c_1}{1+c_1}\log(c_1) + \frac{1}{1+c_1}\sum_{i=1}^{n}\sum_{k=1}^{K}\frac{-\gamma\boldsymbol{w}_k^\top\boldsymbol{h}_i^{(k)} + b_k}{nK}$$

$$-(K-1)\frac{c_2}{1+c_2}\log(c_2) + \frac{1}{1+c_2}\sum_{i=1}^{n}\sum_{k=1}^{K}\sum_{\substack{j=1 \\ j\neq k}}^{K}\frac{\gamma\boldsymbol{w}_j^\top\boldsymbol{h}_i^{(k)} - b_j}{nK}$$

$$+ \underbrace{\log(1+c_1) + (K-1)\log(1+c_2)}_{C}, \quad (61)$$

where the inequality becomes an equality if and only if

$$\frac{1+c_1}{c_1} = (1+c_1)\exp\left(\sum_{i=1}^{n}\sum_{k=1}^{K}\frac{-\gamma\boldsymbol{w}_k^\top\boldsymbol{h}_i^{(k)} + b_k}{nK}\right) \ \text{ or } \ c_1 = 0 \ \text{ or } \ c_1 = +\infty, \quad (62)$$

$$\frac{1+c_2}{c_2} = (1+c_2)\exp\left(\sum_{i=1}^{n}\sum_{k=1}^{K}\sum_{\substack{j=1 \\ j\neq k}}^{K}\frac{\gamma\boldsymbol{w}_j^\top\boldsymbol{h}_i^{(k)} - b_j}{nK(K-1)}\right) \ \text{ or } \ c_2 = 0 \ \text{ or } \ c_2 = +\infty. \quad (63)$$

The expressions are trivial when $c_1 = 0, c_1 = +\infty, c_2 = 0$, or $c_2 = +\infty$. Therefore, if and only if

$$c_1 = \exp\left(\sum_{i=1}^{n}\sum_{k=1}^{K}\frac{\gamma\boldsymbol{w}_k^\top\boldsymbol{h}_i^{(k)} - b_k}{nK}\right) = \exp\left(\gamma\boldsymbol{w}_k^\top\boldsymbol{h}_i^{(k)} - b_k\right), \tag{64}$$

$$c_2 = \exp\left(\sum_{i=1}^{n}\sum_{k=1}^{K}\sum_{\substack{j=1\\j\neq k}}^{K}\frac{-\gamma\boldsymbol{w}_j^\top\boldsymbol{h}_i^{(k)} + b_j}{nK(K-1)}\right) = \exp\left(-\gamma\boldsymbol{w}_j^\top\boldsymbol{h}_i^{(k)} + b_j\right), \tag{65}$$

the above inequality becomes an equality. The proof is finished. $\qquad\square$

**Lemma 14.** *On the point set $\mathbb{D}_1$, the function $f'_{\mathrm{bce}}(\boldsymbol{W}, \boldsymbol{H}, \boldsymbol{b})$ satisfies*

$$f'_{\mathrm{bce}}(\boldsymbol{W}, \boldsymbol{H}, \boldsymbol{b}) \geq -2\gamma - (K-2)b^\star + \log\left(1 + \mathrm{e}^{\gamma - b^\star}\right) + (K-1)\log\left(1 + \mathrm{e}^{b^\star + \frac{\gamma}{K-1}}\right), \tag{66}$$

*with*

$$b^\star = -\frac{\gamma}{K-1} + \hat{b}^\star = -\frac{\gamma}{K-1} + \log\frac{(K-2) + \sqrt{(K-2)^2 + 4(K-1)\mathrm{e}^{\gamma + \frac{\gamma}{K-1}}}}{2}, \tag{67}$$

*and $f'_{\mathrm{bce}}(\boldsymbol{W}, \boldsymbol{H}, \boldsymbol{b})$ reaches its lower boundary on the following point set*

$$\mathbb{D}_5 = \Big\{(\boldsymbol{W}, \boldsymbol{H}, \boldsymbol{b}) : \gamma\boldsymbol{w}_k^\top\boldsymbol{h}_i^{(k)} - b_k = \gamma\boldsymbol{w}_{k'}^\top\boldsymbol{h}_{i'}^{(k')} - b_{k'}, \ \gamma\boldsymbol{w}_j^\top\boldsymbol{h}_i^{(k)} - b_j = \gamma\boldsymbol{w}_{j'}^\top\boldsymbol{h}_{i'}^{(k')} - b_{j'},$$

$$b_k = \gamma\boldsymbol{w}_k^\top\boldsymbol{h}_i^{(j)} + \hat{b}_k, \ \boldsymbol{w}_k = \boldsymbol{h}_i^{(k)}, \ \sum_{k=1}^{K}\boldsymbol{w}_k = \boldsymbol{0},$$

$$\forall k, k' \in [K], \ \forall i, i' \in [n], \ \forall j \neq k, j' \neq k' \in [K]\Big\}. \tag{68}$$

*Proof.* According to Lemmas 12 and 13, we define

$$\mathbb{D}_4 = \Big\{(\boldsymbol{W}, \boldsymbol{H}, \boldsymbol{b}; c_1, c_2) : (\boldsymbol{W}, \boldsymbol{H}, \boldsymbol{b}) \in \mathbb{D}_2, \ c_1 = \boldsymbol{w}_k^\top\boldsymbol{h}_i^{(k)} - b_k, \ c_2 = -\gamma\boldsymbol{w}_j^\top\boldsymbol{h}_i^{(k)} + b_j\Big\}$$

$$= \Big\{(\boldsymbol{W}, \boldsymbol{H}, \boldsymbol{b}; c_1, c_2) : \gamma\boldsymbol{w}_k^\top\boldsymbol{h}_i^{(k)} - b_k = \gamma\boldsymbol{w}_{k'}^\top\boldsymbol{h}_{i'}^{(k')} - b_{k'}, \gamma\boldsymbol{w}_j^\top\boldsymbol{h}_i^{(k)} - b_j = \gamma\boldsymbol{w}_{j'}^\top\boldsymbol{h}_{i'}^{(k')} - b_{j'},$$

$$b_k = \gamma\boldsymbol{w}_k^\top\boldsymbol{h}_i^{(j)} + \hat{b}_k, \ c_1 = \boldsymbol{w}_k^\top\boldsymbol{h}_i^{(k)} - b_k, \ c_2 = -\gamma\boldsymbol{w}_j^\top\boldsymbol{h}_i^{(k)} + b_j,$$

$$\forall k, k' \in [K], \ \forall j \neq k, j' \neq k' \in [K], \ \forall i, i' \in [n]\Big\}. \tag{69}$$

On $\mathbb{D}_4$,

$$f'_{\mathrm{bce}}(\boldsymbol{W}, \boldsymbol{H}, \boldsymbol{b}) = f''_{\mathrm{bce}}(\boldsymbol{W}, \boldsymbol{H}, \boldsymbol{b})$$

$$= \frac{1}{nK}\sum_{k=1}^{K}\sum_{i=1}^{n}\left[-\gamma\boldsymbol{w}_k^\top\boldsymbol{h}_i^{(k)} + b_k + \sum_{\substack{j=1\\j\neq k}}^{K}\left(\gamma\boldsymbol{w}_j^\top\boldsymbol{h}_i^{(k)} - b_j\right)\right] + C \tag{70}$$

$$= \frac{1}{nK}\sum_{k=1}^{K}\sum_{i=1}^{n}\left(\sum_{\substack{j=1\\j\neq k}}^{K}\gamma\boldsymbol{w}_j^\top\boldsymbol{h}_i^{(k)} - \gamma\boldsymbol{w}_k^\top\boldsymbol{h}_i^{(k)}\right) + \frac{1}{nK}\sum_{k=1}^{K}\sum_{i=1}^{n}\left(\frac{1}{K-1}\sum_{\substack{j=1\\j\neq k}}^{K}b_k - \sum_{\substack{j=1\\j\neq k}}^{K}b_j\right) + C \tag{71}$$

$$= \frac{\gamma}{nK}\sum_{k=1}^{K}\sum_{i=1}^{n}\left(\sum_{j=1}^{K}\boldsymbol{w}_j^\top\boldsymbol{h}_i^{(k)} - 2\boldsymbol{w}_k^\top\boldsymbol{h}_i^{(k)}\right) + C$$

$$+ \frac{1}{nK}\sum_{k=1}^{K}\sum_{i=1}^{n}\left[\frac{1}{K-1}\sum_{\substack{j=1\\j\neq k}}^{K}\left(\gamma\boldsymbol{w}_k^\top\boldsymbol{h}_i^{(j)} + \hat{b}_k\right) - \sum_{\substack{j=1\\j\neq k}}^{K}\left(\gamma\boldsymbol{w}_j^\top\boldsymbol{h}_i^{(k)} + \hat{b}_j\right)\right] \tag{72}$$

$$= \frac{\gamma}{nK}\sum_{i=1}^{n}\left(\sum_{j=1}^{K}\sum_{k=1}^{K}\boldsymbol{w}_k^\top\boldsymbol{h}_i^{(j)} - 2\sum_{k=1}^{K}\boldsymbol{w}_k^\top\boldsymbol{h}_i^{(k)}\right) - \frac{K-2}{K}\sum_{k=1}^{K}\hat{b}_k + C$$

$$+ \frac{\gamma}{nK} \sum_{i=1}^{n} \left( \frac{1}{K-1} \sum_{k=1}^{K} \sum_{\substack{j=1 \\ j \neq k}}^{K} \boldsymbol{w}_k^\top \boldsymbol{h}_i^{(j)} - \sum_{k=1}^{K} \sum_{\substack{j=1 \\ j \neq k}}^{K} \boldsymbol{w}_j^\top \boldsymbol{h}_i^{(k)} \right) \tag{73}$$

$$= \frac{\gamma}{nK} \sum_{i=1}^{n} \left( \sum_{j=1}^{K} \sum_{k=1}^{K} \boldsymbol{w}_k^\top \boldsymbol{h}_i^{(j)} - 2 \sum_{k=1}^{K} \boldsymbol{w}_k^\top \boldsymbol{h}_i^{(k)} \right) - \frac{K-2}{K} \sum_{k=1}^{K} \hat{b}_k + C$$

$$- \frac{\gamma}{nK} \frac{K-2}{K-1} \sum_{i=1}^{n} \left( \sum_{k=1}^{K} \sum_{j=1}^{K} \boldsymbol{w}_k^\top \boldsymbol{h}_i^{(j)} - \sum_{k=1}^{K} \boldsymbol{w}_k^\top \boldsymbol{h}_i^{(k)} \right) \tag{74}$$

$$= \frac{\gamma}{nK} \sum_{i=1}^{n} \left( K \sum_{k=1}^{K} \boldsymbol{w}_k^\top \tilde{\boldsymbol{h}}_i - 2 \sum_{k=1}^{K} \boldsymbol{w}_k^\top \boldsymbol{h}_i^{(k)} \right) - \frac{K-2}{K} \sum_{k=1}^{K} \hat{b}_k + C$$

$$- \frac{\gamma}{nK} \frac{K-2}{K-1} \sum_{i=1}^{n} \left( K \sum_{k=1}^{K} \boldsymbol{w}_k^\top \tilde{\boldsymbol{h}}_i - \sum_{k=1}^{K} \boldsymbol{w}_k^\top \boldsymbol{h}_i^{(k)} \right) \tag{75}$$

$$= \frac{\gamma}{n(K-1)} \sum_{i=1}^{n} \sum_{k=1}^{K} \boldsymbol{w}_k^\top \left( \tilde{\boldsymbol{h}}_i - \boldsymbol{h}_i^{(k)} \right) - \frac{K-2}{K} \sum_{k=1}^{K} \hat{b}_k + C, \tag{76}$$

in the above expressions, $\tilde{\boldsymbol{h}}_i = \frac{1}{K} \sum_{j=1}^{K} \boldsymbol{h}_i^{(j)} = \frac{1}{K} \sum_{k=1}^{K} \boldsymbol{h}_i^{(k)}$. Furthermore, according to the AM-GM inequality (see Lemma 8),

$$\boldsymbol{u}^\top \boldsymbol{v} \geq - \left( \frac{c}{2} \|\boldsymbol{u}\|_2^2 + \frac{1}{2c} \|\boldsymbol{v}\|_2^2 \right), \ \forall \boldsymbol{u}, \boldsymbol{v} \in \mathbb{R}^d, \ \forall c > 0, \tag{77}$$

which becomes an equality when $\boldsymbol{v} = -c\boldsymbol{u}$. Then,

$$f'_{\text{bce}}(\boldsymbol{W}, \boldsymbol{H}, \boldsymbol{b}) \geq \frac{\gamma}{n(K-1)} \sum_{i=1}^{n} \sum_{k=1}^{K} \boldsymbol{w}_k^\top \left( \tilde{\boldsymbol{h}}_i - \boldsymbol{h}_i^{(k)} \right) - \frac{K-2}{K} \sum_{k=1}^{K} \hat{b}_k + C$$

$$\geq - \frac{\gamma}{n(K-1)} \sum_{i=1}^{n} \sum_{k=1}^{K} \left( \frac{c_3}{2} \|\boldsymbol{w}_k\|_2^2 + \frac{1}{2c_3} \left\| \tilde{\boldsymbol{h}}_i - \boldsymbol{h}_i^{(k)} \right\|_2^2 \right) - \frac{K-2}{K} \sum_{k=1}^{K} \hat{b}_k + C \tag{78}$$

$$= - \frac{\gamma}{n(K-1)} \sum_{i=1}^{n} \sum_{k=1}^{K} \left( \frac{c_3}{2} + \frac{1}{2c_3} \left( \|\tilde{\boldsymbol{h}}_i\|_2^2 - 2\tilde{\boldsymbol{h}}_i^\top \boldsymbol{h}_i^{(k)} + 1 \right) \right) - \frac{K-2}{K} \sum_{k=1}^{K} \hat{b}_k + C \tag{79}$$

$$= - \frac{\gamma K}{K-1} \left( \frac{c_3}{2} + \frac{1}{2c_3} \right) + \frac{\gamma K}{n(K-1)} \frac{1}{2c_3} \sum_{i=1}^{n} \|\tilde{\boldsymbol{h}}_i\|_2^2 - \frac{K-2}{K} \sum_{k=1}^{K} \hat{b}_k + C \tag{80}$$

$$\geq - \frac{\gamma K}{K-1} \left( \frac{c_3}{2} + \frac{1}{2c_3} \right) - \frac{K-2}{K} \sum_{k=1}^{K} \hat{b}_k + C. \tag{81}$$

According to AM-GM inequality, the inequality (78) becomes an equality if and only if

$$- c_3 \boldsymbol{w}_k = \tilde{\boldsymbol{h}}_i - \boldsymbol{h}_i^{(k)}, \quad \forall k \in [K], \ i \in [n], \ \forall c_3 > 0. \tag{82}$$

The inequality (81) becomes an equality if and only if

$$\tilde{\boldsymbol{h}}_i = \boldsymbol{0}, \quad \forall i \in [n]. \tag{83}$$

Combining Eqs. (82) and (83), one can get

$$c_3 \boldsymbol{w}_k = \boldsymbol{h}_i^{(k)} \Rightarrow \|c_3 \boldsymbol{w}_k\|_2^2 = \|\boldsymbol{h}_i^{(k)}\|_2^2 \Rightarrow c_3^2 = 1 \Rightarrow c_3 = 1, \tag{84}$$

$$\boldsymbol{w}_k = \boldsymbol{h}_i^{(k)}, \quad \forall k \in [K], \ i \in [n], \tag{85}$$

$$\text{and} \quad \sum_{k=1}^{K} \boldsymbol{w}_k = \sum_{i=1}^{K} \boldsymbol{h}_i^{(k)} = K \tilde{\boldsymbol{h}}_i = \boldsymbol{0}. \tag{86}$$

Furthermore, according to Lemma 15, on $\mathbb{D}_5 = \mathbb{D}_5 \bigcap \left\{ \boldsymbol{w}_k = \boldsymbol{h}_i^{(k)}, \sum_{k=1}^K \boldsymbol{w}_k = \boldsymbol{0} \right\}$, one can get $b_k = b^\star$, $\forall k \in [K]$, and

$$c_1 = \mathrm{e}^{\gamma - b^\star}, \; c_2 = \mathrm{e}^{b^\star + \frac{\gamma}{K-1}}, \; b^\star = -\frac{\gamma}{K-1} + \hat{b}^\star, \tag{87}$$

$$\text{with} \qquad \hat{b}^\star = \log \frac{(K-2) + \sqrt{(K-2)^2 + 4(K-1)\mathrm{e}^{\gamma + \frac{\gamma}{K-1}}}}{2}. \tag{88}$$

Then, on $\mathbb{D}_4$,

$$f_{\mathrm{bce}}(\boldsymbol{W}, \boldsymbol{H}, \boldsymbol{b}) \geq -\frac{\gamma K}{K-1} - \frac{K-2}{K} \sum_{k=1}^K \hat{b}_k + C$$

$$= -\frac{\gamma K}{K-1} - (K-2)\hat{b}^\star + \log\left(1 + \mathrm{e}^{\gamma - b^\star}\right) + (K-1)\log\left(1 + \mathrm{e}^{b^\star + \frac{\gamma}{K-1}}\right) \tag{89}$$

$$\overset{(87)}{=} -2\gamma - (K-2)b^\star + \log\left(1 + \mathrm{e}^{\gamma - b^\star}\right) + (K-1)\log\left(1 + \mathrm{e}^{b^\star + \frac{\gamma}{K-1}}\right), \tag{90}$$

and the equality is achieved only on $\mathbb{D}_5$.

According to the above derivation, the points in $\mathbb{D}_5$ are minimum points of $f'_{\mathrm{bce}}(\boldsymbol{W}, \boldsymbol{H}, \boldsymbol{b})$ on the set $\mathbb{D}_4$, i.e., for any $(\boldsymbol{W}^\star, \boldsymbol{H}^\star, \boldsymbol{b}^\star) \in \mathbb{D}_5$,

$$f'_{\mathrm{bce}}(\boldsymbol{W}^\star, \boldsymbol{H}^\star, \boldsymbol{b}^\star) < f'_{\mathrm{bce}}(\boldsymbol{W}, \boldsymbol{H}, \boldsymbol{b}), \quad \forall (\boldsymbol{W}, \boldsymbol{H}, \boldsymbol{b}) \in \mathbb{D}_4 - \mathbb{D}_5. \tag{91}$$

By the concavity of $f'_{\mathrm{bce}}(\boldsymbol{W}, \boldsymbol{H}, \boldsymbol{b})$, it achieves its global minimum value on $\mathbb{D}_1$, which completes the proof. $\qquad \square$

**Lemma 15.** *On the point set of $\mathbb{D}_5$ defined in Eq. (68),*

$$\boldsymbol{w}_k = \boldsymbol{h}_i^{(k)}, \; \forall k \in [K], \; i \in [n], \tag{92}$$

$$\tilde{\boldsymbol{h}}_i := \frac{1}{K} \sum_{k=1}^K \boldsymbol{h}_i^{(k)} = \boldsymbol{0}, \; \forall i \in [n], \tag{93}$$

$$\boldsymbol{W}\boldsymbol{W}^\top = \frac{K}{K-1}\left(\boldsymbol{I}_K - \frac{1}{K}\boldsymbol{1}_K\boldsymbol{1}_K^\top\right), \tag{94}$$

$$\boldsymbol{b}^\star = b^\star \boldsymbol{1}, \tag{95}$$

$$c_1 = \exp\left(\gamma - b^\star\right), \quad \text{and} \quad c_2 = \exp\left(b^\star + \frac{\gamma}{K-1}\right), \tag{96}$$

*where $b^\star$ satisfies*

$$b^\star = \log \frac{(K-2)\mathrm{e}^{-\frac{\gamma}{K-1}} + \sqrt{(K-2)^2\mathrm{e}^{-\frac{2\gamma}{K-1}} + 4(K-1)\mathrm{e}^{-\frac{\gamma}{K-1} + \gamma}}}{2}$$

$$= -\frac{\gamma}{K-1} + \underbrace{\log \frac{(K-2) + \sqrt{(K-2)^2 + 4(K-1)\mathrm{e}^{\gamma + \frac{\gamma}{K-1}}}}{2}}_{\hat{b}^\star}. \tag{97}$$

*Proof.* In Eq. (68),

$$\mathbb{D}_5 = \Big\{ (\boldsymbol{W}, \boldsymbol{H}, \boldsymbol{b}) : \gamma \boldsymbol{w}_k^\top \boldsymbol{h}_i^{(k)} - b_k = \gamma \boldsymbol{w}_j^\top \boldsymbol{h}_{i'}^{(j)} - b_j, \; \gamma \boldsymbol{w}_j^\top \boldsymbol{h}_i^{(k)} - b_j = \gamma \boldsymbol{w}_\ell^\top \boldsymbol{h}_i^{(k)} - b_\ell,$$

$$b_k = \gamma \boldsymbol{w}_k^\top \boldsymbol{h}_i^{(j)} + \hat{b}_k, \; \boldsymbol{w}_k = \boldsymbol{h}_i^{(k)}, \; \sum_{k=1}^K \boldsymbol{w}_k = \boldsymbol{0}, \; \forall i, i' \in [n], \; \forall j, \ell \neq k \in [K] \Big\}.$$

Therefore, on $\mathbb{D}_5$,

$$\tilde{\boldsymbol{h}}_i = \frac{1}{K}\sum_{k=1}^{K}\boldsymbol{h}_i^{(k)} = \frac{1}{K}\sum_{k=1}^{K}\boldsymbol{w}_k = \boldsymbol{0}, \tag{98}$$

$$c_1 = \exp\left(\gamma\boldsymbol{w}_k^\top\boldsymbol{h}_i^{(k)} - b_k\right) = \exp\left(\gamma\|\boldsymbol{w}_k\|_2^2 - b_k\right), \ \forall k \in [K], \tag{99}$$

$$c_2 = \exp\left(b_j - \gamma\boldsymbol{w}_j^\top\boldsymbol{h}_i^{(k)}\right) = \exp\left(b_j - \gamma\boldsymbol{w}_k^\top\boldsymbol{w}_j\right), \ \forall j \neq k \in [K]. \tag{100}$$

On $\mathbb{D}_5$, the $c_1, c_2 > 0$ are the same for all $j \neq k \in [K]$. Then,

$$\gamma\|\boldsymbol{w}_k\|_2^2 - b_k = \gamma\|\boldsymbol{w}_j\|_2^2 - b_j, \ \forall k,j \in [K], \tag{101}$$

$$\gamma\boldsymbol{w}_k^\top\boldsymbol{w}_j - b_j = \gamma\boldsymbol{w}_k^\top\boldsymbol{w}_\ell - b_\ell, \ \forall j \neq \ell \in [K], \ \forall k \in [K], \tag{102}$$

In the normalized feature space, *Eq.* (101) implies

$$\gamma\|\boldsymbol{w}_k\|_2^2 - b_k = \gamma\|\boldsymbol{w}_j\|_2^2 - b_j \Rightarrow \gamma - b_k = \gamma - b_j \Leftrightarrow b_k = b_j, \ \forall k,j \in [K], \tag{103}$$

Therefore, we can write $\boldsymbol{b} = b\boldsymbol{1}$ for some $b \in \mathbb{R}$. Then, from Eq. (102),

$$\gamma\boldsymbol{w}_k^\top\boldsymbol{w}_j = \gamma\boldsymbol{w}_k^\top\boldsymbol{w}_\ell \Rightarrow \boldsymbol{w}_k^\top\boldsymbol{w}_j = \boldsymbol{w}_k^\top\boldsymbol{w}_\ell, \ \forall j \neq \ell \in [K], \ \forall k \in [K], \tag{104}$$

and combining with *Eq.* (98),

$$1 = \|\boldsymbol{w}_k\|_2^2 = -\sum_{\substack{j=1 \\ j \neq k}}^{K}\boldsymbol{w}_k^\top\boldsymbol{w}_j = -(K-1)\boldsymbol{w}_k^\top\boldsymbol{w}_j \tag{105}$$

$$\Rightarrow \boldsymbol{w}_k^\top\boldsymbol{w}_j = -\frac{1}{K-1}, \ \forall j \neq k \in [K]. \tag{106}$$

Therefore, we get

$$\boldsymbol{W}\boldsymbol{W}^\top = \frac{K}{K-1}\left(\boldsymbol{I}_K - \frac{1}{K}\boldsymbol{1}_K\boldsymbol{1}_K^\top\right). \tag{107}$$

Plugging *Eqs.* (103), (106) into *Eqs.* (99), (100), we have

$$c_1 = \exp\left(\gamma - b^\star\right) \quad \text{and} \quad c_2 = \exp\left(b^\star + \frac{\gamma}{K-1}\right). \tag{108}$$

where $b_k = b_j = b^\star$, for $\forall k \neq j \in [K]$.

We can derivative that $b^\star$ should hold

$$b^\star \overset{(50)}{=} \gamma\boldsymbol{w}_k^\top\boldsymbol{h}_i^{(j)} + \log\frac{(K-2) + \sqrt{(K-2)^2 + 4(K-1)\mathrm{e}^{\gamma\boldsymbol{w}_k^\top\boldsymbol{h}_i^{(k)} - \gamma\boldsymbol{w}_k^\top\boldsymbol{h}_i^{(j)}}}}{2}$$

$$= -\frac{\gamma}{K-1} + \log\frac{(K-2) + \sqrt{(K-2)^2 + 4(K-1)\mathrm{e}^{\gamma + \frac{\gamma}{K-1}}}}{2} \tag{109}$$

$$= -\frac{\gamma}{K-1} + \hat{b}^\star. \tag{110}$$

as desired. $\square$

**Lemma 16.** *When the classes number $K > 2$, the scale factor $\gamma > 0$, and*

$$\log(2K-3)(1 - \frac{1}{K}) < \gamma, \tag{111}$$

*the final critical bias $\boldsymbol{b}^\star$ could separate the all positive unbiased decision scores*

$$\left\{\gamma\boldsymbol{w}_k^{\star\top}\boldsymbol{h}_i^{(k)\star} : \ k \in [K], \ i \in [n]\right\}, \tag{112}$$

*and the all negative ones*

$$\left\{\gamma\boldsymbol{w}_j^{\star\top}\boldsymbol{h}_i^{(k)\star} : \ k,j \in [K], \ i \in [n], \ k \neq j\right\}. \tag{113}$$

*Proof.* According to Lemma 15, when the function $f_{\text{bce}}$ achieves the lower bound, we have

$$\gamma \boldsymbol{w}_k^{\star\top} \boldsymbol{h}_i^{(k)\star} = \gamma, \quad \forall k \in [K], \ i \in [n], \tag{114}$$

$$\gamma \boldsymbol{w}_j^{\star\top} \boldsymbol{h}_i^{(k)\star} = -\frac{\gamma}{K-1}, \quad \forall j \neq k \in [K], \ i \in [n], \tag{115}$$

Let $b_{\text{pos}} = \gamma, b_{\text{neg}} = -\frac{\gamma}{K-1}$. Then the critical $b^\star$ separating the all positive and negative decision scores means

$$b_{\text{neg}} = -\frac{\gamma}{K-1} < b^\star < \gamma = b_{\text{pos}}. \tag{116}$$

Due to

$$-\frac{\gamma}{K-1} < b^\star \Leftrightarrow 2\,\mathrm{e}^{-\frac{\gamma}{K-1}} < (K-2)\,\mathrm{e}^{-\frac{\gamma}{K-1}} + \sqrt{(K-2)^2\,\mathrm{e}^{-\frac{2\gamma}{K-1}} + 4\,(K-1)\,\mathrm{e}^{-\frac{\gamma}{K-1}+\gamma}} \tag{117}$$

$$\Leftarrow 2\,\mathrm{e}^{-\frac{\gamma}{K-1}} < (K-2)\,\mathrm{e}^{-\frac{\gamma}{K-1}} + \sqrt{4\,(K-1)\,\mathrm{e}^{\left(-\frac{\gamma}{K-1}+\gamma\right)}} \tag{118}$$

$$\Leftrightarrow 2 < (K-2) + 2\sqrt{K-1}\mathrm{e}^{\left(\frac{\gamma}{2(K-1)}+\frac{\gamma}{2}\right)} \tag{119}$$

$$\Leftarrow 2 < (K-2) + 2\sqrt{K-1} \tag{120}$$

$$\Leftarrow 2 < K, \tag{121}$$

$$b^\star < \gamma \Leftrightarrow (K-2)\,\mathrm{e}^{-\frac{\gamma}{K-1}} + \sqrt{(K-2)^2\,\mathrm{e}^{-\frac{2\gamma}{K-1}} + 4\,(K-1)\,\mathrm{e}^{-\frac{\gamma}{K-1}+\gamma}} < 2\mathrm{e}^\gamma \tag{122}$$

$$\Leftrightarrow (K-1)\,\mathrm{e}^{-\frac{\gamma}{K-1}} < \mathrm{e}^\gamma - (K-2)\,\mathrm{e}^{-\frac{\gamma}{K-1}} \tag{123}$$

$$\Leftrightarrow \log\,(2K-3)\,(1-\frac{1}{K}) < \gamma \tag{124}$$

$$\Leftarrow \log\,(2K-3) < \gamma \tag{125}$$

$$\Leftrightarrow K < \frac{\mathrm{e}^\gamma + 3}{2}, \tag{126}$$

which completes the proof. $\qquad\square$

### E.3 PROOF OF THEOREM 1

**Theorem 17.** *Assume that the feature dimension $d$ is larger than the number of classes $K$, i.e., $d \geq K-1$, and the scale factor $\gamma > 0$. In the normalized feature space, any global minimizer $(\boldsymbol{W}^\star, \boldsymbol{H}^\star, \boldsymbol{b}^\star)$ of*

$$\min_{\boldsymbol{W},\boldsymbol{H},\boldsymbol{b}} f_{\text{ce}}(\boldsymbol{W},\boldsymbol{H},\boldsymbol{b}) := \frac{1}{nK}\sum_{k=1}^{K}\sum_{i=1}^{n}\mathcal{L}_{\text{ce}}(\gamma\boldsymbol{W}\boldsymbol{h}_i^{(k)} - \boldsymbol{b}) \tag{127}$$

*with $\|\boldsymbol{w}_j\| = \|\boldsymbol{h}_i^{(k)}\| = 1$, obeys the following*

$$\boldsymbol{w}_k^\star = \boldsymbol{h}_i^{(k)\star}, \quad \forall k \in [K], i \in [n], \tag{128}$$

$$\tilde{\boldsymbol{h}}_i^\star := \frac{1}{K}\sum_{j=1}^{K}\boldsymbol{h}_i^{(k)\star} = \boldsymbol{0}, \quad \forall i \in [n], \tag{129}$$

$$\boldsymbol{W}^\star\boldsymbol{W}^{\star\top} = \frac{K}{K-1}\left(\boldsymbol{I}_K - \frac{1}{K}\boldsymbol{1}_K\boldsymbol{1}_K^\top\right), \tag{130}$$

$$\boldsymbol{b}^\star = b\boldsymbol{1}, \quad \forall b \in \mathbb{R}. \tag{131}$$

*Proof.* According to Lemma 19, for any fixed $\gamma > 0$, we have

$$f_{\text{ce}}(\boldsymbol{W},\boldsymbol{H},\boldsymbol{b}) \geq \log\left(1 + (K-1)\mathrm{e}^{-\frac{\gamma K}{K-1}}\right). \tag{132}$$

According to Lemma 20, the inequality (132) achieves its equality when *Eqs.* (128, 129, 130, 131) hold, which finishes the proof. $\qquad\square$

**Lemma 18.** *For any $\boldsymbol{h}_i^{(k)}$ with $c_1' > 0$ and $\gamma > 0$, the normalized CE loss is lower bounded by*

$$\mathcal{L}_{\text{ce}}(\gamma \boldsymbol{W} \boldsymbol{h}_i^{(k)} - \boldsymbol{b}) \geq \frac{\sum_{j=1}^K (\gamma \boldsymbol{w}_j^\top \boldsymbol{h}_i^{(k)} - b_j) - K(\gamma \boldsymbol{w}_k^\top \boldsymbol{h}_i^{(k)} - b_k)}{(1 + c_1')(K - 1)} + C', \tag{133}$$

*where*

$$C' = \frac{c_1'}{1 + c_1'} \log\left(\frac{1 + c_1'}{c_1'}\right) + \frac{1}{1 + c_1'} \log\left[(1 + c_1')(K - 1)\right]. \tag{134}$$

*The inequality becomes an equality when*

$$c_1' = \left[(K - 1)\exp\left(\frac{\sum_{j=1}^K (\gamma \boldsymbol{w}_j^\top \boldsymbol{h}_i^{(k)} - b_j) - K(\gamma \boldsymbol{w}_k^\top \boldsymbol{h}_i^{(k)} - b_k)}{K - 1}\right)\right]^{-1}, \tag{135}$$

*and*

$$\gamma \boldsymbol{w}_j^\top \boldsymbol{h}_i^{(k)} - b_j = \gamma \boldsymbol{w}_\ell^\top \boldsymbol{h}_i^{(k)} - b_\ell, \ \forall j, \ell \neq k \in [K]. \tag{136}$$

*Proof.* According to Jensen's inequality in Lemma 9, the normalized CE loss can be lower bounded,

$$\mathcal{L}_{\text{ce}}(\gamma \boldsymbol{W} \boldsymbol{h}_i^{(k)} - \boldsymbol{b}) = \log\left(1 + \sum_{\substack{j=1 \\ j \neq k}}^K \frac{\exp\left(\gamma \boldsymbol{w}_j^\top \boldsymbol{h}_i^{(k)} - b_j\right)}{\exp\left(\gamma \boldsymbol{w}_k^\top \boldsymbol{h}_i^{(k)} - b_k\right)}\right) \tag{137}$$

$$= \log\left(1 + (K - 1)\sum_{\substack{j=1 \\ j \neq k}}^K \frac{\exp\left(\gamma \boldsymbol{w}_j^\top \boldsymbol{h}_i^{(k)} - b_j\right)}{(K - 1)\exp\left(\gamma \boldsymbol{w}_k^\top \boldsymbol{h}_i^{(k)} - b_k\right)}\right) \tag{138}$$

$$\geq \log\left(1 + (K - 1)\exp\left(\sum_{\substack{j=1 \\ j \neq k}}^K \frac{\gamma \boldsymbol{w}_j^\top \boldsymbol{h}_i^{(k)} - b_j - \gamma \boldsymbol{w}_k^\top \boldsymbol{h}_i^{(k)} + b_k}{K - 1}\right)\right) \tag{139}$$

$$= \log\left(1 + (K - 1)\exp\left(\frac{\sum_{j=1}^K (\gamma \boldsymbol{w}_j^\top \boldsymbol{h}_i^{(k)} - b_j) - K(\gamma \boldsymbol{w}_k^\top \boldsymbol{h}_i^{(k)} - b_k)}{K - 1}\right)\right), \tag{140}$$

which achieves the equality if and only if $\gamma \boldsymbol{w}_j^\top \boldsymbol{h}_i^{(k)} - b_j = \gamma \boldsymbol{w}_\ell^\top \boldsymbol{h}_i^{(k)} - b_\ell$ for all $j, \ell \neq k \in [K]$.

Second, by the concavity of the function $\log(1 + e^x)$, for any $c_1' > 0$, we get

$$\mathcal{L}_{\text{ce}}(\gamma \boldsymbol{W} \boldsymbol{h}_i^{(k)} - \boldsymbol{b})$$

$$\geq \log\left(1 + (K - 1)\exp\left(\frac{\sum_{j=1}^K (\gamma \boldsymbol{w}_j^\top \boldsymbol{h}_i^{(k)} - b_j) - K(\gamma \boldsymbol{w}_k^\top \boldsymbol{h}_i^{(k)} - b_k)}{K - 1}\right)\right) \tag{141}$$

$$= \log\left(\frac{c_1'}{1 + c_1'}\frac{1 + c_1'}{c_1'} + \frac{1 + c_1'}{1 + c_1'}(K - 1)\exp\left(\frac{\sum_{j=1}^K (\gamma \boldsymbol{w}_j^\top \boldsymbol{h}_i^{(k)} - b_j) - K(\gamma \boldsymbol{w}_k^\top \boldsymbol{h}_i^{(k)} - b_k)}{K - 1}\right)\right) \tag{142}$$

$$\geq \frac{c_1'}{1 + c_1'} \log\left(\frac{1 + c_1'}{c_1'}\right)$$

$$+ \frac{1}{1 + c_1'} \log\left((1 + c_1')(K - 1)\exp\left(\frac{\sum_{j=1}^K (\gamma \boldsymbol{w}_j^\top \boldsymbol{h}_i^{(k)} - b_j) - K(\gamma \boldsymbol{w}_k^\top \boldsymbol{h}_i^{(k)} - b_k)}{K - 1}\right)\right) \tag{143}$$

$$= \frac{1}{1 + c_1'}\frac{\sum_{j=1}^K (\gamma \boldsymbol{w}_j^\top \boldsymbol{h}_i^{(k)} - b_j) - K(\gamma \boldsymbol{w}_k^\top \boldsymbol{h}_i^{(k)} - b_k)}{K - 1}$$

$$+ \underbrace{\frac{c_1'}{1 + c_1'} \log\left(\frac{1 + c_1'}{c_1'}\right) + \frac{1}{1 + c_1'} \log\left[(1 + c_1)(K - 1)\right]}_{C'}. \tag{144}$$

The last inequality achieves its equality if and only if

$$\frac{1 + c_1'}{c_1'} = (1 + c_1')(K - 1) \exp \left( \frac{\sum_{j=1}^{K} (\gamma \boldsymbol{w}_j^\top \boldsymbol{h}_i^{(k)} - b_j) - K(\gamma \boldsymbol{w}_k^\top \boldsymbol{h}_i^{(k)} - b_k)}{K - 1} \right), \quad (145)$$

$$\text{or } c_1' = 0 \text{ or } c_1' = +\infty. \quad (146)$$

Actually, when $c_1' = 0$ or $c_1' = +\infty$, the equality is trivial. Therefore, we have

$$c_1' = \left[ (K - 1) \exp \left( \frac{\sum_{j=1}^{K} (\gamma \boldsymbol{w}_j^\top \boldsymbol{h}_i^{(k)} - b_j) - K(\gamma \boldsymbol{w}_k^\top \boldsymbol{h}_i^{(k)} - b_k)}{K - 1} \right) \right]^{-1}, \quad (147)$$

as desired. $\qquad\square$

**Lemma 19.** *For the function $f_{\mathrm{ce}}(\boldsymbol{W}, \boldsymbol{H}, \boldsymbol{b})$ defined in Eq. (127),*

$$f_{\mathrm{ce}}(\boldsymbol{W}, \boldsymbol{H}, \boldsymbol{b}) \geq \log \left( 1 + (K - 1) \mathrm{e}^{-\frac{\gamma K}{K-1}} \right). \quad (148)$$

*Proof.* According to Lemma 18, *Eq.* (133) holds for any $c_1' > 0$ and any $\boldsymbol{h}_i^{(k)}$ with $k \in [K]$, $i \in [n]$. We take the same $c_1'$ for all $\boldsymbol{h}_i^{(k)}$, then we have the following lower bound for the function $f_{\mathrm{ce}}$ as

$$(1 + c_1')(K - 1) [f_{\mathrm{ce}}(\boldsymbol{W}, \boldsymbol{H}, \boldsymbol{b}) - C] \quad (149)$$

$$= (1 + c_1')(K - 1) \left[ \frac{1}{nK} \sum_{k=1}^{K} \sum_{i=1}^{n} \mathcal{L}_{\mathrm{ce}}(\gamma \boldsymbol{W} \boldsymbol{h}_i^{(k)} - \boldsymbol{b}) - C \right] \quad (150)$$

$$\geq \frac{1}{nK} \sum_{k=1}^{K} \sum_{i=1}^{n} \left[ \sum_{j=1}^{K} \left( \gamma \boldsymbol{w}_j^\top \boldsymbol{h}_i^{(k)} - b_j \right) - K \left( \gamma \boldsymbol{w}_k^\top \boldsymbol{h}_i^{(k)} - b_k \right) \right] \quad (151)$$

$$= \frac{1}{nK} \sum_{i=1}^{n} \left[ \left( \sum_{k=1}^{K} \sum_{j=1}^{K} \gamma \boldsymbol{w}_j^\top \boldsymbol{h}_i^{(k)} - K \sum_{k=1}^{K} \gamma \boldsymbol{w}_k^\top \boldsymbol{h}_i^{(k)} \right) + \underbrace{\sum_{k=1}^{K} \sum_{j=1}^{K} (b_k - b_j)}_{0} \right] \quad (152)$$

$$= \frac{1}{nK} \sum_{i=1}^{n} \left( \sum_{k=1}^{K} \sum_{j=1}^{K} \gamma \boldsymbol{w}_k^\top \boldsymbol{h}_i^{(j)} - K \sum_{k=1}^{K} \gamma \boldsymbol{w}_k^\top \boldsymbol{h}_i^{(k)} \right) \quad (153)$$

$$= \frac{1}{n} \sum_{i=1}^{n} \sum_{k=1}^{K} \left[ \gamma \boldsymbol{w}_k^\top \left( \frac{1}{K} \sum_{j=1}^{K} \boldsymbol{h}_i^{(j)} - \boldsymbol{h}_i^{(k)} \right) \right] \quad (154)$$

$$= \frac{1}{n} \sum_{i=1}^{n} \sum_{k=1}^{K} \gamma \boldsymbol{w}_k^\top \left( \tilde{\boldsymbol{h}}_i - \boldsymbol{h}_i^{(k)} \right), \quad (155)$$

where $\tilde{\boldsymbol{h}}_i = \frac{1}{K} \sum_{j=1}^{K} \boldsymbol{h}_i^{(j)}$.

Then, according to the AM-GM inequality (see Lemma 8), we have

$$\boldsymbol{u}^\top \boldsymbol{v} \leq \frac{c}{2} \|\boldsymbol{u}\|_2^2 + \frac{1}{2c} \|\boldsymbol{v}\|_2^2, \ \forall \boldsymbol{u}, \boldsymbol{v} \in \mathbb{R}^d, \ \forall c > 0, \quad (156)$$

which becomes an equality when $\boldsymbol{v} = c\boldsymbol{u}$. Then, based on *Eq.* (155), we get

$$(1 + c_1')(K - 1) [f_{\mathrm{ce}}(\boldsymbol{W}, \boldsymbol{H}, \boldsymbol{b}) - C']$$

$$\geq \frac{1}{n} \sum_{i=1}^{n} \sum_{k=1}^{K} \gamma \boldsymbol{w}_k^\top \left( \tilde{\boldsymbol{h}}_i - \boldsymbol{h}_i^{(k)} \right) \quad (157)$$

$$\geq -\frac{c_2'}{2} \sum_{k=1}^{K} \|\gamma \boldsymbol{w}_k\|_2^2 - \frac{1}{2c_2' n} \sum_{i=1}^{n} \sum_{k=1}^{K} \|\tilde{\boldsymbol{h}}_i - \boldsymbol{h}_i^{(k)}\|_2^2 \quad (158)$$

$$= -\frac{c_2'}{2} \sum_{k=1}^{K} \gamma^2 \|\boldsymbol{w}_k\|_2^2 - \frac{1}{2c_2'n} \sum_{i=1}^{n} \left[ \sum_{k=1}^{K} \|\boldsymbol{h}_i^{(k)}\|_2^2 - K\|\tilde{\boldsymbol{h}}_i\|_2^2 \right] \tag{159}$$

$$= -\frac{c_2'}{2} \gamma^2 K - \frac{1}{2c_2'n} \left( nK - K \sum_{i=1}^{n} \|\tilde{\boldsymbol{h}}_i\|_2^2 \right), \tag{160}$$

where the second inequality becomes an equality if and only if

$$c_2'\gamma \boldsymbol{w}_k = \boldsymbol{h}_i^{(k)} - \tilde{\boldsymbol{h}}_i, \ \forall k \in [K], \ i \in [n], \ \gamma > 0. \tag{161}$$

Therefore,

$$f_{\text{ce}}(\boldsymbol{W}, \boldsymbol{H}, \boldsymbol{b})$$

$$\geq -\frac{c_2'\gamma^2 K}{2(1+c_1')(K-1)} - \frac{K}{2c_2'n(1+c_1')(K-1)} \left( n - \sum_{i=1}^{n} \|\tilde{\boldsymbol{h}}_i\|_2^2 \right) + C' \tag{162}$$

$$\geq -\frac{c_2'\gamma^2 K}{2(1+c_1')(K-1)} - \frac{K}{2c_2'(1+c_1')(K-1)} + C' \tag{163}$$

$$= -\frac{K}{2(1+c_1')(K-1)} \left( c_2'\gamma^2 + \frac{1}{c_2'} \right) + C', \tag{164}$$

where the second inequality becomes an equality if and only if

$$\tilde{\boldsymbol{h}}_i = \frac{1}{K} \sum_{j=1}^{K} \boldsymbol{h}_i^{(j)} = \boldsymbol{0}, \ \forall i \in [n]. \tag{165}$$

Based on *Eqs.* (161) and (165), we get

$$c_2'\gamma \boldsymbol{w}_k = \boldsymbol{h}_i^{(k)} \Rightarrow \|c_2'\gamma \boldsymbol{w}_k\|_2^2 = \|\boldsymbol{h}_i^{(k)}\|_2^2 \Rightarrow (c_2'\gamma)^2 = 1 \Rightarrow c_2' = \frac{1}{\gamma}. \tag{166}$$

Plugging *Eq.* (166) into (164), we have

$$f_{\text{ce}}(\boldsymbol{W}, \boldsymbol{H}, \boldsymbol{b}) \geq -\frac{K(c_2'\gamma^2 + \frac{1}{c_2'})}{2(1+c_1')(K-1)} + C' = -\frac{\gamma K}{(1+c_1')(K-1)} + C'. \tag{167}$$

Furthermore, according to Lemma 20, when the objective function $f_{\text{ce}}$ achieves its lower bound, we have $\frac{\partial f_{\text{ce}}}{\partial b_k} \equiv 0$, and $c_1'$ satisfies

$$c_1' = \left[ (K-1) \exp\left( -\frac{\gamma K}{K-1} \right) \right]^{-1}. \tag{168}$$

Then, we have

$$f_{\text{ce}}(\boldsymbol{W}, \boldsymbol{H}, \boldsymbol{b}) \geq -\frac{\gamma K}{(1+c_1')(K-1)} + C' = \log\left( 1 + (K-1)\mathrm{e}^{-\frac{\gamma K}{K-1}} \right), \tag{169}$$

as suggested in *Eq.* (148). $\qquad \square$

**Lemma 20.** *Under the same assumption of Lemma 19, the lower bound in Eq. (148) is achieved for any critical point $(\boldsymbol{W}, \boldsymbol{H}, \boldsymbol{b})$ of Eq. (127) if and only if the following hold*

$$\boldsymbol{w}_k = \boldsymbol{h}_i^{(k)}, \quad \forall k \in [K], \ i \in [n], \tag{170}$$

$$\tilde{\boldsymbol{h}}_i := \frac{1}{K} \sum_{k=1}^{K} \boldsymbol{h}_i^{(k)} = \boldsymbol{0}, \quad \forall i \in [n], \tag{171}$$

$$\boldsymbol{W}\boldsymbol{W}^\top = \frac{K}{K-1} \left( \boldsymbol{I}_K - \frac{1}{K} \boldsymbol{1}_K \boldsymbol{1}_K^\top \right), \tag{172}$$

$$\boldsymbol{b}^\star = b^\star \boldsymbol{1}, \quad \forall b^\star \in \mathbb{R}, \tag{173}$$

$$c_1' = \left[ (K-1) \exp\left( -\frac{\gamma K}{K-1} \right) \right]^{-1}. \tag{174}$$

*Proof.* With the proof of Lemma 19, to achieve the lower bound, we need at least *Eqs.* (161) and (165) to hold, *i.e,*

$$\tilde{\boldsymbol{h}}_i = \frac{1}{K}\sum_{k=1}^{K}\boldsymbol{h}_i^{(k)} = \boldsymbol{0}, \ \forall i \in [n] \quad \text{and} \quad \boldsymbol{w}_k = \boldsymbol{h}_i^{(k)}, \ \forall k \in [K], \ i \in [n], \tag{175}$$

which further implies that

$$\sum_{k=1}^{K}\boldsymbol{w}_k = \sum_{k=1}^{K}\boldsymbol{h}_i^{(k)} = \boldsymbol{0}. \tag{176}$$

Then,

$$\sum_{j=1}^{K}(\gamma\boldsymbol{w}_j^{\top}\boldsymbol{h}_i^{(k)} - b_j) = \gamma\left(\sum_{j=1}^{K}\boldsymbol{w}_j^{\top}\right)\boldsymbol{h}_i^{(k)} - \sum_{j=1}^{K}b_j = 0 - \sum_{j=1}^{K}b_j = -K\bar{b}, \tag{177}$$

where $\bar{b} = \frac{1}{K}\sum_{j=1}^{K}b_j$, and

$$K(\gamma\boldsymbol{w}_k^{\top}\boldsymbol{h}_i^{(k)} - b_k) = K\left(\gamma\|\boldsymbol{w}_k\|_2^2 - b_k\right) = K\gamma - Kb_k. \tag{178}$$

Combining *Eqs.* (135), (136), (177), and (178), we have

$$c_1' = \left[(K-1)\exp\left(\frac{\sum_{j=1}^{K}(\gamma\boldsymbol{w}_j^{\top}\boldsymbol{h}_i^{(k)} - b_j) - K(\gamma\boldsymbol{w}_k^{\top}\boldsymbol{h}_i^{(k)} - b_k)}{K-1}\right)\right]^{-1}$$

$$= \left[(K-1)\exp\left(\frac{-K\bar{b} - (K\gamma - Kb_k)}{K-1}\right)\right]^{-1} \tag{179}$$

$$= \left[(K-1)\exp\left(\frac{K}{K-1}\left(b_k - \bar{b} - \gamma\right)\right)\right]^{-1}. \tag{180}$$

Since the $c_1' > 0$ can be arbitrary number, we choose the same $c_1'$ for all $k \in [K]$. Then, we have

$$b_k - \bar{b} - \gamma = b_j - \bar{b} - \gamma \Leftrightarrow b_k = b_j, \ \forall j \neq k \in [K]. \tag{181}$$

Therefore, we can write $\boldsymbol{b} = b\mathbf{1}_K$. Moreover, plugging *Eqs.* (175) and (181) into *Eq.* (178), we get

$$\gamma\boldsymbol{w}_j^{\top}\boldsymbol{h}_i^{(k)} - b_j = \gamma\boldsymbol{w}_\ell^{\top}\boldsymbol{h}_i^{(k)} - b_\ell \Leftrightarrow \boldsymbol{w}_j^{\top}\boldsymbol{h}_i^{(k)} = \boldsymbol{w}_\ell^{\top}\boldsymbol{h}_i^{(k)} \Leftrightarrow \boldsymbol{w}_j^{\top}\boldsymbol{w}_k = \boldsymbol{w}_\ell^{\top}\boldsymbol{w}_k. \tag{182}$$

Then, combining with *Eq.* (176), we have

$$1 = \|\boldsymbol{w}_k\|_2^2 = -\sum_{\substack{j=1\\j\neq k}}^{K}\boldsymbol{w}_k^{\top}\boldsymbol{w}_j = -(K-1)\boldsymbol{w}_k^{\top}\boldsymbol{w}_j \tag{183}$$

$$\Rightarrow \boldsymbol{w}_k^{\top}\boldsymbol{w}_j = -\frac{1}{K-1}, \ \forall j \neq k \in [K]. \tag{184}$$

Therefore,

$$\boldsymbol{W}\boldsymbol{W}^{\top} = \frac{K}{K-1}\left(\boldsymbol{I}_K - \frac{1}{K}\mathbf{1}_K\mathbf{1}_K^{\top}\right). \tag{185}$$

Finally, plugging the results in *Eq.* (181) into (180), we have

$$c_1' = \left[(K-1)\exp\left(\frac{K}{K-1}\left(b_k - \bar{b} - \gamma\right)\right)\right]^{-1} = \left[(K-1)\exp\left(-\frac{\gamma K}{K-1}\right)\right]^{-1}.$$

When $f_{\text{ce}}$ defined in *Eq.* (127) achieves its lower bound, it theoretically satisfies

$$\frac{\partial f_{\text{ce}}}{\partial b_k} = \frac{1}{nK}\left(n - \sum_{j=1}^{K}\sum_{i=1}^{n}\frac{\exp(\gamma\boldsymbol{w}_k^{\top}\boldsymbol{h}_i^{(j)} - b_k)}{\sum_{\ell=1}^{K}\exp(\gamma\boldsymbol{w}_\ell^{\top}\boldsymbol{h}_i^{(j)} - b_\ell)}\right) = 0, \ \forall k, \ell \in [K]. \tag{186}$$

However, combining with *Eqs.* (181) and (184), we have

$$n - \sum_{j=1}^{K} \sum_{i=1}^{n} \frac{\exp(\gamma \boldsymbol{w}_k^\top \boldsymbol{h}_i^{(j)} - b_k)}{\sum_{\ell=1}^{K} \exp(\gamma \boldsymbol{w}_\ell^\top \boldsymbol{h}_i^{(j)} - b_\ell)} \tag{187}$$

$$= n - \sum_{j=1}^{K} \sum_{i=1}^{n} \frac{\exp(\gamma \boldsymbol{w}_k^\top \boldsymbol{h}_i^{(j)} - b_k)}{\sum_{\ell \neq j}^{K} \exp(-\frac{\gamma}{K-1} - b_\ell) + \exp(\gamma - b_j)} \tag{188}$$

$$= n - \sum_{i=1}^{n} \left( \frac{\sum_{j \neq k}^{K} \exp(-\frac{\gamma}{K-1} - b) + \exp(\gamma - b)}{(K-1) \exp(-\frac{\gamma}{K-1} - b) + \exp(\gamma - b)} \right) \tag{189}$$

$$= n - n \tag{190}$$

$$\equiv 0, \tag{191}$$

which means when the function $f_{\mathrm{ce}}$ achieves its lower bound, the classifier biases $\boldsymbol{b}^\star$ only needs to satisfies $\boldsymbol{b}^\star = b^\star \mathbf{1}$. In other words, in terms of the classifier biases, the function $f_{\mathrm{ce}}$ has infinitely many minima. $\qquad\square$

### E.4 Proof of Theorems 5 and 6

**Lemma 21.** *(1) For a polynomial $\boldsymbol{P}(\gamma)$ and Sigmoid function $\sigma(s\gamma - b) = \frac{1}{1+\exp(-s\gamma+b)}$ with $s, b \in \mathbb{R}$, as $\gamma$ approaches positive infinity, their product exhibits exponential decay, converging to zero,*

$$\lim_{\gamma \to +\infty} \boldsymbol{P}(\gamma)\sigma(s\gamma - b) = \lim_{\gamma \to +\infty} \boldsymbol{P}(\gamma)\frac{1}{1 + \exp(-s\gamma + b)} = 0, \quad \forall s < 0, \ b \in \mathbb{R}, \tag{192}$$

$$\lim_{\gamma \to +\infty} \boldsymbol{P}(\gamma)\big(1 - \sigma(s\gamma - b)\big) = \lim_{\gamma \to +\infty} \boldsymbol{P}(\gamma)\frac{\exp(-s\gamma + b)}{1 + \exp(-s\gamma + b)} = 0, \quad \forall s > 0, \ b \in \mathbb{R}. \tag{193}$$

*(2) For a polynomial $\boldsymbol{P}(\gamma)$ without constant term and Sigmoid function $\sigma(s\gamma - b) = \frac{1}{1+\exp(-s\gamma+b)}$ with $s, b \in \mathbb{R}$, their product approaches $0$ as $\gamma$ approaches zero, i.e.,*

$$\lim_{\gamma \to 0} \boldsymbol{P}(\gamma)\sigma(s\gamma - b) = \lim_{\gamma \to 0} \boldsymbol{P}(\gamma)\frac{1}{1 + \exp(-s\gamma + b)} = 0, \quad \forall s, b \in \mathbb{R}, \tag{194}$$

$$\lim_{\gamma \to 0} \boldsymbol{P}(\gamma)\big(1 - \sigma(s\gamma - b)\big) = \lim_{\gamma \to 0} \boldsymbol{P}(\gamma)\frac{\exp(-s\gamma + b)}{1 + \exp(-s\gamma + b)} = 0, \quad \forall s, b \in \mathbb{R}. \tag{195}$$

*Proof.* (1) Suppose that $\boldsymbol{P}(\gamma)$ is an $p$-degree polynomial in terms of $\gamma$, with the leading coefficient being $a_p$. Then by applying L'hospital's rule $p$ times, we have

$$\lim_{\gamma \to +\infty} \boldsymbol{P}(\gamma)\big(1 - \sigma(s\gamma - b)\big) = \lim_{\gamma \to +\infty} \boldsymbol{P}(\gamma)\frac{\exp(-s\gamma + b)}{1 + \exp(-s\gamma + b)}$$

$$= \lim_{\gamma \to +\infty} \frac{\boldsymbol{P}(\gamma)}{1 + \exp(s\gamma - b)}$$

$$= \lim_{\gamma \to +\infty} \frac{\mathrm{d}\boldsymbol{P}(\gamma)/\mathrm{d}\gamma}{s\exp(s\gamma - b)} = \cdots$$

$$= \lim_{\gamma \to +\infty} \frac{\mathrm{d}^p \boldsymbol{P}(\gamma)/\mathrm{d}\gamma^p}{s^p \exp(s\gamma - b)}$$

$$= \lim_{\gamma \to +\infty} \frac{p!\, a_p}{s^p \exp(s\gamma - b)} = 0, \quad \forall s > 0. \tag{196}$$

When $s < 0$, one can similarly prove Eq. (192).

(2) As $\boldsymbol{P}(\gamma)$ is without constant term, $\lim_{\gamma \to 0} \boldsymbol{P}(\gamma) = 0$. Therefore,

$$\lim_{\gamma \to 0} \boldsymbol{P}(\gamma) \sigma(s\gamma - b) = \lim_{\gamma \to 0} \boldsymbol{P}(\gamma) \lim_{\gamma \to 0} \sigma(s\gamma - b) = 0 \cdot \lim_{\gamma \to 0} \frac{1}{1 + \exp(-s\gamma + b)} = 0 \cdot \frac{1}{1 + \mathrm{e}^b} = 0,$$

$$\lim_{\gamma \to 0} \boldsymbol{P}(\gamma)\big(1 - \sigma(s\gamma - b)\big) = \lim_{\gamma \to 0} \boldsymbol{P}(\gamma) \lim_{\gamma \to 0} \big(1 - \sigma(s\gamma - b)\big)$$

$$= 0 \cdot \lim_{\gamma \to 0} \frac{\exp(-s\gamma + b)}{1 + \exp(-s\gamma + b)} = 0 \cdot \frac{\mathrm{e}^b}{1 + \mathrm{e}^b} = 0,$$

which are desired. $\qquad\square$

**Lemma 22.** *Let $\{s_j\}_{j=1}^K \subset \mathbb{R}$ and $\{b_j\}_{j=1}^K \subset \mathbb{R}$ be two sets of real numbers and*

$$s_k = \max\{s_j\}_{j=1}^K > \max\{s_j\}_{\substack{j=1 \\ j \neq k}}^K. \tag{197}$$

*(1) For a polynomial $\boldsymbol{P}(\gamma)$ and Softmax function $\mathtt{Softmax}_\ell(\gamma) = \frac{\exp(s_\ell\gamma - b_\ell)}{\sum_{j=1}^K \exp(s_j\gamma - b_j)}$, as $\gamma$ approaches positive infinity, their product exhibits exponential decay, converging to zero,*

$$\lim_{\gamma \to +\infty} \boldsymbol{P}(\gamma)\mathtt{Softmax}_\ell(\gamma) = \lim_{\gamma \to +\infty} \boldsymbol{P}(\gamma) \frac{\exp(s_\ell\gamma - b_\ell)}{\sum_{j=1}^K \exp(s_j\gamma - b_j)} = 0, \qquad \forall \ell \neq k, \tag{198}$$

$$\lim_{\gamma \to +\infty} \boldsymbol{P}(\gamma)\big(1 - \mathtt{Softmax}_k(\gamma)\big) = \lim_{\gamma \to +\infty} \boldsymbol{P}(\gamma) \frac{\sum_{\substack{j=1 \\ j \neq k}}^K \exp(s_j\gamma - b_j)}{\sum_{j=1}^K \exp(s_j\gamma - b_j)} = 0. \tag{199}$$

*(2) For a polynomial $\boldsymbol{P}(\gamma)$ without constant term and $\mathtt{Softmax}_\ell(\gamma) = \frac{\exp(s_\ell\gamma - b_\ell)}{\sum_{j=1}^K \exp(s_j\gamma - b_j)}$, their product approaches 0 as $\gamma$ approaches 0, and*

$$\lim_{\gamma \to 0} \boldsymbol{P}(\gamma)\mathtt{Softmax}_\ell(\gamma) = \lim_{\gamma \to 0} \boldsymbol{P}(\gamma) \frac{\exp(s_\ell\gamma - b_\ell)}{\sum_{j=1}^K \exp(s_j\gamma - b_j)} = 0, \qquad \forall j \neq k, \tag{200}$$

$$\lim_{\gamma \to 0} \boldsymbol{P}(\gamma)\big(1 - \mathtt{Softmax}_k(\gamma)\big) = \lim_{\gamma \to 0} \boldsymbol{P}(\gamma) \frac{\sum_{\substack{j=1 \\ j \neq k}}^K \exp(s_j\gamma - b_j)}{\sum_{j=1}^K \exp(s_j\gamma - b_j)} = 0. \tag{201}$$

*Proof.* (1) Suppose $\boldsymbol{P}(\gamma) = a_p\gamma^p + a_{p-1}\gamma^{p-1} + \cdots + a_1\gamma + a_0$. Then,

$$0 \leq \left|\boldsymbol{P}(\gamma)\mathtt{Softmax}_\ell(\gamma)\right| \leq \sum_{i=1}^p |a_i|\gamma^i \frac{\exp(s_\ell\gamma - b_\ell)}{\sum_{j=1}^K \exp(s_j\gamma - b_j)}$$

$$\leq \sum_{i=1}^p |a_i|\gamma^i \frac{\exp(s_\ell\gamma - b_\ell)}{\exp(s_k\gamma - b_k)} = \sum_{i=1}^p |a_i|\gamma^i \frac{1}{\mathrm{e}^{(s_k - s_\ell)\gamma - (b_k - b_\ell)}}. \tag{202}$$

As $s_k - s_\ell > 0$ for $\ell \neq k$, the right side approaches zero when $\gamma$ approaches positive infinity, according to Lemma 21(1). Therefore, $\lim_{\gamma \to 0} \boldsymbol{P}(\gamma)\mathtt{Softmax}_\ell(\gamma) = 0$.

By applying the arithmetic operations of limits,

$$\lim_{\gamma \to +\infty} \boldsymbol{P}(\gamma)\big(1 - \mathtt{Softmax}_k(\gamma)\big) = \sum_{\substack{\ell=1 \\ \ell \neq k}}^K \lim_{\gamma \to +\infty} \boldsymbol{P}(\gamma) \frac{\exp(s_\ell\gamma - b_\ell)}{\sum_{j=1}^K \exp(s_j\gamma - b_j)} = \sum_{\substack{\ell=1 \\ \ell \neq k}}^K 0 = 0. \tag{203}$$

(2) As $\boldsymbol{P}(\gamma)$ is without constant term, $\lim_{\gamma \to 0} \boldsymbol{P}(\gamma) = 0$. Therefore,

$$0 \leq \lim_{\gamma \to 0} \left|\boldsymbol{P}(\gamma)\mathtt{Softmax}_\ell(\gamma)\right| \leq \lim_{\gamma \to 0} \left|\boldsymbol{P}(\gamma)\right| \lim_{\gamma \to 0} \left|\mathtt{Softmax}_\ell(\gamma)\right| \leq 0 \cdot 1 = 0 \quad \text{and}$$

$$0 \leq \lim_{\gamma \to 0} \left|\boldsymbol{P}(\gamma)\big(1 - \mathtt{Softmax}_\ell(\gamma)\big)\right| \leq \lim_{\gamma \to 0} \left|\boldsymbol{P}(\gamma)\right| \lim_{\gamma \to 0} \left|1 - \mathtt{Softmax}_\ell(\gamma)\right| \leq 0 \cdot 1 = 0,$$

which lead to the conclusions. $\qquad\square$

**Theorem 23.** *When training the model using the BCE loss $f_{bce}$, as $\gamma$ approaches zero, the linear decrease in $\gamma$ leads to a linear decay in the convergence rate of unbiased decision scores. In contrary, once the **critical condition II** is satisfied, as $\gamma$ linearly approaches positive infinity, the convergence rate of unbiased decision scores decay exponentially.*

*Proof.* In the model training, for any unbiased decision score $s_i^{(kj)} = \gamma \boldsymbol{w}_k^\top \boldsymbol{h}_i^{(j)}$, its update step is determined by that of the classifier vector $\boldsymbol{w}_k$ and the feature $\boldsymbol{h}_i^{(j)}$. For an unconstrained feature model (UFM), $\{\boldsymbol{w}_k\}_k$ and $\{\boldsymbol{h}_i^{(j)}\}_{i,j}$ are independent variables. Therefore, with an iteration, they are updated as

$$\hat{\boldsymbol{w}}_k = \boldsymbol{w}_k - \eta \frac{\partial f_\mu}{\partial \boldsymbol{w}_k} \quad \text{and} \quad \hat{\boldsymbol{h}}_i^{(j)} = \boldsymbol{h}_i^{(j)} - \eta \frac{\partial f_\mu}{\partial \boldsymbol{h}_i^{(j)}}, \tag{204}$$

where $\eta$ is the learning rate and $\mu \in \{\text{ce}, \text{bce}\}$. Then, the unbiased decision score is updated as

$$\hat{s}_i^{(kj)} = \gamma \hat{\boldsymbol{w}}_k^\top \hat{\boldsymbol{h}}_i^{(j)} = \gamma \Big( \boldsymbol{w}_k - \eta \frac{\partial f_\mu}{\partial \boldsymbol{w}_k} \Big)^\top \Big( \boldsymbol{h}_i^{(j)} - \eta \frac{\partial f_\mu}{\partial \boldsymbol{h}_i^{(j)}} \Big) \tag{205}$$

$$= \gamma \boldsymbol{w}_k^\top \boldsymbol{h}_i^{(j)} - \gamma\eta \boldsymbol{w}_k^\top \frac{\partial f_\mu}{\partial \boldsymbol{h}_i^{(j)}} - \gamma\eta \Big( \frac{\partial f_\mu}{\partial \boldsymbol{w}_k} \Big)^\top \boldsymbol{h}_i^{(j)} + \gamma\eta^2 \Big( \frac{\partial f_\mu}{\partial \boldsymbol{w}_k} \Big)^\top \frac{\partial f_\mu}{\partial \boldsymbol{h}_i^{(j)}}, \tag{206}$$

and its update step is

$$\triangle(s_i^{(kj)}) = s_i^{(kj)} - \hat{s}_i^{(kj)} = -\gamma\eta \boldsymbol{w}_k^\top \frac{\partial f_\mu}{\partial \boldsymbol{h}_i^{(j)}} - \gamma\eta \Big( \frac{\partial f_\mu}{\partial \boldsymbol{w}_k} \Big)^\top \boldsymbol{h}_i^{(j)} + \gamma\eta^2 \Big( \frac{\partial f_\mu}{\partial \boldsymbol{w}_k} \Big)^\top \frac{\partial f_\mu}{\partial \boldsymbol{h}_i^{(j)}}. \tag{207}$$

For the BCE loss,

$$\frac{\partial f_{\text{bce}}}{\partial \boldsymbol{h}_i^{(j)}} = \frac{\gamma}{nK} \Bigg( - \frac{e^{-\gamma \boldsymbol{w}_j^\top \boldsymbol{h}_i^{(j)} + b_j}}{1 + e^{-\gamma \boldsymbol{w}_j^\top \boldsymbol{h}_i^{(j)} + b_j}} \boldsymbol{w}_j + \sum_{\substack{\ell=1 \\ \ell \neq j}}^{K} \frac{e^{\gamma \boldsymbol{w}_\ell^\top \boldsymbol{h}_i^{(j)} - b_\ell}}{1 + e^{\gamma \boldsymbol{w}_\ell^\top \boldsymbol{h}_i^{(j)} - b_\ell}} \boldsymbol{w}_\ell \Bigg), \tag{208}$$

$$\frac{\partial f_{\text{bce}}}{\partial \boldsymbol{w}_k} = \frac{\gamma}{nK} \sum_{i'=1}^{n} \Bigg( - \frac{e^{-\gamma \boldsymbol{w}_k^\top \boldsymbol{h}_{i'}^{(k)} + b_k}}{1 + e^{-\gamma \boldsymbol{w}_k^\top \boldsymbol{h}_{i'}^{(k)} + b_k}} \boldsymbol{h}_{i'}^{(k)} + \sum_{\substack{\ell'=1 \\ \ell' \neq k}}^{K} \frac{e^{\gamma \boldsymbol{w}_k^\top \boldsymbol{h}_{i'}^{(\ell')} - b_k}}{1 + e^{\gamma \boldsymbol{w}_k^\top \boldsymbol{h}_{i'}^{(\ell')} - b_k}} \boldsymbol{h}_{i'}^{(\ell')} \Bigg), \tag{209}$$

and

$$\triangle(s_i^{(kj)}) = \frac{\gamma^2 \eta}{nK} \boldsymbol{w}_k^\top \Bigg( \frac{e^{-\gamma \boldsymbol{w}_j^\top \boldsymbol{h}_i^{(j)} + b_j}}{1 + e^{-\gamma \boldsymbol{w}_j^\top \boldsymbol{h}_i^{(j)} + b_j}} \boldsymbol{w}_j - \sum_{\substack{\ell=1 \\ \ell \neq j}}^{K} \frac{e^{\gamma \boldsymbol{w}_\ell^\top \boldsymbol{h}_i^{(j)} - b_\ell}}{1 + e^{\gamma \boldsymbol{w}_\ell^\top \boldsymbol{h}_i^{(j)} - b_\ell}} \boldsymbol{w}_\ell \Bigg)$$

$$+ \frac{\gamma^2 \eta}{nK} \sum_{i'=1}^{n} \Bigg( \frac{e^{-\gamma \boldsymbol{w}_k^\top \boldsymbol{h}_{i'}^{(k)} + b_k}}{1 + e^{-\gamma \boldsymbol{w}_k^\top \boldsymbol{h}_{i'}^{(k)} + b_k}} \boldsymbol{h}_{i'}^{(k)} - \sum_{\substack{\ell'=1 \\ \ell' \neq k}}^{K} \frac{e^{\gamma \boldsymbol{w}_k^\top \boldsymbol{h}_{i'}^{(\ell')} - b_k}}{1 + e^{\gamma \boldsymbol{w}_k^\top \boldsymbol{h}_{i'}^{(\ell')} - b_k}} \boldsymbol{h}_{i'}^{(\ell')} \Bigg)^\top \boldsymbol{h}_i^{(j)}$$

$$+ \frac{\gamma^3 \eta^2}{n^2 K^2} \sum_{i'=1}^{n} \Bigg( \frac{e^{-\gamma \boldsymbol{w}_k^\top \boldsymbol{h}_{i'}^{(k)} + b_k}}{1 + e^{-\gamma \boldsymbol{w}_k^\top \boldsymbol{h}_{i'}^{(k)} + b_k}} \boldsymbol{h}_{i'}^{(k)} - \sum_{\substack{\ell'=1 \\ \ell' \neq k}}^{K} \frac{e^{\gamma \boldsymbol{w}_k^\top \boldsymbol{h}_{i'}^{(\ell')} - b_k}}{1 + e^{\gamma \boldsymbol{w}_k^\top \boldsymbol{h}_{i'}^{(\ell')} - b_k}} \boldsymbol{h}_{i'}^{(\ell')} \Bigg)^\top$$

$$\Bigg( \frac{e^{-\gamma \boldsymbol{w}_j^\top \boldsymbol{h}_i^{(j)} + b_j}}{1 + e^{-\gamma \boldsymbol{w}_j^\top \boldsymbol{h}_i^{(j)} + b_j}} \boldsymbol{w}_j - \sum_{\substack{\ell=1 \\ \ell \neq j}}^{K} \frac{e^{\gamma \boldsymbol{w}_\ell^\top \boldsymbol{h}_i^{(j)} - b_\ell}}{1 + e^{\gamma \boldsymbol{w}_\ell^\top \boldsymbol{h}_i^{(j)} - b_\ell}} \boldsymbol{w}_\ell \Bigg) \tag{210}$$

$$= \frac{\gamma^2 \eta}{nK} \Bigg( \frac{e^{-\gamma \boldsymbol{w}_j^\top \boldsymbol{h}_i^{(j)} + b_j}}{1 + e^{-\gamma \boldsymbol{w}_j^\top \boldsymbol{h}_i^{(j)} + b_j}} \boldsymbol{w}_k^\top \boldsymbol{w}_j - \sum_{\substack{\ell=1 \\ \ell \neq j}}^{K} \frac{e^{\gamma \boldsymbol{w}_\ell^\top \boldsymbol{h}_i^{(j)} - b_\ell}}{1 + e^{\gamma \boldsymbol{w}_\ell^\top \boldsymbol{h}_i^{(j)} - b_\ell}} \boldsymbol{w}_k^\top \boldsymbol{w}_\ell \Bigg)$$

$$+ \frac{\gamma^2 \eta}{nK} \sum_{i'=1}^{n} \Bigg( \frac{e^{-\gamma \boldsymbol{w}_k^\top \boldsymbol{h}_{i'}^{(k)} + b_k}}{1 + e^{-\gamma \boldsymbol{w}_k^\top \boldsymbol{h}_{i'}^{(k)} + b_k}} \boldsymbol{h}_{i'}^{(k)\top} \boldsymbol{h}_i^{(j)} - \sum_{\substack{\ell'=1 \\ \ell' \neq k}}^{K} \frac{e^{\gamma \boldsymbol{w}_k^\top \boldsymbol{h}_{i'}^{(\ell')} - b_k}}{1 + e^{\gamma \boldsymbol{w}_k^\top \boldsymbol{h}_{i'}^{(\ell')} - b_k}} \boldsymbol{h}_{i'}^{(\ell')\top} \boldsymbol{h}_i^{(j)} \Bigg)$$

$$
+ \frac{\gamma^3 \eta^2}{n^2 K^2} \Bigg( \sum_{i'=1}^{n} \frac{\mathrm{e}^{-\gamma \boldsymbol{w}_k^\top \boldsymbol{h}_{i'}^{(k)} + b_k}}{1 + \mathrm{e}^{-\gamma \boldsymbol{w}_k^\top \boldsymbol{h}_{i'}^{(k)} + b_k}} \frac{\mathrm{e}^{-\gamma \boldsymbol{w}_j^\top \boldsymbol{h}_i^{(j)} + b_j}}{1 + \mathrm{e}^{-\gamma \boldsymbol{w}_j^\top \boldsymbol{h}_i^{(j)} + b_j}} \boldsymbol{w}_j^\top \boldsymbol{h}_{i'}^{(k)}
$$

$$
- \sum_{i'=1}^{n} \sum_{\substack{\ell=1 \\ \ell \neq j}}^{K} \frac{\mathrm{e}^{-\gamma \boldsymbol{w}_k^\top \boldsymbol{h}_{i'}^{(k)} + b_k}}{1 + \mathrm{e}^{-\gamma \boldsymbol{w}_k^\top \boldsymbol{h}_{i'}^{(k)} + b_k}} \frac{\mathrm{e}^{\gamma \boldsymbol{w}_\ell^\top \boldsymbol{h}_i^{(j)} - b_\ell}}{1 + \mathrm{e}^{\gamma \boldsymbol{w}_\ell^\top \boldsymbol{h}_i^{(j)} - b_\ell}} \boldsymbol{w}_\ell^\top \boldsymbol{h}_{i'}^{(k)}
$$

$$
- \sum_{i'=1}^{n} \sum_{\substack{\ell'=1 \\ \ell' \neq k}}^{K} \frac{\mathrm{e}^{\gamma \boldsymbol{w}_k^\top \boldsymbol{h}_{i'}^{(\ell')} - b_k}}{1 + \mathrm{e}^{\gamma \boldsymbol{w}_k^\top \boldsymbol{h}_{i'}^{(\ell')} - b_k}} \frac{\mathrm{e}^{-\gamma \boldsymbol{w}_j^\top \boldsymbol{h}_i^{(j)} + b_j}}{1 + \mathrm{e}^{-\gamma \boldsymbol{w}_j^\top \boldsymbol{h}_i^{(j)} + b_j}} \boldsymbol{w}_j^\top \boldsymbol{h}_{i'}^{(\ell')}
$$

$$
+ \sum_{i'=1}^{n} \sum_{\substack{\ell'=1 \\ \ell' \neq k}}^{K} \sum_{\substack{\ell=1 \\ \ell \neq k}}^{K} \frac{\mathrm{e}^{\gamma \boldsymbol{w}_k^\top \boldsymbol{h}_{i'}^{(\ell')} - b_k}}{1 + \mathrm{e}^{\gamma \boldsymbol{w}_k^\top \boldsymbol{h}_{i'}^{(\ell')} - b_k}} \frac{\mathrm{e}^{\gamma \boldsymbol{w}_\ell^\top \boldsymbol{h}_i^{(j)} - b_\ell}}{1 + \mathrm{e}^{\gamma \boldsymbol{w}_\ell^\top \boldsymbol{h}_i^{(j)} - b_\ell}} \boldsymbol{w}_\ell^\top \boldsymbol{h}_{i'}^{(\ell')} \Bigg). \tag{211}
$$

Then,

$$
0 \leq \left| \triangle(s_i^{(kj)}) \right| \leq \frac{\gamma^2 \eta}{nK} \Bigg( \frac{\mathrm{e}^{-\gamma \boldsymbol{w}_j^\top \boldsymbol{h}_i^{(j)} + b_j}}{1 + \mathrm{e}^{-\gamma \boldsymbol{w}_j^\top \boldsymbol{h}_i^{(j)} + b_j}} + \sum_{\substack{\ell=1 \\ \ell \neq j}}^{K} \frac{\mathrm{e}^{\gamma \boldsymbol{w}_\ell^\top \boldsymbol{h}_i^{(j)} - b_\ell}}{1 + \mathrm{e}^{\gamma \boldsymbol{w}_\ell^\top \boldsymbol{h}_i^{(j)} - b_\ell}} \Bigg)
$$

$$
+ \frac{\gamma^2 \eta}{nK} \sum_{i'=1}^{n} \Bigg( \frac{\mathrm{e}^{-\gamma \boldsymbol{w}_k^\top \boldsymbol{h}_{i'}^{(k)} + b_k}}{1 + \mathrm{e}^{-\gamma \boldsymbol{w}_k^\top \boldsymbol{h}_{i'}^{(k)} + b_k}} + \sum_{\substack{\ell'=1 \\ \ell' \neq k}}^{K} \frac{\mathrm{e}^{\gamma \boldsymbol{w}_k^\top \boldsymbol{h}_{i'}^{(\ell')} - b_k}}{1 + \mathrm{e}^{\gamma \boldsymbol{w}_k^\top \boldsymbol{h}_{i'}^{(\ell')} - b_k}} \Bigg)
$$

$$
+ \frac{\gamma^3 \eta^2}{n^2 K^2} \Bigg( \sum_{i'=1}^{n} \frac{\mathrm{e}^{-\gamma \boldsymbol{w}_k^\top \boldsymbol{h}_{i'}^{(k)} + b_k}}{1 + \mathrm{e}^{-\gamma \boldsymbol{w}_k^\top \boldsymbol{h}_{i'}^{(k)} + b_k}} \frac{\mathrm{e}^{-\gamma \boldsymbol{w}_j^\top \boldsymbol{h}_i^{(j)} + b_j}}{1 + \mathrm{e}^{-\gamma \boldsymbol{w}_j^\top \boldsymbol{h}_i^{(j)} + b_j}}
$$

$$
+ \sum_{i'=1}^{n} \sum_{\substack{\ell=1 \\ \ell \neq j}}^{K} \frac{\mathrm{e}^{-\gamma \boldsymbol{w}_k^\top \boldsymbol{h}_{i'}^{(k)} + b_k}}{1 + \mathrm{e}^{-\gamma \boldsymbol{w}_k^\top \boldsymbol{h}_{i'}^{(k)} + b_k}} \frac{\mathrm{e}^{\gamma \boldsymbol{w}_\ell^\top \boldsymbol{h}_i^{(j)} - b_\ell}}{1 + \mathrm{e}^{\gamma \boldsymbol{w}_\ell^\top \boldsymbol{h}_i^{(j)} - b_\ell}}
$$

$$
+ \sum_{i'=1}^{n} \sum_{\substack{\ell'=1 \\ \ell' \neq k}}^{K} \frac{\mathrm{e}^{\gamma \boldsymbol{w}_k^\top \boldsymbol{h}_{i'}^{(\ell')} - b_k}}{1 + \mathrm{e}^{\gamma \boldsymbol{w}_k^\top \boldsymbol{h}_{i'}^{(\ell')} - b_k}} \frac{\mathrm{e}^{-\gamma \boldsymbol{w}_j^\top \boldsymbol{h}_i^{(j)} + b_j}}{1 + \mathrm{e}^{-\gamma \boldsymbol{w}_j^\top \boldsymbol{h}_i^{(j)} + b_j}}
$$

$$
+ \sum_{i'=1}^{n} \sum_{\substack{\ell'=1 \\ \ell' \neq k}}^{K} \sum_{\substack{\ell=1 \\ \ell \neq k}}^{K} \frac{\mathrm{e}^{\gamma \boldsymbol{w}_k^\top \boldsymbol{h}_{i'}^{(\ell')} - b_k}}{1 + \mathrm{e}^{\gamma \boldsymbol{w}_k^\top \boldsymbol{h}_{i'}^{(\ell')} - b_k}} \frac{\mathrm{e}^{\gamma \boldsymbol{w}_\ell^\top \boldsymbol{h}_i^{(j)} - b_\ell}}{1 + \mathrm{e}^{\gamma \boldsymbol{w}_\ell^\top \boldsymbol{h}_i^{(j)} - b_\ell}} \Bigg). \tag{212}
$$

Each term in the right side of Eq. (212) is product of a polynomial without constant term and Sigmoid functions in terms of $\gamma$. When the **critical condition II** holds, i.e.,

$$
\min \bigcup_{k=1}^{K} \{\boldsymbol{w}_k^\top \boldsymbol{h}_i^{(k)}\}_{i=1}^{n} > 0 > \max \bigcup_{k=1}^{K} \bigcup_{\substack{j=1 \\ j \neq k}}^{K} \{\boldsymbol{w}_k^\top \boldsymbol{h}_i^{(j)}\}_{i=1}^{n}, \tag{213}
$$

every term in Eq. (212) satisfies the requirements of Lemma 21(1); then, as $\gamma$ increases, $\triangle(s_i^{(kj)})$ decays exponentially and converges to 0.

When $\gamma$ approaches zero, each term in the right side of Eq. (212) satisfies the requirements of Lemma 21(2). Then, as $\gamma$ decreases linearly toward 0, every term in the right side of Eq. (212) decays to 0 at a quadratic or cubic rate; consequently, $\triangle(s_i^{(kj)})$ decays quadratically to 0. Consider that the optimal values of unbiased decision scores are $\gamma$ or $-\frac{\gamma}{K-1}$, which are linearly decay as $\gamma$ decreases. Therefore, the convergence rate of the unbiased decision scores decay linearly when $\gamma$ linearly decreases to zero. $\qquad \square$

**Theorem 24.** *When training the model using the CE loss $f_{ce}$, as $\gamma$ approaches zero, the linear decrease in $\gamma$ leads to a linear decay in the convergence rate of unbiased decision scores. Once the **critical condition I** is satisfied, as $\gamma$ linearly approaches positive infinity, the convergence rate of unbiased decision scores decay exponentially.*

*Proof.* For the CE loss,

$$\frac{\partial f_{\text{ce}}}{\partial \boldsymbol{h}_i^{(j)}} = \frac{\gamma}{nK} \left[ \left( \frac{e^{\gamma \boldsymbol{w}_j^\top \boldsymbol{h}_i^{(j)} - b_j}}{\sum_{m=1}^{K} e^{\gamma \boldsymbol{w}_m^\top \boldsymbol{h}_i^{(j)} - b_m}} - 1 \right) \boldsymbol{w}_j + \sum_{\substack{\ell=1 \\ \ell \neq j}}^{K} \frac{e^{\gamma \boldsymbol{w}_\ell^\top \boldsymbol{h}_i^{(j)} - b_\ell}}{\sum_{m=1}^{K} e^{\gamma \boldsymbol{w}_m^\top \boldsymbol{h}_i^{(j)} - b_m}} \boldsymbol{w}_\ell \right], \tag{214}$$

$$\frac{\partial f_{\text{ce}}}{\partial \boldsymbol{w}_k} = \frac{\gamma}{nK} \sum_{i'=1}^{n} \left[ \left( \frac{e^{\gamma \boldsymbol{w}_k^\top \boldsymbol{h}_{i'}^{(k)} - b_k}}{\sum_{m=1}^{K} e^{\gamma \boldsymbol{w}_m^\top \boldsymbol{h}_{i'}^{(k)} - b_m}} - 1 \right) \boldsymbol{h}_{i'}^{(k)} + \sum_{\substack{\ell'=1 \\ \ell' \neq k}}^{K} \frac{e^{\gamma \boldsymbol{w}_k^\top \boldsymbol{h}_{i'}^{(\ell')} - b_k}}{\sum_{m=1}^{K} e^{\gamma \boldsymbol{w}_m^\top \boldsymbol{h}_{i'}^{(\ell')} - b_m}} \boldsymbol{h}_{i'}^{(\ell')} \right]. \tag{215}$$

The critical condition I is

$$\boldsymbol{w}_k \boldsymbol{h}_i^{(k)} > \max\{\boldsymbol{w}_j \boldsymbol{h}_i^{(k)} : j = 1, 2, \cdots, K, \text{and } j \neq k\}, \quad \forall k, i. \tag{216}$$

Similar to the proof of Theorem 23, one can easily get the conclusions with help of Lemma 22. $\quad\square$

Table 11: The number of training epochs required by ResNet18 to reach the stopping criterion in *Eq.* (217) on MNIST trained by CE and BCE with different $\gamma$. "$*$" indicating that the data is used for fitting the functions describing the variation of epoch number with respect to $\gamma$.

| CE | | | | BCE | | | |
|---|---|---|---|---|---|---|---|
| $\gamma$ | Ep. No. | $\gamma$ | Ep. No. | $\gamma$ | Ep. No. | $\gamma$ | Ep. No. |
| $0.01^*$ | 2743 | 2 | 28 | $0.01^*$ | 2708 | 2 | 24 |
| 0.02 | 1420 | 3 | 31 | 0.02 | 1469 | 3 | 22 |
| $0.04^*$ | 792 | 4 | 64 | $0.04^*$ | 822 | 4 | 22 |
| 0.06 | 593 | 5 | 31 | 0.06 | 513 | 5 | 14 |
| $0.08^*$ | 392 | $6^*$ | 77 | $0.08^*$ | 420 | 6 | 15 |
| $0.1^*$ | 303 | 6.5 | 145 | $0.1^*$ | 310 | 7 | 21 |
| $0.2^*$ | 160 | $7^*$ | 180 | $0.2^*$ | 163 | $8^*$ | 37 |
| 0.3 | 122 | 7.5 | 308 | 0.3 | 119 | 9 | 53 |
| $0.4^*$ | 83 | $8^*$ | 435 | $0.4^*$ | 82 | $10^*$ | 83 |
| 0.5 | 66 | 8.5 | 741 | 0.5 | 65 | 11 | 137 |
| 0.6 | 58 | $9^*$ | 1448 | 0.6 | 52 | $12^*$ | 243 |
| 0.7 | 51 | $9.5^*$ | 2083 | 0.7 | 42 | 13 | 411 |
| 0.8 | 42 | | | 0.8 | 39 | $14^*$ | 737 |
| 0.9 | 42 | | | 0.9 | 37 | 15 | 1165 |
| 1.0 | 37 | | | 1.0 | 33 | $16^*$ | 2076 |

## F  EXPERIMENTAL VERIFICATION FOR THEOREMS 5 AND 6

In Sec. 4.2 of the paper, we theoretically analyze how the convergence rate of unbiased positive and negative decision scores changes as $\gamma$ linearly increases or decreases, which reflect the convergence behavior of the CE and BCE losses. Specifically, (1) when $\gamma$ decreases linearly toward 0, the convergence rate of the losses decays linearly, meaning that the number of training epochs required to reach their minima increases linearly; (2) when critical condition I or II is satisfied, as $\gamma$ increases linearly, the convergence rate of the losses decays exponentially, meaning that the number of training epochs required to reach their minima increases exponentially.

To directly verify the above conclusions, we set the NC structure at the minimum point as the stopping criterion, and train ResNet18 on MNIST with varying gamma. In these experiments, the stopping criterion is

$$
\left| \boldsymbol{w}_k^\top \boldsymbol{h}_i^{(k)} - 1 \right| \le t, \quad \forall k \in [K], i \in [n],
$$
$$
\left| \boldsymbol{w}_j^\top \boldsymbol{h}_i^{(k)} + \frac{1}{K-1} \right| \le t, \quad \forall j \ne k \in [K], i \in [n].
\tag{217}
$$

When $\gamma < 5$, we set $t = 0.001$; when $\gamma \ge 5$, we set $t = 0.0035$. Moreover, we use SGD optimizer and fix the learning rate at $0.01$. Table 11 and Fig. 16 show the epoch numbers required by CE and BCE losses when they satisfying the criterion. For cases with $\gamma < 0.5$ and $\gamma > 5$, we took six and five points to fit the functions that reflect the variation of epoch number with respect to $\gamma$, for the CE and BCE losses, respectively. When $\gamma < 0.5$, the fitted curves for CE and BCE are

$$
\frac{27.14}{\gamma} + 44.41 \quad \text{and} \quad \frac{26.68}{\gamma} + 61.50,
\tag{218}
$$

respectively. In contrast, when $\gamma > 5$, the fitted curves for the CE and BCE are

$$
0.54 \mathrm{e}^{0.87\gamma} - 61.83 \quad \text{and} \quad 0.49 \mathrm{e}^{0.52\gamma} - 3.79,
\tag{219}
$$

respectively. These results align with Theorems 5 and 6. Moreover, when $\gamma$ is large, the coefficient $(0.52)$ of $\gamma$ in the fitting curve for BCE is less than that $(0.87)$ for CE, indicating that BCE converges faster than CE in the large feature spaces.

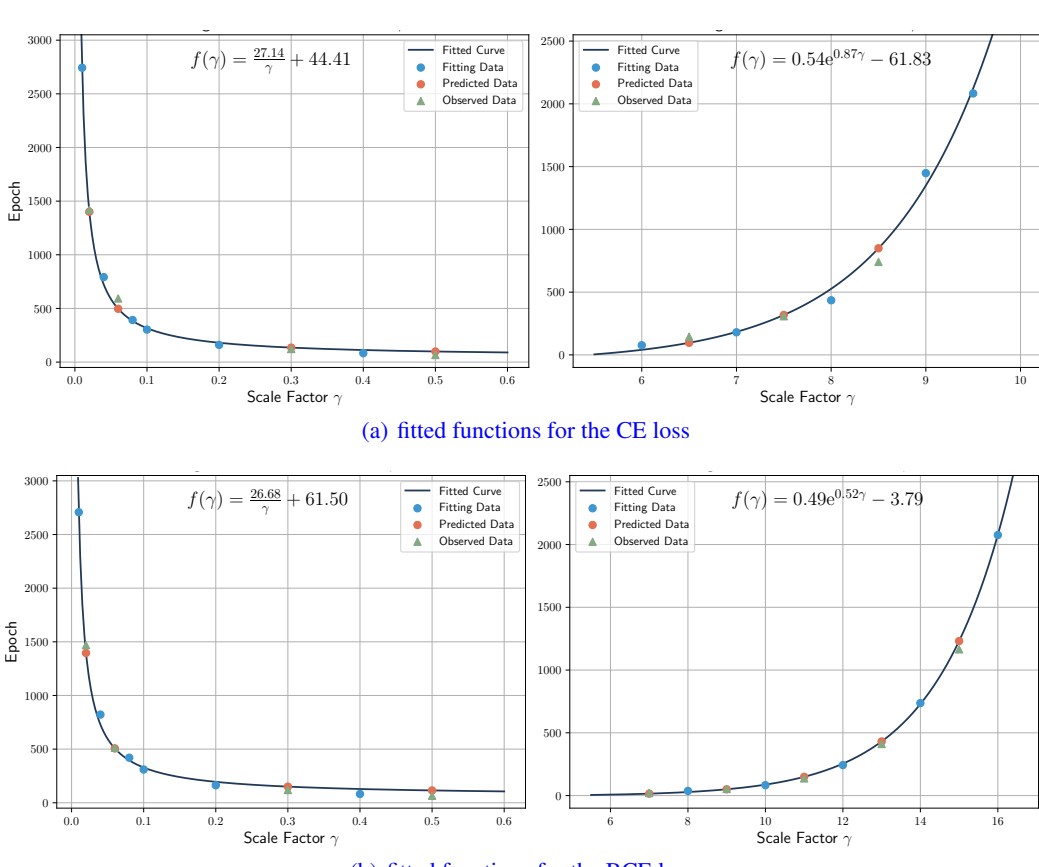

(a) fitted functions for the CE loss

(b) fitted functions for the BCE loss

Figure 16: The fitted functions describing the variation of epoch number with respect to $\gamma$ for CE (top) and BCE (bottom) losses. The epoch numbers in the figure indicate the training duration required to achieve minimum values (i.e., the criterion in Eq. (217)) of the two losses for training ResNet18 on MNIST, with different scale factors $\gamma$. The blue dots represent that the data was used for function fitting, the red dots represent the predicted values, and the green triangles represent the observed data in experiments.

