# OpenReview forum: "Classification vs. Deep Feature Learning in Normalized Spaces with Different Scaling"
_ICLR.cc/2026/Conference — Submitted to ICLR 2026_

### Official Review · Reviewer_bo89 · 2025-10-25

**Soundness:** 3
**Presentation:** 3
**Contribution:** 1
**Rating:** 2
**Confidence:** 5

**Summary:**

This paper first propose a loss function for deep feature learning based on global constraints—the binary cross-entropy (BCE) loss, applied to multi-class tasks. Then the authors perform a detailed comparison between cross-entropy (CE) and BCE losses in the normalized space, showing that normalized BCE can also achieve neural collapse (NC). As a last contribution, the authors highlight the key differences between classification and deep feature learning by analyzing how the convergence rates of CE and BCE vary with the scale factor. The authors conduct experiments and show that compared to the CE loss, their proposed BCE loss achieves better results on long-tailed recognition and open-set recognition.

**Strengths:**

The main strengths of the paper can be summarized as follows:
i) The authors propose a method that use binary cross-entropy (BCE) loss function (unfortunately the method is not novel) and provide a detailed comparison against the classical cross entropy (CE) loss.
ii) The authors analyse how the loss functions CE and BCE behaves with varying scale factor $\gamma$.
iii) Conducted experiments show that their proposed BCE loss achieves better results on long-tailed recognition and open-set recognition compared to CE loss.

**Weaknesses:**

The primary weakness of the paper lies in its lack of novelty. Both the proposed loss functions and the analyses concerning the scale factor are not new contributions. My detailed comments are as follows:

The authors conduct an extensive analysis on the scale factor. However, this concept merely represents the radius of the hypersphere on which data are distributed in methods such as ArcFace and CosFace. This same hypersphere also contains the regular simplex in deep simplex classifiers [R2, R3], all of which pursue the same objective as neural collapse. The choice of hypersphere radius has been thoroughly investigated in the literature. For instance, the CosFace paper [R1] provides a theoretical lower bound for the radius based on the number of classes and the minimum posterior probability of class centers (see Eq. 6 in their paper). In practice, larger datasets with more classes benefit from larger radius values, and this is theoretically linked to the feature dimension. ArcFace, for example, fixes the radius at 64 for datasets with thousands of identities, a value shown to work well on large-scale benchmarks. Similarly, as stated in [R2], data samples theoretically lie on the surface of an expanding hypersphere as the feature dimension increases. Thus, smaller radii may suffice for low-dimensional illustrative experiments, but larger values are required for high-dimensional cases with high-dimensional spaces. There are also experimental results with varying scale factors in [R2] . Overall, the relationship between the scale factor and hypersphere radius is already well-established, making this part of the paper non-novel.

Regarding the proposed methodology, there exist closely related approaches in prior work. The proposed loss is highly similar to the Dot-Regression Loss in [R3], while [R2] presents an even simpler loss function that also induces neural collapse. Numerous other loss functions targeting neural collapse have been proposed, yet the authors neither cite nor compare against these alternatives.

Finally, the proposed method has a critical limitation that restricts its applicability to large-scale datasets. When d<C−1, the class centers cannot be arranged on the vertices of a regular simplex. This condition commonly arises in modern large-scale datasets, rendering the proposed approach impractical. The authors should therefore suggest modifications or alternatives that can address such cases.

In conclusion, while the paper aligns with the broader line of research on neural collapse, its contributions are incremental. The authors fail to provide compelling justification for why their method should be preferred over many existing alternatives.

Minor Issues: i) There are some typos and English Grammar mistakes that must be corrected, e.g., page 3, libe 155, ... implemented by divided with ...
ii) I did not get the part that is related to unique global minimum. In neural collapse, there are many global minimums that can be simply obtained by rotating the regular simplex.


[R1] H. Wang, Y. W. Z. Zhou, X. Ji, D. Gong, J. Zhou, Z. Li, and W. Liu, ‘‘Cosface: Large margin cosine loss for deep face recognition,’’ in IEEE Society Conference on Computer Vision and Pattern Recognition (CVPR), 2018.
[R2] H. Cevikalp, H. Saribas, B. Uzun, “Reaching nirvana: Maximizing the margin in both Euclidean and angular spaces for deep neural network classification,” IEEE Transaction on Neural Networks and Learning Systems, vol. 36, 2025.
[R3] Y. Yang, S. Chen, X. Li, L. Xie, Z. Lin, and D. Tao, “Inducing neural collapse in imbalanced learning: Do we really need a learnable classifier at the end of deep neural network?” in Proc. Adv. Neural Inf. Process. Syst., 2022, pp. 37991–38002.

**Questions:**

I raised some questions in Weaknesses part. I really appreciate if the authors answer them.

---

> ### Author Response · Authors · 2025-11-19
> **Part I**
>
> Thank you for taking the time to review our paper.
>
> Our work is the first to clearly distinguish the tasks of classification and deep feature learning, while the former reflects the model’s local, sample-wise behavior on a dataset, and the latter focuses on training the model to capture global, sample-independent representations.
>
> The main contributions of our paper are summarized as follows:
> 1. We demonstrate that, compared with CE, BCE is more suitable for deep feature learning.
> 2. We provide the first in-depth analysis of the differences between classification and deep feature learning.
>
> ---
> **W1:**
>
> 	The authors conduct an extensive analysis on the scale factor. However, this concept merely represents the radius of the hypersphere on which data are distributed in methods such as ArcFace and CosFace. This same hypersphere also contains the regular simplex in deep simplex classifiers [R2, R3], all of which pursue the same objective as neural collapse. The choice of hypersphere radius has been thoroughly investigated in the literature. For instance, the CosFace paper [R1] provides a theoretical lower bound for the radius based on the number of classes and the minimum posterior probability of class centers (see Eq. 6 in their paper). In practice, larger datasets with more classes benefit from larger radius values, and this is theoretically linked to the feature dimension. ArcFace, for example, fixes the radius at 64 for datasets with thousands of identities, a value shown to work well on large-scale benchmarks. Similarly, as stated in [R2], data samples theoretically lie on the surface of an expanding hypersphere as the feature dimension increases. Thus, smaller radii may suffice for low-dimensional illustrative experiments, but larger values are required for high-dimensional cases with high-dimensional spaces. There are also experimental results with varying scale factors in [R2] . Overall, the relationship between the scale factor and hypersphere radius is already well-established, making this part of the paper non-novel.
>
> Many previous works, including CosFace [R1] and ArcFace [R4], have indeed analyzed the scale factor. However, we would like to emphasize that the main contribution of our work does not lie in analyzing the scale factor, but rather in analyzing the differences between the tasks of classification and deep feature learning within normalized spaces with different scales. Many researchers, including **ozw4** (and possibly including **bo89**), implicitly assume that these two tasks are equivalent yet. Our work is the first to theoretically and empirically demonstrate that they behave differently under different feature space.
>
> Existing works such as [R2, R3] either determine the specific value of the scale factor empirically or based on prior experimental experience. Some approaches, including CosFace [R1], perform preliminary theoretical analyses of the scale factor. However, CosFace’s treatment of the scale factor has several limitations:
> 1. CosFace is designed for face recognition tasks where the number of classes far exceeds the feature dimension, but its theoretical lower bound on the scale factor is derived under the assumption that the feature dimension is larger than the number of classes (see the paragraph following Eq. (15) in the CosFace’s supplementary material).
> 2. We acknowledge CosFace’s analysis on the lower bound when the feature dimension exceeds the number of classes. However, since CosFace is a face recognition (FR) method built upon the CE loss, its requirement that the scale factor exceed a certain threshold is tailored to the task of classification. For the deep feature learning, where the goal is to achieve better feature properties such as intra-class compactness and inter-class separability, the proper way to set the scale factor remains unexplored.
> 3. In practice of FR, a fixed threshold is commonly used to compare the similarity between two facial features. The FR methods require more compact and discriminative features. Therefore, the CE-based methods such as CosFace and ArcFace are not fully aligned with the inherent objective of FR. For instance, if one increases the scale factor in CosFace and ArcFace from 64 to 640 or 6400 (values still theoretically acceptable according to CosFace’s analysis), would this yield better FR? This question has not been addressed in the existing literature. With our analysis, an excessively large scale factor is detrimental to deep feature learning. Since FR requires models to extract features with better compactness and separability rather than merely achieving higher classification accuracy, setting the too large scale factor would likely fail to produce the optimal FR.
>
> In summary, **bo89**’s interpretation of the scale factor appears to be either specific to classification or largely empirical, and it is not directly related to our work comparing classification and deep feature learning.

---

> ### Author Response · Authors · 2025-11-19
> **Part II**
>
> **W2:**
>
> 	Regarding the proposed methodology, there exist closely related approaches in prior work. The proposed loss is highly similar to the Dot-Regression Loss in [R3], while [R2] presents an even simpler loss function that also induces neural collapse. Numerous other loss functions targeting neural collapse have been proposed, yet the authors neither cite nor compare against these alternatives.
>
> Our work is not aimed at proposing a new method, nor merely at demonstrating that losses such as BCE can lead to neural collapse. Instead, our focus is on analyzing the fundamental differences between the tasks of classification and deep feature learning. Under normalized, supervised, and closed-set settings, we are the first to point out that a large feature space (or large scale factor) benefits classification tasks but is unfavorable for deep feature learning, while an excessively small feature space is unfavorable for deep feature learning and even more detrimental for the classification tasks.
>
> ---
> **W3.**
>
> 	Finally, the proposed method has a critical limitation that restricts its applicability to large-scale datasets. When d<C−1, the class centers cannot be arranged on the vertices of a regular simplex. This condition commonly arises in modern large-scale datasets, rendering the proposed approach impractical. The authors should therefore suggest modifications or alternatives that can address such cases.
> 	In conclusion, while the paper aligns with the broader line of research on neural collapse, its contributions are incremental. The authors fail to provide compelling justification for why their method should be preferred over many existing alternatives.
>
> Finally, we would like to reiterate that our work is the first to analyze the differences between the tasks of classification and deep feature learning, rather than to propose a new method or a new loss function.
>
> Furthermore, we conjecture that the conclusions drawn in our paper regarding the differences between classification and deep feature learning are likely to hold in other settings as well. For instance, in cases where the number of classes exceeds the feature dimension, the existing study [R5] has shown that sample features still converge to be aligned with their corresponding classifier vectors. This result implies that each sample’s unbiased positive decision score converges to a fixed value. Similar to the proof of Theorem 5, once the critical condition I is satisfied, the rate of approaching this fixed value decays exponentially, as the scale factor increases linearly. Therefore, when the number of classes is greater than the feature dimension, an excessively large feature space is also likely to be unsuitable for deep feature learning.
>
> In summary, we provide the first analysis of the differences between classification and deep feature learning. Although our analysis is conducted under specific settings, it offers valuable insight for understanding these two tasks.
>
> ---
> **Minor Issues:**
>
> 	i) There are some typos and English Grammar mistakes that must be corrected, e.g., page 3, libe 155, ... implemented by divided with ...
> 	ii) I did not get the part that is related to unique global minimum. In neural collapse, there are many global minimums that can be simply obtained by rotating the regular simplex.
>
> 1. We appreciate the reviewer **bo89** for pointing out the grammatical error. We will thoroughly check the paper and make efforts to correct them.
>
> 2. The uniqueness of the BCE’s minimizer is with respect to its classifier biases, as we stated in line 78 of the paper. Indeed, rotating the classifier and feature vectors at the minimal point does not change the fact that they remain valid minimizers of BCE; however, such rotation does not alter the value of the classifier biases at these minimal points.
>
> ---
> ---
> **Reference:**
>
> [R1] H. Wang, Y. W. Z. Zhou, X. Ji, D. Gong, J. Zhou, Z. Li, and W. Liu, “CosFace: Large Margin Cosine Loss for Deep Face Recognition,” CVPR 2018.
>
> [R2] H. Cevikalp, H. Saribas, B. Uzun, “Reaching Nirvana: Maximizing the Margin in Both Euclidean and Angular Spaces for Deep Neural Network Classification,” IEEE TNNLS 2025.
>
> [R3] Y. Yang, S. Chen, X. Li, L. Xie, Z. Lin, and D. Tao, “Inducing Neural Collapse in Imbalanced Learning: Do We Really Need a Learnable Classifier at the End of Deep Neural Network?” NeurIPS 2022.
>
> [R4] J. Deng, J. Guo, N. Xue, et al. “ArcFace: Additive Angular Margin Loss for Deep Face Recognition,” CVPR 2019.
>
> [R5] W. Liu, L. Yu, A. Weller, B. Scholkopf, “Generalizing and Decoupling Neural Collapse via Hyperspherical Uniformity Gap,” ICLR2023.

---

> > ### Comment · Reviewer_bo89 · 2025-11-19
> > **response for part 2**
> >
> > You mention that you are not proposing a new loss function; however, the following sentence from your paper suggests otherwise: “we first derive a loss function based on the global constraints required in deep feature learning for the first time, which is the binary cross-entropy (BCE) loss used for multi-class tasks.”
> >
> > This wording gives the impression that you are introducing a new loss function. If that is not your intention, you should revise the sentence accordingly.

---

> > > ### Author Response · Authors · 2025-11-20
> > > **The sentence for BCE loss**
> > >
> > > Thank you again for your valuable suggestion.
> > >
> > > We have revised the relevant sentence in the paper to avoid ambiguous statements. The revised sentence is on lines 69-70 in the paper: “In this paper, for the first time, we derive a loss function from the basic requirements of deep feature learning, which coincidentally is the binary cross-entropy (BCE) loss used for multi-class tasks.”
> > >
> > > The BCE loss for multi-class classification tasks has been widely used in various fields; please refer to the Sec. W7 of our responses to reviewer **ozw4**. The BCE loss was not proposed for the first time by us. However, the previous studies have primarily used BCE in the context of multi-label or multi-class classification tasks. We are the first to reformulate the BCE loss from the perspective of deep feature learning, emphasizing its advantages over the CE loss in the context of deep feature learning.

---

> ### Comment · Reviewer_bo89 · 2025-11-19
> **response to authors**
>
> I strongly believe that there are critical issues in the way your experiments are conducted, as your results contradict both the underlying theory and previously published findings. As I noted in my review, the data samples theoretically lie on the surface of an expanding hypersphere as the feature dimension increases. Consequently, smaller radii may be sufficient for low-dimensional illustrative experiments, but higher-dimensional settings require larger values. This also implies that the scale factor depends on the architecture, since different architectures produce feature vectors of different dimensionalities. Additionally, it is related to the number of classes, as demonstrated in the CosFace paper.
>
> In my own experiments, varying the scale factor clearly shows that very small values such as 0.1 or 1 completely fail, which directly contradicts your reported results. Larger scale values are generally much safer and more consistent with theory. If you are obtaining poor performance with scale values like 32 or 64 for feature learning purposes, this indicates a deeper issue—possibly insufficient training, as neural collapse typically requires more than 300 epochs. There are published results in the literature showing that larger scale values yield better accuracies even for open set recognition, which means that better features are learned with those scale values.
>
> To properly validate your claims, you should test with different numbers of classes as well as different architectures. As they stand, your findings are inconsistent with both theoretical expectations and previously published results.

---

> > ### Author Response · Authors · 2025-11-20
> > **We believe that a larger number $K$ of classes corresponds to a bigger optimal scale factor $\gamma$.**
> >
> > Thanks for your reply.
> >
> > 1.	Our experimental results **do not contradict** the underlying theories and experiments previously published. The misunderstanding from reviewer **bo89** may stem from the data in Table 1, which were obtained on the small dataset, CIFAR10, and only the results for smaller ($\gamma = 0.1, 0.5, 1$) and larger ($\gamma = 12, 16, 32, 64$) scale factors were presented in Table 1.
> > 2.	We believe that as the number of classes $K$ of the dataset increases, the optimal scale factor $\gamma$ should also increase, which would align with the reviewer's understanding. On CIFAR10 with $K=10$, the models achieve both well classification accuracy and feature properties near the scale factor $\gamma = 4$. You can refer to the results in Tables 5 and 6 of the supplementary, for ResNet18 and ViT on the training set of CIFAR10. Here we present the results of ResNet18 on the test set.
> > In practice, when the classification accuracy of a trained model is very low, further examination for the properties of its learned features is typically not pursued.
> >
> > **Table R8:** The classification accuracy and feature properties for ResNet18 on the test set of CIFAR10
> > | CE |with |SGD | | | | | | | | | | | | |
> > |:--:|:----:|:----:|:----:|:----:|:----:|:----:|:----:|:----:|:----:|:----:|:----:|:----:|:----:|:----:|
> > | $\gamma$ | 0.1 | 0.5 | 1 | 2 | 3 | 4 | 5 | 6 |7 | 8 | 12 | 16 | 32 | 64 |
> > | $\mathcal{A}$ | 25.81 | 44.18 | 61.54 | 78.95 | 80.24 | 80.69 | 80.16 | 80.07 | 79.49 | 79.76 | 78.72 | 78.91 | 77.84 | 77.46 |
> > | $\mathcal{A}^*$ | 25.62 | 46.94 | 61.60 | 78.94 | 80.24 | 80.68 | 80.16 | 80.08 | 79.49 | 79.76 | 78.71 | 78.91 | 77.79 | 77.44 |
> > | $\mathcal{E}_{com}$ | 0.8887 | 0.8290 | 0.8240 | 0.8209 | 0.8293 | 0.8318 | 0.8232 | 0.8136 | 0.7972 | 0.7823 | 0.7428 | 0.7251 | 0.6908 | 0.6849 |
> > | $\mathcal{E}_{sep}$ | 0.5153 | 0.5347 | 0.5348 | 0.5348 | 0.5358 | 0.5363 | 0.5350 | 0.5336 | 0.5273 | 0.5114 | 0.4159 | 0.3606 | 0.2658 | 0.2044 |
> >
> > ---
> > | BCE |with |SGD | | | | | | | | | | | | |
> > |:--:|:----:|:----:|:----:|:----:|:----:|:----:|:----:|:----:|:----:|:----:|:----:|:----:|:----:|:----:|
> > | $\gamma$ | 0.1 | 0.5 | 1 | 2 | 3 | 4 | 5 | 6 |7 | 8 | 12 | 16 | 32 | 64 |
> > | $\mathcal{A}$ | 24.98 | 44.12 | 58.41 | 76.76 | 81.13 | 82.34 | 82.70 | 82.54 | 82.62 | 81.83 | 81.63 | 82.14 | 81.69 | 82.54 |
> > | $\mathcal{A}^*$ | 26.55 | 45.67 | 59.71 | 76.75 | 81.13 | 82.34 | 82.70 | 82.55 | 82.62 | 81.82 | 81.63 | 82.12 | 81.72 | 82.56 |
> > | $\mathcal{E}_{com}$ | 0.8891 | 0.8244 | 0.8254 | 0.8370 | 0.8324 | 0.8430 | 0.8464 | 0.8455 | 0.8457 | 0.8355 | 0.8226 | 0.8073 | 0.7491 | 0.7261 |
> > | $\mathcal{E}_{sep}$ | 0.5142 | 0.5349 | 0.5355 | 0.5343 | 0.5364 | 0.5375 | 0.5380 | 0.5379 | 0.5378 | 0.5362 | 0.4239 | 0.3680 | 0.3177 | 0.2934 |
> >
> >
> >
> > ---
> > ---
> > 3.	For larger datasets, such as ImageNet or face recognition datasets with more classes, our experience also indicates that larger scale factors/feature spaces are necessary for better training of models to achieve good performance. However, even on datasets like CASIA-WebFace with over ten thousand of identities, a scale factor of 64 is sufficient; we have also experimented with using larger scale factors such as 96 or 128 on CASIA-WebFace and we **did not obtain** better face recognition results.
> > 4.	When the scale factor is large, such as $\gamma = 32, 64$ for CIFAR10 or MNIST, training a model to reach neural collapse indeed requires more epochs, potentially thousands and more, as indicated in the results provided in Sec. W10 of our responses to reviewer **ozw4**. It is for this reason that when we only trained the model for 200 epochs in Table 1, the properties of features learned by the model had not yet achieved optimality.
> > 5.	Our theoretical analysis is established under the premise that the feature dimension $d$ is greater than the number of classes $K$, i.e., within relatively **high-dimensional** feature spaces. When $d<K$, the theoretical analysis becomes a more challenging issue, as mentioned in the previous responses.
> > 6.	Finally, we would like to emphasize again that our paper focuses on comparison for classification tasks and deep feature learning, highlighting their differences in normalized feature spaces with varying scales, rather than seeking the optimal scale for the feature space. Our experimental results consists with our conclusions.

---

### Official Review · Reviewer_ozw4 · 2025-10-30

**Soundness:** 2
**Presentation:** 2
**Contribution:** 1
**Rating:** 2
**Confidence:** 4

**Summary:**

The paper presents Neural Collapse (NC) based analysis (using the unconstrained features model setup) for cross entropy (CE) and Binary CE (BCE) losses and feature normalization (to unit norm). Based on this, the authors make some claims on feature learning.

**Strengths:**

The writing is OK but I have concerns about the novelty, motivation, and relevance for the study of "feature learning".

**Weaknesses:**

1.
Stronger NC, which is typically measured on training data, may degrade generalization, e.g., when the training set is small and/or in case of distribution shift and/or transferability to other tasks. Thus, it is not an indication for good feature learning.

2.
I believe that "intra-class compactness" is not a desirable property for general feature learning, which should be transferable between tasks. Extreme compactness can be good in classification setups where NC is desirable, but then there is no difference between "feature learning" that is supervised and "classification" (which also includes feature learning).

3.
Due to focusing on the unconstrained features model, the analysis does not imply anything on generalization (which is necessary for analyzing feature mappings).

4.
Please provide detailed discussion on the differences between the current work and (Li et al. 2025) ("BCE vs. CE in deep feature learning"), which also analyzed unconstrained features models.

5.
Please point to supervised learning references where the normalization model (z=\gamma Wh-b, with unit ||u||,||w||) has been used exactly.

6.
In feature/representation learning, the goal is to learn good transferable embeddings that would perform well on new tasks/classes. The final layer of the downstream task need not be related to the classes of the samples during the feature learning. Thus, the arguments in Section 3.2 seem wrong, as they assume that the features will only be used for the same classification task.

7.
The motivation for studying the BCE loss should be improved as it does not scale well with the number of classes. Also, as the goal is good feature learning, I suggest contrasting it with supervised contrastive loss and self-supervised approaches.

8.
Theorem 1 seems to be extremely related to existing results in the literature on NC.

9.
Is there technical novelty in the proof of Theorem 2 compared to the proof of Theorem 1?

10.
The discussions below Theorems 5 and 6 are not clear enough. State formally the convergence rates in the different cases.

11.
The experiments section is not satisfactory.
In Table 1 the classification accuracy and the quality of the features should be computed on the test set and not on the train set.
If motivation of the paper is to study "deep feature learning" it should more deeply examine transfer learning, distribution shifts, etc., and compare BCE+normalization with representation/feature learning approaches.

12.
Compare BCE and CE with normalization with BCE and CE with small weight decay instead of normalization.

**Questions:**

Stated above.

---

> ### Author Response · Authors · 2025-11-19
> **Part I**
>
> Thank you for taking the time to review our paper.
>
> ---
> **W1:**
>
> 	Stronger NC, which is typically measured on training data, may degrade generalization, e.g., when the training set is small and/or in case of distribution shift and/or transferability to other tasks. Thus, it is not an indication for good feature learning.
>
> The contributions of our work mainly lie in two aspects:
> 1. We demonstrate that, compared with CE, BCE is more suitable for deep feature learning.
> 2. We provide, for the first time, an in-depth theoretical analysis of the fundamental differences between the tasks of classification and deep feature learning.
>
> These two conclusions will not be changed by the factors such as the size of the training set. Although our arguments and analysis are conducted on the supervised, normalized, and closed-set setting, as we responded to Reviewer **62s4**, our work will profoundly reshape the understanding of deep feature learning and classification.
>
> As mentioned above, our work focuses on theoretical analysis rather than pursuing superior performance on specific tasks. The reviewer raised a question regarding generalization, which is beyond the scope of this paper. Nevertheless, it should be noted that the generalization fundamentally requires that a model first, on the training set, achieves good classification performance and strong feature learning capability.
>
> ---
> **W2:**
>
> 	I believe that "intra-class compactness" is not a desirable property for general feature learning, which should be transferable between tasks. Extreme compactness can be good in classification setups where NC is desirable, but then there is no difference between "feature learning" that is supervised and "classification" (which also includes feature learning).
>
> As we have pointed out in the second paragraph of Introduction (line 40), the desired properties of learned features vary across different scenarios and tasks. However, when the models are pre-trained under a supervised setting using the CE loss, it is often implicitly assumed that the trained models will learn features with good intra-class compactness and inter-class separability.
>
> More importantly, one of the key conclusions of our research is that classification and deep feature learning are not fully aligned in feature spaces with different scaling. In other words, the high intra-class compactness and inter-class separability do not necessarily imply the high classification accuracy, and conversely, the high classification accuracy does not guarantee the good feature compactness and separability. Both our theoretical analyses and experimental results support this conclusion. As discussed in our paper, classification accuracy reflects the local sample-wise property of a model on a given dataset, whereas the compactness and separability of features represent the global, sample-independent properties of the model on the whole dataset. We believe that many researchers, including reviewer ozw4, have not fully grasped this fundamental distinction.
>
> ---
> **W3:**
>
> 	Due to focusing on the unconstrained features model, the analysis does not imply anything on generalization (which is necessary for analyzing feature mappings).
>
> The generalization of models is indeed an important and challenging topic. However, we believe that recognizing the fundamental distinction between the tasks of classification and deep feature learning is also crucial. Although our theoretical analysis is based on an unconstrained feature model (UFM), we believe that the conclusions drawn from our work can inspire a deeper understanding of deep feature learning in other areas, such as self-supervised learning, cross-modal learning, and knowledge distillation. Furthermore, we would like to reiterate that a prerequisite for a model to generalize to other domains is that it must first perform sufficiently well on the training set.
>
> ---
> **W4:**
>
> 	The differences between the current work and [R1].
>
> In the first paragraph of Sec. 6 of our paper and in Sec. D of its supplementary, we have discussed in detail the differences between our work and that in [R1]. Both the two works use the UFM to analyze the minima of CE and BCE losses, demonstrating that both losses can lead to neural collapse and that the BCE is more suitable for deep feature learning compared with CE. The differences are as follows:
> 1. In [R1], the authors analyze the minima of the losses in the Euclidean space and employs regularizations with weight decay, whereas our work conducts the analysis in the normalized feature space with a scaling factor.
> 2. More importantly, we go beyond the work in [R1] by analyzing the convergence behavior of CE and BCE in depth. Based on this analysis, we are the first to provide a detailed theoretical investigation into the differences between classification and deep feature learning.
>
> ---
> ---
> **Reference:**
>
> [R1] Qiufu Li, Huibin Xiao, and Linlin Shen. BCE vs. CE in deep feature learning. ICML 2025.

---

> ### Author Response · Authors · 2025-11-19
> **Part II**
>
> **W5:**
>
> 	Please point to supervised learning references where the normalization model (z=\gamma Wh-b, with unit ||u||,||w||) has been used exactly.
>
> In the supervised learning, normalizing features and classifier vectors has been widely applied in tasks such as face recognition and long-tailed recognition. In face recognition, representative works include CosFace [R2], ArcFace [R3], NormFace [R4], SphereFace [R5], SphereFace2 [R6], and UniFace [R7], etc. In the context of long-tailed recognition, it includes the works of [R8] and [R9], etc.
>
> ---
> **W6:**
>
> 	In feature/representation learning, the goal is to learn good transferable embeddings that would perform well on new tasks/classes. The final layer of the downstream task need not be related to the classes of the samples during the feature learning. Thus, the arguments in Section 3.2 seem wrong, as they assume that the features will only be used for the same classification task.
>
> As the reviewer correctly pointed out, in many cases, researchers are more concerned with the feature learning or representation capability of a trained deep model on downstream tasks, which indeed relates to the issue of generalization. However, it is important to emphasize that generalization presupposes that the deep model first achieves sufficient feature learning or representation capability on the training set. Therefore, with respect to the intra-class compactness and inter-class separability of features, the derivation in Sec. 3.2 of the paper is entirely valid.
>
> ---
> **W7:**
>
> 	The motivation for studying the BCE loss should be improved as it does not scale well with the number of classes. Also, as the goal is good feature learning, I suggest contrasting it with supervised contrastive loss and self-supervised approaches.
>
> We are not sure what exactly you mean by “scale well”. However, in terms of computing loss, BCE has better scalability than CE as the class number $K$ increases. Suppose it computes the BCE and CE losses for a sample $\textbf{X}$. If $K$ increases by one, then for BCE, it only needs to compute the ordinary binary cross-entropy between the sample $\textbf{X}$ and the new classifier vector $\textbf{W}_{K+1}$ and add the result to the previous loss result. Moreover, for this sample, the loss gradient with respect to the new class does not affect those of other classes. In contrast, for CE, it must recompute every term in the Softmax-based function because its denominator has changed; the previous result computed over $K$ classes is completely discarded. Therefore, as $K$ increases, BCE shows better scalability than CE.
>
> Meanwhile, BCE has in fact been applied in many fields, such as image classification [R10, R11, R12, R13], face recognition [R6, R7], long-tailed recognition [R14, R15], self-supervised pretraining [R16], and knowledge distillation [R17], etc. It should be noted that these works involve multi-class tasks rather than the simply binary classification.
>
> At present, we are also working on theoretical analyses of feature learning in supervised and self-supervised contrastive learning, including studying the minima of loss functions with formulation similar to BCE. However, this is not an easy work and goes beyond the scope of this paper, which focuses on comparing classification and feature learning.
>
> ---
> ---
> **Reference:**
>
> [R2] H. Wang, et al. “CosFace: Large Margin Cosine Loss for Deep Face Recognition,” CVPR 2018.
>
> [R3] J. Deng, et al. “ArcFace: Additive Angular Margin Loss for Deep Face Recognition,” CVPR 2019.
>
> [R4] F. Wang, et al. “NormFace: L2 Hypersphere Embedding for Face Verification,” ACM MM 2017.
>
> [R5] W. Liu, et al. “SphereFace: Deep Hypersphere Embedding for Face Recognition,” CVPR 2017.
>
> [R6] Y. Wen, et al. “SphereFace2: Binary Classification is All You Need for Deep Face Recognition,” ICLR 2022.
>
> [R7] J. Zhou, et al. “Uniface: Unified Cross-entropy Loss for Deep Face Recognition,” ICCV 2023.
>
> [R8] J. Shi, et al. “Long-Tail Learning with Foundation Model: Heavy Fine-Tuning Hurts,” ICML 2024.
>
> [R9] B. Kang, et al. “Decoupling Representation and Classifier for Long-Tailed Recognition,” ICLR 2020.
>
> [R10] H. Touvron, et al. “Deit III: Revenge of the ViT,” ECCV 2022.
>
> [R11] R. Wightman, et al. “ResNet Strikes Back: An Improved Training Procedure in Timm,” NeurIPS 2021.
>
> [R12] J. Liu, et al. “TokenMix: Rethinking Image Mixing for Data Augmentation in Vision Transformers,” ECCV 2022.
>
> [R13] H. Wang, et al. “Get the Best of Both Worlds: Improving Accuracy and Transferability by Grassmann Class Representation,” ICCV 2023.
>
> [R14] Y. Cui, et al. “Class-balanced Loss based on Effective Number of Samples,” CVPR 2019.
>
> [R15] Z. Xu, et al. “Learning Imbalanced Data with Vision Transformers,” CVPR 2023.
>
> [R16] Y. Fang, et al. “Corrupted Image Modeling for Self-Supervised Visual Pre-Training,” ICLR 2022.
>
> [R17] Z. Hao, et al. “Revisit the Power of Vanilla Knowledge Distillation: from Small Scale to Large Scale,” NeurIPS 2023.

---

> ### Author Response · Authors · 2025-11-19
> **Part III**
>
> **W8:**
>
> 	Theorem 1 seems to be extremely related to existing results in the literature on NC.
>
> The conclusion of Theorem.1 is indeed similar to that of existing works on neural collapse, as we have noted in lines 272–273 of the paper. However, as we mentioned in our response (**W1**), the main contributions of our work are as follows:
> 1. We demonstrate that, compared with CE, BCE is more suitable for deep feature learning.
> 2. We provide the first in-depth analysis of the differences between classification and deep feature learning.
>
> ---
> **W9:**
>
> 	Is there technical novelty in the proof of Theorem 2 compared to the proof of Theorem 1?
>
> The proof methods of Theorems 1 and 2 are similar, both seeking the constant lower bound of the losses and identifying the necessary and sufficient conditions under which this constant bound is achieved. Through this approach, we demonstrate that the CE and BCE can lead to neural collapse at their minima. Therefore, the proof of Theorem.2 does not involve additional technical innovations compared with Theorem 1. However, we would like to emphasize that the main contribution of our paper does not lie in the novelty of the proof techniques.
>
> ---
> **W10:**
>
> 	The discussions below Theorems 5 and 6 are not clear enough. State formally the convergence rates in the different cases.
>
> The discussion following Theorems 5 and 6 mainly illustrates two points:
> 1. When the size of the normalized feature space is too large, it is suitable for classification but not for deep feature learning.
> 2. When the size of the normalized feature space is too small, it is neither suitable for deep feature learning and even less for classification.
>
> At present, we are not yet able to formally provide the convergence rates of the two losses under different settings. However, we here present a numerical example to directly verify the conclusions of Theorems 5 and 6. We train ResNet18 on MNIST using SGD with a fixed learning rate of $0.01$, comparing the number of epochs required by CE and BCE to reach their minima and exhibit neural collapse. We define the criterion for the losses to reach their minima as follows:
> $$\big|\textbf{w}_k^\top \textbf{h}_i^{(k)}-1\big|<t, \forall k,i,$$
> $$\big|\textbf{w}_j^\top \textbf{h}_i^{(k)}+\frac{1}{K-1}\big|<t, \forall j\neq k,i.$$
> In the criterion, when $\gamma<0.5$, it sets $t=0.001$; when $\gamma \geq5$, $t=0.0035$. The results are shown in Tables R4 and R5.
>
> ---
> **Table R4:** The epoch number when the model reach NC, with various $\gamma$, for CE loss.
> | CE loss | | | | | | | | | | | |
> |:----:|:----:|:----:|:----:|:----:|:----:|:----:|:----:|:----:|:----:|:----:|:----:|
> | $\gamma$ | 0.01 | 0.04 | 0.08 | 0.1 | 0.2 | 0.4 | 6 | 7 | 8 | 9 | 9.5 |
> | Ep. No.| 2743 | 792 | 392 | 303 | 160 | 83 | 77 | 180 | 435 | 1448 | 2083 |
>
> ---
> **Table R5:** The epoch number when the model reach NC, with various $\gamma$, for BCE loss.
> | BCE loss | | | | | | | | | | | |
> |:----:|:----:|:----:|:----:|:----:|:----:|:----:|:----:|:----:|:----:|:----:|:----:|
> | $\gamma$ | 0.01 | 0.04 | 0.08 | 0.1 | 0.2 | 0.4 | 8 | 10 | 12 | 14 | 16 |
> | Ep. No. | 2708 | 822 | 420 | 310 | 163 | 82 | 37 | 83 | 243 | 737 | 2076 |
>
> ---
> When $\gamma$ is very small and approaches $0$, the fitted curves for the number of epochs required by CE and BCE to reach their minima are $$\frac{27.14}{\gamma}+44.41$$ and $$\frac{26.68}{\gamma}+61.50,$$ respectively.
>
> Conversely, when $\gamma$ is large and increases linearly, the fitted curves for CE and BCE are $$0.54 \exp(0.87 \gamma) - 61.83$$ and $$0.49 \exp(0.52\gamma) - 3.79,$$ respectively. These results directly verify Theorems 5 and 6. Moreover, when $\gamma$ is large, the coefficient (0.52) before $\gamma$ in the fitted curve for BCE is smaller than that (0.87) in the fitted curve for CE, indicating that BCE converges faster than CE in the large normalized feature space.
>
> More results can be found in Sec. F in Supplementary of the revised manuscript.

---

> ### Author Response · Authors · 2025-11-19
> **Part IV**
>
> **W11:**
>
> 	The experiments section is not satisfactory.
> 	In Table 1 the classification accuracy and the quality of the features should be computed on the test set and not on the train set. If motivation of the paper is to study "deep feature learning" it should more deeply examine transfer learning, distribution shifts, etc., and compare BCE+normalization with representation/feature learning approaches.
>
> The motivation of our work is to provide an in-depth explanation for the differences between the tasks of classification and deep feature learning. Our main contributions are as follows:
> 1. We demonstrate that, compared with CE, BCE is more suitable for deep feature learning.
> 2. We present, for the first time, an in-depth analysis of the differences between classification and deep feature learning.
>
> The experiments in the paper fully support these conclusions.
>
> As the reviewer noted, comparing the tasks of classification and deep feature learning on the test set is also important. Here, corresponding to the trained models in Table 1 in our paper, on the test set, we provide their classification accuracy, as well as the intra-class compactness and inter-class separability of the features, in Table R6. From the table, one can observe that when $\gamma$ is small (e.g., $0.5$ or $0.1$), the models’ classification is poor, but the intra-class compactness and inter-class separability of the features are higher than $0.8$ and $0.5$, respectively. Conversely, when $\gamma$ is large (e.g., $32$ or $64$), the models achieve higher classification accuracy, but the feature properties significantly degrade. These results demonstrate that the differences between the tasks of classification and deep feature learning, as analyzed on the training set, also generalize to the test set.
>
> ---
> **Table 6:** The classification accuracy and feature properties for ResNet18 on the test set of CIFAR10
>
> | CE with SGD | | | | | | | |
> |:--:|:----:|:----:|:----:|:----:|:----:|:----:|:----:|
> | $\gamma$ | 0.1 | 0.5 | 1 | 12 | 16 | 32 | 64 |
> | $\mathcal{A}$ | 25.81 | 44.18 | 61.54 | 78.72 | 78.91 | 77.84 | 77.46 |
> | $\mathcal{A}^*$ | 25.62 | 46.94 | 61.60 | 78.71 | 78.91 | 77.79 | 77.44 |
> | $\mathcal{E}_{com}$ | 0.8887 | 0.8290 | 0.8240 | 0.7428 | 0.7251 | 0.6908 | 0.6849 |
> | $\mathcal{E}_{sep}$ | 0.5153 | 0.5347 | 0.5348 | 0.4159 | 0.3606 | 0.2658 | 0.2044 |
> ---
> | BCE with SGD | | | | | | | |
> |:--:|:----:|:----:|:----:|:----:|:----:|:----:|:----:|
> | $\gamma$ | 0.1 | 0.5 | 1 | 12 | 16 | 32 | 64 |
> | $\mathcal{A}$ | 24.98 | 44.12 | 58.41 | 81.63 | 82.14 | 81.69 | 82.54 |
> | $\mathcal{A}^*$ | 26.55 | 45.67 | 59.71 | 81.63 | 82.12 | 81.72 | 82.56 |
> | $\mathcal{E}_{com}$ | 0.8891 | 0.8244 | 0.8254 | 0.8226 | 0.8073 | 0.7491 | 0.7261 |
> | $\mathcal{E}_{sep}$ | 0.5142 | 0.5349 | 0.5355 | 0.4239 | 0.3680 | 0.3177 | 0.2934 |
>
> ---
> **W12:**
>
> 	Compare BCE and CE with normalization with BCE and CE with small weight decay instead of normalization.
>
> When normalization is not applied and weight decay is instead used to indirectly constrain the norms of classifier vectors and sample features, the size of the feature space cannot be predetermined. This makes it difficult to analyze the convergence behavior of the two losses and thus to compare classification and deep feature learning.
>
> Here, we conduct a set of experiments to compare the performance of CE and BCE when using normalization versus weight decay. Using normalization or regularization with weight decay, we train ResNet18 and ResNet50 on the CIFAR10 and CIFAR100 using the SGD for 200 epochs, with an initial learning rate of 0.1 that decays by a factor of 10 at the 100th and 150th epochs. The results are presented in Table R7. It can be observed that models trained using normalization generally achieve higher classification accuracy compared with those trained using weight decay.
>
> ---
> **Table R7:** Comparison between losses with normalization and weight decay.
>
> | | | CIFAR10 | CIFAR10 | CIFAR100 | CIFAR100 |
> |:--------:|:----:|:--------:|:--------:|:--------:|:--------:|
> | | Loss | $\gamma=32$ | $\lambda=1e-4$ | $\gamma=32$ | $\lambda=1e-4$ |
> | ResNet18 | CE | **95.90**±0.071 | 94.47±0.159 | **78.39**±0.190 | 74.59±0.595 |
> |ResNet18 | BCE | **95.97**±0.068 | 94.86±0.142 | **78.46**±0.132 | 75.36±0.406 |
> | ResNet50 | CE | **96.30**±0.245 | 94.49±0.145 | **80.89**±0.266 | 74.60±0.493 |
> |ResNet50 | BCE | **96.40**±0.086 | 94.88±0.147 | **80.87**±0.277 | 69.67±1.022 |

---

> > ### Comment · Reviewer_ozw4 · 2025-11-27
> >
> > I thank the authors for their response, but my concerns about the novelty, motivation, and the relevance to actual "feature learning" (for which transferability must be considered) remain the same.
> > Therefore, I have decided to keep my score unchanged.

---

> > > ### Author Response · Authors · 2025-11-27
> > >
> > > We understand the reviewers' concerns. However, at this time, it indeed goes beyond the scope of theoretical comparisons between classification and deep feature learning in our paper.

---

### Official Review · Reviewer_62s4 · 2025-10-31

**Soundness:** 2
**Presentation:** 3
**Contribution:** 2
**Rating:** 4
**Confidence:** 4

**Summary:**

This paper gives a neat, self-contained theoretical and empirical study of the gap between classification and deep feature learning (DFL) in a very specific regime: supervised, normalized, closed-set classification under UFM-style assumptions. The central idea is to decouple two objectives that are often conflated: (i) achieving nearly perfect, sample-wise local classification, and (ii) achieving a global feature geometry (NC and ETF-like) that enforces intra-class compactness and inter-class separability. To make this separation concrete, the paper associates standard CE with the local objective and reformulates BCE as a better proxy for the global DFL objective, arguing that, unlike CE’s shift-invariant bias, the BCE bias has a unique, substantial optimum that actually shapes the learned geometry.

A key contribution is the analysis of the scaling factor $\gamma$: the paper shows that as $\gamma$ increases linearly, the convergence toward the desired NC geometry decays exponentially. This creates a tension: large $\gamma$ can drive classification to 100% but stall feature-quality convergence; very small $\gamma$ improves geometric convergence but hurts accuracy. Experiments suggest a moderate $\gamma$ balances the two, and BCE-trained features transfer better to long-tailed recognition and open-set recognition.

**Strengths:**

- The paper clearly separates two confused goals: getting samples classified correctly vs. learning a good feature geometry.
- Within the stated regime (supervised, normalized, closed-set, UFM), the analysis is coherent, and the CE/BCE/NC/ETF relationships are tied together convincingly.
- The role of the scaling factor $\gamma$ is articulated concretely, showing how increasing $\gamma$ creates a trade-off by speeding up classification but slowing convergence to the NC/ETF geometry.
- The paper makes a useful point that, in this setup, the BCE bias is not incidental but actually shapes the feature geometry.

**Weaknesses:**

## 1. The applied scope is narrow and should be made explicit
The core Theorems [1-6] are all derived under a stylized configuration: fully supervised learning, a closed label set of size $K$, and feature and classifier normalization. This is essentially the NC (Neural Collapse) theory world. In such a context, using ETF-like target geometries and measuring feature–weight alignment is legitimate. However, today's deep feature learning is primarily driven by contrastive self-supervised methods, which aim for open-set vocabularies, class-weight ($w$)- agnostic batches, and sample-sample semantic alignment. These regimes do not satisfy the paper's core assumptions. The current paper sometimes slides from “we showed this under UFM + fixed-\(K\) + normalization” to “therefore this is how deep feature learning behaves”. This should be tightened. A fairer statement would be:
> “We provide a self-contained analysis for normalized, supervised, closed-set classification with fixed $K$; outside this regime, the behavior may differ.”

## 2. The advantage of bias in BCE can act as a constraint
This paper argues that BCE is superior to CE because the bias $b$ in BCE is substantial and unique, while the bias in CE is somehow useless due to its shift-invariance in softmax. However, we can have some opposite opinions:
- The shift-invariance of bias in CE can be a robustness property that makes the model not over-sensitive to the absolute scale of logits;
- The biases in BCE (Eq. 11) are a clean object only because we are in the aforementioned stylized settings. It is also tightly tied to the training-time number of classes $K$;
Therefore, the same thing the paper calls an advantage can also be read as a strong label-dependent constraint. This constraint removes the translation robustness of CE and locks the optimal geometry to the training value of $K$ in BCE in class-agnostic regimes.

## 3. On the "exponential decay" narrative
This paper claims that, for large $gamma$, the convergence rate decays exponentially; therefore, the model cannot reach the desired NC/ETF geometry. This is one possible reading, but there is also a very natural alternative: a large $gamma$ makes the optimization goal into a simpler one. Since it can make the data separably classified with high confidence ($P \to 1$), SGD can quickly succeed at this easier goal. Once this easy goal is reached, gradients necessarily get tiny (since $1-P \to 0$). What we observe as “exponential decay” is a symptom of success on the easy goal, not a cause of failure on the hard geometric goal. In other words, the paper currently treats a post-separability slowdown as evidence that "large $gamma$ harmed feature learning". However, a post-separability slowdown is expected once the “critical conditions” (I (Eq. 14) and II (Eq. 15) in the paper) are satisfied. Those conditions are themselves local separability conditions, inducing slower dynamics.


## 4. The Math Issue
To begin with, I'd like to say that the theoretical section would benefit from tightening the assumptions and motivating the key inequality from the problem structure, rather than presenting a long algebraic chain.

The most serious mathematical issue in this paper appears in the BCE part of the appendix, where the authors reuse an AM–GM type inequality but violate its own stated precondition. In Appendix E.2, the authors attempt to derive a lower bound for the BCE loss by invoking the standard AM–GM type inequality: $u^\top v \le \frac{c}{2}\|u\|_2^2 + \frac{1}{2c}\|v\|_2^2, \text{for } c > 0.$ They explicitly require $c > 0$ for this inequality to hold.

However, in the tightness part of the argument, where they try to make the inequality achieve equality and at the same time recover the NC/ETF-like structure, the derivation effectively enforces $h_i^{(k)} = - c_4 \gamma w_k$, and, combined with the target condition $h_i^{(k)} = w_k$, this forces $c_4 = -\frac{1}{\gamma} < 0$. That is, to make the proof work, the authors end up choosing a negative value for the constant, which must be positive according to the inequality they cited. This is a direct violation of the premise $c > 0$. As a result, the corresponding lemma (and the theorem that depends on it) does not currently have a valid proof in the appendix.

**Questions:**

- The theory in this paper is derived in a strict NC-style setting (supervised, normalized, closed-set, fixed $K$). What is the concrete insight or usefulness of your results for current self-supervised contrastive feature learning, which does not satisfy these assumptions?
- The most serious issue is the one in weakness #4. This breaks the stated condition and renders Lemma 12 invalid as written. Please clarify or fix it.
- Please also address the other weaknesses.

One tricky point is that, under today’s dominant self-supervised contrastive deep feature learning, the practical value of this paper’s classification-based setup is limited. That said, given ICLR’s openness to theory, I am willing to accept the authors’ chosen supervised closed-set feature learning framework; within this framework, the result that BCE can be preferable to CE is meaningful. However, the mathematical issue in Weakness #4 (violation of the inequality’s own precondition) affects Lemma 12 and is critical. I therefore currently place the paper below the acceptance threshold. If the authors can fix this issue, and if other reviewers are comfortable with the restricted learning paradigm, I would be happy to raise my score.

---

> ### Author Response · Authors · 2025-11-19
> **Part I**
>
> Thank you for reviewing our work and providing valuable feedback.
>
> ---
> **W1. The applied scope is narrow and should be made explicit.**
>
> As the reviewer pointed out, the deep feature learning discussed in our manuscript is limited in the supervised, closed-set, and normalized setting. We have roughly **revised** the manuscript according to your suggestions to clarify the scope of our work.
>
> Although the recent researches in deep feature learning has expanded into areas such as self-supervised, cross-modal, and open-set settings, the deep feature learning with supervised, closed set remains frequently adopted in practice. For example, in various application fields, such as in face recognition, researchers still perform feature learning via supervised classification on closed sets, or conduct supervised pre-training on large-scale labeled datasets like ImageNet. Although the pre-trained models may face new categories during testing or when transferred to downstream tasks, their supervised upstream pre-training typically aims to strengthen their feature-extraction capability by enhancing classification performance.
>
> However, the classification performance reflects the sample-wise local property of a model on a dataset, while the intra-class compactness and inter-class separability represent the global sample-independent properties of the model on the dataset. Currently, there has to date been no published work offering a rigorous analysis of the distinction between the tasks of classification and deep feature learning. This motivates our comparison of the two tasks in the setting considered in our paper; to the best of our knowledge, this is the first attempt to theoretically compare them.
>
> ---
> **W2. The advantage of bias in BCE can act as a constraint.**
>
> When training deep models using gradient descent-based methods, the classifier biases in BCE indeed provide a substantial constraint which benefits the deep feature learning. According to the derivation in Sec. 3.2, during the minimization of BCE, the classifier biases could explicitly constrain the positive and negative decision scores of all the samples, while these decision scores reflect the intra-class compactness and inter-class separability of the features. In Eq.(11), when the number of classes $K$ changes, the unique extremum of the classifier biases in BCE will also change. However, during the model training, their substantive constraint on the feature learning remains and does not vanish as $K$ varies. In addition, in Euclidean space, the authors of [R1] compares CE and BCE and points out that the classifier biases in BCE substantially promote the deep feature learning.
>
> The classifier biases in CE exhibit shift-invariance, which provides an advantage in classification and makes the model less sensitive to the absolute scale of the logits (i.e., the decision scores). However, the shift-invariance does not offer a direct benefit for the deep feature learning. As described in Sec. 3.1 and 3.2, classification performance depends on the relative values of each sample’s positive and negative decision scores (logits), while the compactness and separability of features depend on the absolute values of all decision scores for all the samples. Therefore, in general classification task, BCE may not outperform CE [R2]. Nevertheless, in tasks that require stronger feature properties, such as face recognition [R3] and long-tailed recognition [R4], BCE tends to perform better than CE. Our experiments also confirmed this expectation.
>
> ---
> ---
> **Reference:**
>
> [R1] Q. Li, H. Xiao, L. Shen, “BCE vs. CE in Deep Feature Learning,” ICML 2025.
>
> [R2] R. Wightman, H. Touvron, and H. Jegou, “ResNet Strikes Back: An Improved Training Procedure in Timm,” NeurIPS Workshop, 2021.
>
> [R3] J. Zhou, X. Jia, Q. Li, L. Shen, J. Duan, “UniFace: Unified Cross-entropy Loss for Deep Face Recognition,” ICCV 2023.
>
> [R4] Z. Xu, R. Liu, S. Yang, Z. Chai, and C. Yuan, “Learning imbalanced data with vision transformers,” CVPR 2023.

---

> ### Author Response · Authors · 2025-11-19
> **Part II**
>
> **W3. On the “exponential decay” narrative.**
>
> We believe that the “high confidence” mentioned by the reviewer refer to the confidence level of classifying each individual sample. Compared with achieving the optimal feature structure (i.e., reaching the NC/ETF geometry), correctly classifying all samples with high confidence is indeed a relatively easier goal. Therefore, even after achieving 100% classification accuracy, a long training period is still required to reach the NC/ETF geometry. This phenomenon has been discussed in neural collapse [R5, R6].
>
> As the reviewer pointed out, once the easier objectives (i.e., the critical conditions I and II) are achieved, it naturally induces slower dynamics. However, the extent to which this dynamic slows down remains unknown. According to Theorems 5 and 6, when $\gamma$ increases linearly, the updating amplitude of the unbiased decision scores decays exponentially, which leads to an exponential reduction in the dynamics, and consequently, the convergence rate of the losses also decays exponentially. Thus, using the same training setting, although a larger $\gamma$ does not directly harm feature learning, it indeed makes it increasingly difficult for the model to reach the limiting state with NC/ETF geometry.
>
> In Fig. 1 and Table 1 in the manuscript, the maximum training epoch is fixed at 200; they present the distributions of unbiased positive and negative decision scores corresponding to different $\gamma$ values after the model training, as well as the final classification accuracy and the feature properties. One can find that using the same training strategy, the larger $\gamma$ lead to the poorer final feature properties.
>
> To more directly illustrate the impact of different $\gamma$ on the deep feature learning, taking the NC/ETF geometry as the stopping criterion, we record the epoch number for models with different $\gamma$ when reaching the stopping criterion. We set the NC/ETF geometry/stopping criterion as that the unbiased positive and negative decision scores of all samples are within 0.0035 of their respective extrema, i.e.,
> $$\big|\textbf{w}_k^\top \textbf{h}_i^{(k)} - 1\big| < 0.0035,\quad \forall k,i,$$
> $$\big|\textbf{w}_j^\top \textbf{h}_i^{(k)} + \frac{1}{K-1}\big| < 0.0035,\quad \forall j\neq k,i.$$
> We fix the learning rate at $0.01$ and train ResNet18 on the MNIST. Tables R2 and R3 show the epoch number for different values of $\gamma$ when using CE and BCE, respectively. We then fit the curves of epoch number with respect to $\gamma$, obtaining the following functions: for CE,
> $$0.54e^{0.87\gamma}-61.83,$$
> and for BCE,
> $$0.49e^{0.52\gamma}-3.79.$$
> These results indicate that as $\gamma$ increases, the number of epochs required for both CE- and BCE- trained models to reach the NC/ETF geometry grows exponentially, which means their convergence rates decay exponentially.
>
> ---
> **Table R2.** The epoch number when the model reach NC, with various $\gamma$, for CE loss.
> | CE loss | | | | | |
> |:----:|:----:|:----:|:----:|:----:|:----:|
> | $\gamma$ | 6 | 7 | 8 | 9 | 9.5 |
> | Ep. No. | 77 | 180 | 435 | 1448 | 2083 |
>
> ---
> **Table R3.** The epoch number when the model reach NC, with various $\gamma$, for BCE loss.
> | BCE loss | | | | | |
> |:----:|:----:|:----:|:----:|:----:|:----:|
> | $\gamma$ | 8 | 10 | 12 | 14 | 16 |
> | Ep. No.| 37 | 83 | 243 | 737 | 2076 |
>
> ---
> ---
> **W4. The math issue.**
>
> We sincerely thank the reviewer for pointing out the error in our theorem proof. **We have revised the proof of Lemma 12**, and the revised proof avoids solving for a specific $c_4$; thus, the derivation no longer violates the precondition. In the revised manuscript, the revised proof is highlighted in blue. Fortunately, the new proof does not alter the conclusion of Lemma 12, and therefore **Theorem 2 remains valid**.
>
> In the theorem proof, some of the inequality derivations are indeed lengthy. However, we believe that providing detailed derivations will help readers who are interested in the theoretical analysis of neural collapse to better understand and verify our proof. We would like to once again thank the reviewer for helping us improve the theorem.
>
> ---
> ---
> **Reference:**
>
> [R5] V. Papyan, X. Han, D. L. Donoho, “Prevalence of Neural Collapse During the Terminal Phase of Deep Learning Training,” PNAS 2020.
>
> [R6] C. Fang, H. He, Q. Long, W. J. Su, “Exploring Deep Neural Networks via Layer-peeled Model: Minority Collapse in Imbalanced Training,” PNAS 2021.

---

> ### Author Response · Authors · 2025-11-19
> **Part III**
>
> **Q1. The insight for self-supervised contrastive learning and knowledge distillation.**
>
> Although our analysis for deep feature learning is constrained to the supervised, normalized, and closed-set setting, we believe that the theoretical analysis presented in the paper can inspire understanding of feature learning in other scenarios, such as self-supervised contrastive learning [R7, R8], and even knowledge distillation [R9]. In the case of self-supervised contrastive learning [R10], it has been shown that when the Softmax-based contrastive losses reach their minima, the features of different views or augmentations of the same sample converge to an identical feature vector, that is, the similarity between positive sample features converges to a fixed value. Similar to the analysis in our Theorems 5 and 6, a smaller temperature coefficient $\tau$ (i.e., a larger scaling factor $\gamma$) would exponentially decay the convergence rate of the similarity. In other words, when using a fixed training strategy, a too small temperature coefficient would leads to suboptimal contrastive learning. This is only a preliminary conjecture, as it currently lacks rigorous theoretical proof or experimental validation. In practice, the temperature coefficient $\tau$ in self-supervised contrastive learning tasks is usually around 0.07.
>
> For knowledge distillation, we have not found theoretical analyses addressing the minima of its loss function (which is typically based on the KL divergence). However, we conjecture that if the student network has sufficient learning capacity, then when the distillation loss reaches its minimum, the features extracted by the student network for any given sample will totally align with those extracted by the teacher network. In this case, different temperature coefficients $\tau$ would similarly affect the convergence rate of the distillation loss.
>
> In practice, the temperature coefficient $\tau$ used in distillation is often around 4, which differs significantly from the value used in contrastive learning. We further conjecture that this difference arises because, in knowledge distillation, the outputs of the teacher and student networks are typically not normalized, meaning their feature space has unbounded size. A temperature coefficient greater than $1$ thus serves to scale down the feature space to an appropriate size and adjust the update amplitude of the feature similarity, enabling the student network to achieve good feature learning capability and generalization after the distillation. In contrast, in self-supervised contrastive learning, the sample features were normalized onto a unit hypersphere. In this case, a temperature coefficient less than $1$ expands the feature space to an appropriate size and correspondingly adjusts the update amplitude of feature similarity, allowing the model to achieve good feature extraction ability and generalization performance after the training.
>
> The above analysis is our preliminary conjecture and lacks solid theoretical analysis or experimental verification at this stage. However, as indirect evidence, the authors of [R11] selected a temperature coefficient smaller than $1$ in their distillation setting, when the teacher and student network outputs are both normalized.
>
>
> ---
> ---
> **Reference:**
>
> [R7] K. He, H. Fan, Y. Wu, S. Xie, and R. Girshick, “Momentum Contrast for Unsupervised Visual Representation Learning,” CVPR 2020.
>
> [R8] T. Chen, S. Kornblith, M. Norouzi, G. Hinton, “A Simple Framework for Contrastive Learning of Visual Representations,” ICML 2020.
>
> [R9] G. Hinton, O. Vinyals, and J. Dean. “Distilling the Knowledge in a Neural Network,” arXiv preprint arXiv:1503.02531, 2015.
>
> [R10] P. Koromilas, G. Bouritsas, T. Giannakopoulos, M. Nicolaou, and Y. Panagakis, “Bridging Mini-batch and Asymptotic Analysis in Contrastive Learning: From InfoNCE to Kernel-based Losses,” ICML 2024.
>
> [R11] C. Yang, Z. An, L. Huang, J. Bi, X. Yu, H. Yang, B. Diao, Y. Xu, “CLIP-KD: An Empirical Study of CLIP Model Distillation,” CVPR 2024.

---

> ### Author Response · Authors · 2025-11-19
> **Part IV**
>
> **Q2&Q3.**
>
> We have revised the proof of Lemma 12 and have addressed the weaknesses pointed out by the reviewer as thoroughly as possible. If the reviewer still has any concerns, we would be very happy to discuss them further.
>
> We also noticed that reviewers **ozw4** and **bo89** encouraged us to explore the deep feature learning in other scenarios, such as supervised contrastive learning or feature learning in low-dimensional spaces where the feature dimension is smaller than the number of classes (i.e., $d < K – 1$). However, these two reviewers did not recognize that the main focus of our work lies in distinguishing between the tasks of classification and deep feature learning. As you have mentioned, our work “clearly separates two confused goals: getting samples classified correctly vs. learning a good feature geometry.” To the best of our knowledge, this is the first work that rigorously compare classification and deep feature learning through theoretical analysis. Although our study is limited to the supervised, normalized, and closed-set setting, we believe it provides valuable insights that can help researchers better understand feature learning in broader contexts.
>
> Furthermore, according to the data we provided in our response to reviewer **ozw4**, the conclusions we obtained regarding classification and deep feature learning in the supervised, closed-set setting can generalize well from the training set to the test set.

---

> ### Author Response · Authors · 2025-12-03
> **The Math Issue**
>
> We conducted a stringent review of the theoretical derivation process. We found subtle logical flaws in the proof of an earlier revised version. **We have rederived the proofs to complete our theoretical justification and have uploaded a new revised manuscript**. We would like to thank you for your acknowledgment of our work and your constructive feedback.

---

### Official Review · Reviewer_pBJ8 · 2025-10-31

**Soundness:** 3
**Presentation:** 3
**Contribution:** 3
**Rating:** 8
**Confidence:** 3

**Summary:**

The authors analyze cross entropy (CE) and binay cross entropy (BCE)
in normalized spaces for classification and feature learning.  In
normalized spaces, both feature (h) and weight vectors (w_j) are
normalized with norm 1.  The decision vector (z) is adjusted with a
scaling factor (gamma), where z = gamma*Wh - b.  For classification,
class probabilities are obtained via softmax and the loss function is
cross entropy (CE).  For feature learning, intra-class compactness and
inter-class separation are considered.  For inter-class compactness,
min{w_k . h_i} is greater than some threshold.  Similarly for
inter-class separation, max{w_j . h_i} is less than some threshold.
Binary Cross Entropy (BCE) is used as is in multi-class classification.

Based on a number of assumptions, the theoretical analysis indicates
that normalized CE and BCE lead to neural collapse (NC) when minimum
is achieved.  The CE loss cannot enhance compactness and separability
and has many minima.  However, normalized BCE, which incorporates
compactness and separability, has only one minimum.

For normalized spaces, another analysis shows that when gamma is very
large, classification performs well while feature learning performs
poorly.  When gamma is very small, both classification and feature
learning perform poorly.

For empirical analysis, they use 2 existing network architecture over
3 datasets.  The use existing NC metrics to measure NC progress.  When
gamma is 8, the empirical results indicate that both BCE and CE can
lead to NC when they reach the minimum.  On varying gamma values, both
CE and BCE perform poorly when gamma is less than 1.  Both CE and BCE
performs well when gamma is larger than 1.  When gamma < 8,
convergence can be achieved.  As expected BCE generally has better
compactness and convergence.  On long-tailed recognition and open-set
recognition, BCE performs better than CE.

**Strengths:**

1.  Analyzing CE and BCE in normalized spaces is interesting.

2.  The theoretical analysis shows that CE and BCE under certain
assumptions can lead to NC, and larger gamma is useful for
classification.

3.  The empirical analysis indicates that the results roughly follow
the theoretical analysis.

**Weaknesses:**

1.  Since BCE explicitly incorporates compactness and separability,
BCE has better feature learning is expected.

**Questions:**

p8: "In contrast, when gammaγ = 32 or 64, the models’ accuracies reach
100%, while the intra-class compactness and inter-class separability
of their features are comparatively poor and worse than that learned
with small" Why compactness and separability did not further improve?
Is it because convergence was not achieved?

How would BCE perform without incorporating compactness and separability?

---

> ### Author Response · Authors · 2025-11-19
>
> Thank you for your recognition and support of our work.
>
> ---
> **W1: Since BCE explicitly incorporates compactness and separability, BCE has better feature learning is expected.**
>
> It aligns with our expectations that BCE exhibits superior feature learning compared to CE. We have supported this expectation through theoretical analysis and experimental validation, establishing a theoretical foundation for the application of BCE in image classification [R1, R2, R3, R4], face recognition [R5], and long-tailed recognition [R6].
> Moreover, by analyzing the convergence behavior of CE and BCE within the normalized feature space, we provide the first rigorous comparative analysis of classification and feature learning tasks.
>
> ---
> **Q1: Why compactness and separability did not further improve when gamma = 32 or 64? Is it because convergence was not achieved?**
>
> No, the CE and BCE losses did not converge in our experiments when $\gamma=32$ or $64$. The experimental results in Table 1 were obtained by training ResNet18 on CIFAR10 under the same training setting, specifically using SGD for 200 epochs, with all other hyperparameters kept constant across different values of $\gamma$.
>
> According to our Theorems 5 and 6, as $\gamma$ increases, the convergence rates of the two losses decay exponentially. Consequently, for the losses to converge and reach their minima, the number of epochs required will increase exponentially with $\gamma$ increases. Therefore, with the training fixed at 200 epochs, a too large $\gamma$ (such as $32$ or $64$) results in non-convergence of the losses. More results for Theorems 5 and 6 are presented in our responses to reviewers **62s4** and **ozw4**.
>
> ---
> **Q2: How would BCE perform without incorporating compactness and separability?**
>
> We speculate that you are asking how BCE performs when it does not incorporate the classifier biases that explicitly constrains the compactness and separability.
>
> Currently, we have not strictly derived whether the minimization of BCE without classifier biases can still lead to neural collapse. However, based on intuitive understanding, the absence of classifier biases will substantively diminish the performance of BCE in deep feature learning and classification tasks. For example, in [R5] (Table C in its supplementary), the authors compared the performance of BCE in facial recognition, with and without the classifier biases. Their results show that with biases, BCE significantly outperforms the results obtained without biases in facial recognition.
>
> Furthermore, with and without biases, we take BCE to train ResNet18 and ResNet50 on CIFAR10 and CIFAR100, respectively, and evaluate the trained models on the test set. The results are shown in Table R1. It can be observed that the classification accuracy of BCE with biases consistently surpasses that of the one without biases.
>
> **Table R1.** Classification of models trained by BCE with and without classifier biases.
> | | CIFAR10 | CIFAR10 | CIFAR100 | CIFAR100 |
> |:----------:|:----------:|:----------:|:----------:|:----------:|
> |    | w/ | w/o | w/ | w/o |
> | ResNet18 | **95.97**±0.068 | 95.65±0.173 | **78.46**±0.132 | 77.98±0.264 |
> | ResNet50 | **96.40**±0.086 | 96.10±0.193 | **80.87**±0.277 | 79.25±0.742 |
>
> ---
> ---
> **Reference:**
>
> [R1] H. Touvron, M. Cord, H. Jegou. “Deit III: Revenge of the ViT,” ECCV 2022.
>
> [R2] R. Wightman, H. Touvron, H. Jegou. “ResNet Strikes Back: An Improved Training Procedure in Timm,” NeurIPS 2021.
>
> [R3] J. Liu, B. Liu, H. Zhou, et al. “TokenMix: Rethinking Image Mixing for Data Augmentation in Vision Transformers,” ECCV 2022.
>
> [R4] H. Wang, Z. Li, W. Zhang. “Get the Best of Both Worlds: Improving Accuracy and Transferability by Grassmann Class Representation,” ICCV 2023.
>
> [R5] Jiancan Zhou, Xi Jia, Qiufu Li, Linlin Shen, and Jinming Duan. “UniFace: Unified cross-entropy loss for deep face recognition,” ICCV 2023.
>
> [R6] Zhengzhuo Xu, Ruikang Liu, Shuo Yang, Zenghao Chai, and Chun Yuan. “Learning imbalanced data with vision transformers,” CVPR 2023.

---

> ### Author Response · Authors · 2025-12-03
> **Convergence Behavior Without Bias**
>
> We carefully examined the theoretical proof process and confirmed that **without the bias term, the BCE loss cannot achieve neural collapse/ETF structure**, and the learned feature properties are not as good as those with the bias term. It fails to maximize inter-class separability, further confirming **the substantial constraining effect of the classifier bias on feature learning in BCE loss**. However, the positive and negative decision scores obtained during training without the bias still converge to a fixed value. We leave it for future research to explore how this phenomenon affects tasks that demand high requirements for feature properties, such as long-tail recognition and open-set recognition.

---

### Author Response · Authors · 2025-12-03
**Authors' Official Summary on Submission 1265**

We would like to express our gratitude to all the reviewers for their reviews of our work, their constructive feedback, and their recognition of our research on the differences between BCE and CE losses under normalization constraints in the phenomenon of neural collapse. We are pleased that the reviewers noted that the paper “clearly distinguishes between the two confounding objectives of sample classification and good feature geometry,” has “thorough empirical analysis,” and is “well-written and well-organized.”

In light of the changes in the review process, we would like to **reiterate a few contributions of this work and resubmit a revised manuscript**.

(1) We have rephrased the BCE Loss from the perspective of feature learning and theoretically compared the performances of CE and BCE in the normalized space, demonstrating that BCE Loss can also lead to Neural Collapse. During model training, the classifier bias of BCE plays a significant role in feature learning, while that of CE does not. This substantive difference enables BCE loss to achieve better results than CE in long-tail recognition and open-set recognition.

(2) We conducted an in-depth analysis of the convergence behavior of CE and BCE as the scaling factor $\gamma$ changes, **revealing the differences between classification and deep feature learning**. As $\gamma$ increases linearly, the convergence rates of the two losses may exponentially decay. Therefore, when $\gamma$ is very large within a large normalized space, classification performs well while feature learning performs poorly. Conversely, as $\gamma$ linearly decreases towards zero, the convergence rates decrease linearly, and it is unsuitable for deep feature learning and even less suitable for classification in a very small space.

(3) The experimental results indicate that when the scaling factor $\gamma$ is large, the models achieve satisfactory classification accuracy on the training set, but the feature properties are relatively poor. When $\gamma$ is very small, the model's feature properties are unsatisfactory but superior to those at larger $\gamma$, while classification performance is very poor. When $\gamma$ is at a moderate level, both the classification and feature properties of the models can reach optimal points.

---

### Meta-Review · Area_Chair_Ufha · 2026-01-07

**Summary:**

This paper presents a theoretical and empirical study comparing CE and BCE losses in normalization space, and distinguishes classification performance and deep feature learning (DFL).  A central contribution is the analysis of how the scaling factor $\gamma$ influences convergence behavior, revealing a trade-off in which large $\gamma$ favors classification accuracy but slows convergence toward NC, while very small $\gamma$ harms both tasks. Reviewers generally acknowledged that the paper is clearly written.
Some reviewers appreciated the clear distinction between sample-wise classification performance and global feature geometry, and found the convergence analysis with respect to the scaling factor $\gamma$ to be insightful.

However, the reviewers also raised substantial concerns regarding the paper’s novelty relative to prior work, the narrowness of its theoretical and empirical scope, the debatable use of neural collapse as a proxy for *good* feature learning, etc. Overall, the weaknesses were judged to outweigh the strengths, and AC regards that the paper does not meet the acceptance bar for ICLR.

**Reviewer Concerns:**

### Addressed
**Clarification of scope (62s4):**
The authors clarified that the theoretical analysis is restricted to supervised, normalized, closed-set classification under NC-style assumptions, and revised the manuscript to avoid over-general claims about deep feature learning in general.

**Behavior at large $\gamma$ and convergence (pBJ8, 62s4):**
The authors explained that poor compactness/separability at large $\gamma$ is due to non-convergence under a fixed training budget, and provided additional experiments showing that the number of epochs required to reach NC grows exponentially with $\gamma$.

**Role of classifier bias in BCE (pBJ8, 62s4):**
Additional experiments and theoretical discussion were provided to show that BCE without bias cannot achieve NC/ETF, enhancing the claim that BCE’s bias plays a substantive role in shaping feature geometry.

**Mathematical correctness of Lemma 12 (62s4):**
The authors acknowledged the flaw in an earlier proof pointed out by Reviewer 62s4. They revised Lemma 12 and later re-derived the proof in a further revision to avoid violating the inequality precondition. This addresses the formal correctness concern, assuming the revised proof is valid.

**Missing experimental comparisons (ozw4):**
The authors added test-set evaluations and experiments comparing normalization with weight decay, partially strengthening the empirical section.

**Writing issues and ambiguity (bo89, ozw4):**
Several phrasing issues were corrected (e.g., avoiding the impression that BCE is newly proposed), and additional references to prior normalized-logit work were added.

==========================================

### Outstanding concerns

**Novelty relative to prior work (ozw4, bo89):**
Multiple reviewers remained unconvinced that the core insights go substantially beyond prior analyses of BCE vs. CE and neural collapse (e.g., Li et al. 2025, NC-inducing losses, hyperspherical classifiers). While the normalized-space and convergence-rate analysis adds nuance, the contribution is viewed by some as incremental or reframing rather than fundamentally new. Especially given that Li et al. 2025 also derive similar conclusions that "We first theoretically prove that BCE can maximize the compactness and distinctiveness when reaching its minimums. Then, through in-depth analysis of the decision scores in the training, we explain that BCE can better enhance the compactness and distinctiveness of sample features in practical training."

**Interpretation of neural collapse as DFL indicator (ozw4):**
Strong NC on the training set does not necessarily imply transferable or robust representations. However, the DFL is concerned about generalization. The authors largely treated generalization, transfer learning, and distribution shift as out of scope, leaving this concern unaddressed. In Table R6, the authors provided the empirical results on the test set. It could show the obvious difference between classification and DFL; however, it is unclear if other theoretical analyses still hold.

**Limited scope and applicability (ozw4, bo89):**
The results are confined to supervised, closed-set, normalized regimes. Reviewers questioned the relevance to modern feature learning settings (self-supervised, contrastive, open-set, or downstream-task-agnostic representation learning), and these concerns were not fully resolved.

**Experimental validity controversy (bo89):**
Reviewer bo89 explicitly argued that some experimental findings regarding the optimal scale factor contradict established theory and prior empirical results, and suggested that insufficient training or limited architectural variation may explain the discrepancies. This disagreement remained unresolved.

**Practical limitations (bo89):**
Theoretical reliance on simplex/ETF structures limits applicability to large-class regimes; the authors offered conjectures but no concrete solution.

**Reviewer Scores:**

Reviewer pBJ8 (8 $\rightarrow$ 8):
Strongly positive; concerns were addressed and did not affect the score.

Reviewer 62s4 (4 $\rightarrow$ 4):
Concern about the scope.

Reviewer ozw4 (2 $\rightarrow$ 2):
Remained unconvinced about novelty, relevance to feature learning, and generalization; (explicitly indicated)

Reviewer bo89 (2 → 2):
Maintained rejection due to perceived lack of novelty and unresolved experimental and theoretical disagreements.

---

### Decision · Program_Chairs · 2026-01-26

Reject